# Colonic spatial single-cell proteomics and murine models link mitochondrial dysfunction to dimeric IgA-secreting plasma cell deficiency in Crohn's disease

Annika Raschdorf [1] ✉, Larissa Nogueira de Almeida [1], Philipp Solbach[2], Martha M. Kirstein[2], Jens U. Marquardt [2], Franziska Schmelter[1], Ulrich L. Günther [3], Heidi Schlichting[1], Maren Hicken[1], Lea Christiansen [1], Miriam Wiestler[4], Hauke C. Tews[5], Dominik Bettenworth[6,7], OUTLIVE-CRC consortium*, Matthias Peipp[8], Thomas Valerius [9], Mohab Ragab [1,10], Thorben Sauer[11], Timo Gemoll[11], Marc Ehlers [1,12], Philip Rosenstiel [13], Rudolf A. Manz [14], Axel Künstner [15,16], Hauke Busch[15,16], Christian Sina[1,2] & Stefanie Derer [1] ✉

Secretory IgA (SIgA) is critical for maintaining the intestinal barrier. A dysregulated B-cell compartment and altered Ig secretion have been well documented in Crohn's disease (CD) patients, although their origin is unknown. To unravel the role of mucosal humoral immunity in CD pathogenesis, we in-depth phenotype colonic plasma cell (PC) differentiation in CD at the single-cell level, linked to ex vivo functional characterization and experimental mouse models with a congenital mitochondrial defect or under glucose-free high-protein dietary intervention. Here, we demonstrate that despite expanded colonic B cells, CD patients in remission present significantly diminished mucosal dimeric IgA and fecal SIgA. Colonic plasmablasts and immature CD19+CD45+ PCs are increased at the expense of the mature CD19-CD45- phenotype. Accordingly, CD-derived ex vivo differentiated PCs display impaired maturation into dimeric IgA-secreting PCs. In this study, patient-derived data from colonic RNA-seq, spatial single-cell proteomics, and plasma metabolomics are combined with data from both mouse models and highlight the crucial role of mitochondrial oxidative phosphorylation in colonic IgA+-PC differentiation, suggesting promising directions for future therapeutic strategies.

Crohn's disease (CD) is one of the major entities of inflammatory bowel disease (IBD) with globally rising incidence rates, particularly in newly industrialized countries[1]. It is characterized by chronic, relapsing transmural inflammation, which can affect any portion of the gastrointestinal tract[2].

Many studies revealed an impaired intestinal barrier function in CD patients, resulting in the chronic translocation of luminal, commensal bacteria or their products across the intestinal mucosa, thereby triggering an aberrant immune response[3]. An integral part of the mucosal barrier is the secretory IgA (SIgA)-mediated humoral

A full list of affiliations appears at the end of the paper. *A list of authors and their affiliations appears at the end of the paper.
✉e-mail: Annika.Raschdorf@uksh.de; Stefanie.Derer@uksh.de

immunity primed in gut-associated lymphoid tissues (GALTs) such as Peyer's patches (PPs) or lymphoid follicles (LFs).

In the human intestine, next to immature IgA⁺CD138⁻ plasmablasts (PBs), three distinct IgA⁺CD138⁺ subsets of plasma cells (PCs) have been identified: proliferating, early CD19⁺CD45⁺, intermediate CD19⁻CD45⁺, and terminal mature CD19⁻CD45⁻ CD138⁺ PCs[4]. Secreted joining (J) chain-coupled dimeric IgA1 and IgA2 (dIgA) is transported across the epithelium by the polymeric Ig receptor (pIgR)[5] and anchored as SIgA to the mucus layer. In turn, it affects enteric bacteria and fungi through mechanisms such as immune exclusion/inclusion, antigen neutralization, enchained growth, and functional modulation[6–8]. Numerous studies have demonstrated that deficiencies in the production or function of IgA lead to intestinal dysbiosis and exacerbate spontaneous or trigger-induced inflammation in mouse models[9,10]. Similarly, selective IgA deficiency (SIgAD), the most common primary immunodeficiency in humans, is associated with a >5-fold increased prevalence of CD[11].

Dynamic metabolic adaptations accompany the development of the B-cell lineage. While activated, proliferating B cells mainly rely on aerobic glycolysis, differentiation into Ig-secreting PCs highly depends on mitochondrial oxidative phosphorylation (OXPHOS)[12]. Although less well studied compared to ulcerative colitis (UC), CD has been associated with mitochondrial dysfunction[13], potentially affecting this switch in bioenergetics.

As early as the 1980s, several publications indicated that B cells and humoral immunity were involved in CD pathology[14,15]. In 1984, Danis et al. showed an altered profile of Ig classes secreted by cultivated rectal biopsies of active CD patients, with increased IgM but diminished IgA secretion compared to non-IBD[16]. MacDermott et al. reported that cultivated peripheral blood mononuclear cells (PBMNCs) from CD patients displayed highly increased spontaneous IgA secretion[15]. In contrast, isolated lamina propria mononuclear cells (LPMNCs) from CD patients exhibited decreased IgA secretion and an increased percentage of monomeric IgA1 compared to non-IBD controls[15]. Similarly, another study verified comparable serum IgA levels but substantially reduced SIgA in fluids of the uninvolved jejunum of CD patients[17]. Complementing the expansion of the B-cell proportion in inflamed ileum[18], our previous study revealed increased ileal IgM⁺ B cells in CD remission (CD^rem)[19]. Recently, an enrichment of IgA⁺Ki-67⁺ PBs and IgG⁺ PCs and a proportional depletion of mature IgA⁺CD19⁻ PCs were observed in the inflamed colonic lamina propria (LP) of IBD patients[20,21]. However, in the past decades, findings linked to B cells have been attributed a bystander role in the etiopathogenesis of IBD.

Here, we aimed to build on previous observations of a skewed B-cell response in CD patients and employ state-of-the-art techniques to characterize the colonic humoral immunity in CD. We demonstrate an expansion of B cells, PBs, and immature CD19⁺CD45⁺ PCs in the colonic mucosa, accompanied by impaired terminal CD19⁻CD45⁻ PC differentiation and reduced dIgA secretion in these patients. Furthermore, our data suggest that a compromised mitochondrial function may underlie the defective terminal maturation of IgA-secreting PCs.

## Results

### Colonic CD27⁺CD38⁺CD138⁻ B cells are expanded in CD^rem patients

Colon biopsies were collected from CD^rem patients, as well as from age- and sex-matched non-IBD controls during colonoscopy (detailed information is given in Supplementary Tables 1–3). The colonic immune cell compartment was characterized using RNA sequencing (RNA-seq), two independent spatial single-cell proteomics (SSCP) experiments, and quantification of B-cell marker transcripts in isolated LPMNCs by RT-qPCR (Fig. 1a). Although principal component analysis (PCA) did not reveal a clear separation of CD^rem patients from non-IBD controls (Supplementary Fig. 1a), differential gene expression (DGE)

analysis revealed significantly increased expression of various Ig gene segments of the heavy and light chain in the colonic mucosa of CD^rem patients (Fig. 1b). Correspondingly, gene set enrichment analysis (GSEA) identified B cell-related pathways as significantly enriched within the top 40 REACTOME pathways (Fig. 1c). Transcripts highly expressed in B lymphocytes, plasmablasts (PBs) and PCs, such as CD79A (part of the B-cell receptor complex), MZB1, CD27, CD38, and TNFRSF17 (BCMA), as well as human leukocyte antigen (HLA) transcripts, involved in the major histocomatibility complex (MHC) class II antigen presentation, were significantly upregulated in CD^rem (Fig. 1d). Notably, the expression of SDC1 (CD138), a common marker of PCs but not PBs, was not increased in CD^rem. In line with this, SSCP using colonic formalin-fixed paraffin-embedded (FFPE) tissues revealed a trend towards a higher percentage of CD19⁺ (P = 0.0931) (non-IBD: 6.5%, interquartile range (IQR): 3.6–10.4% versus CD^rem: 16.3%, IQR: 9.9–22.6%), and significantly increased percentages of CD27⁺ (non-IBD: 22.9%, IQR: 19.6–25.6% versus CD^rem: 39.1%, IQR: 29.2–45.0%) and CD38⁺ cells (non-IBD: 22.0%, IQR: 18.3–33.2% versus CD^rem: 44.9%, IQR: 31.8–58.9%) in the total colonic LP immune cells (Vimentin⁺, lymphoid follicles/aggregates excluded) in CD^rem (Fig. 1e, f). Notably, the number of Vim⁺ LP cells per tissue area [mm²] was not increased in colonic biopsies from CD^rem patients, indicating that increased B-cell numbers are not attributable to leukocyte infiltration (Supplementary Fig. 1b). In contrast, the proportion of CD138⁺ PCs was not altered, nor was the percentage of non-B-cells characterized by expression of CD3, CD4, or CD8 (mostly T cells), CD11c (mostly dendritic cells (DCs)), and CD56 (mostly natural killer (NK)) cells. For further characterization of the colonic B-cell compartment, LPMNCs were isolated from respective patients (Fig. 1g). Total LPMNC count as well as plasma CRP levels did not differ between CD^rem and non-IBD (Supplementary Fig. 1c, d), highlighting the inactive disease state of the analyzed CD patients. In contrast, CD patients in active disease had increased colonic LPMNC counts compared to CD^rem patients and elevated plasma CRP levels relative to non-IBD controls (P = 0.0557) (Supplementary Fig. 1c, d). In agreement with our previous observations, colonic LPMNCs from CD^rem presented increased expression levels of CD38, and to a lesser extent, CD19 and CD27 (Fig. 1h and Supplementary Fig. 1e). Strikingly, expression of TNFRSF17 and SDC1 was not increased in CD^rem-derived LPMNCs. Together, these findings indicate that CD19⁺CD27⁺CD38⁺ B cells are overrepresented in the colonic mucosa of CD^rem but do not translate into an increased amount of CD138⁺ PCs (Fig. 1i).

### CD^rem displays increased colonic IgA but reduced fecal IgA

Given the expanded colonic B-cell compartment in CD^rem, we analyzed Ig classes in these patients in depth. While mass spectrometry of plasma samples revealed no quantitative differences in the presence of circulating Ig subclasses (Fig. 2a and Supplementary Fig. 1f), RNA-seq uncovered an elevated colonic expression of IGHA1, IGHA2, IGHG1 (P = 0.0556), IGHG2, and IGHM in CD^rem (Fig. 2b). Increased amounts of total colonic IgA and a trend towards elevated IgM, comprising intracellular, membranous as well as secreted forms, could be verified on protein level (Fig. 2c and Supplementary Fig. 1g). In line with the upregulation of TNFRSF17 (Fig. 1d), which is specifically expressed in PBs and PCs (Supplementary Fig. 1h), the amount of colonic B-cell maturation antigen (BCMA) was significantly higher in CD^rem compared to non-IBD (Fig. 2c). Additionally, IHC suggested increased IgA levels in the colonic LP of CD^rem (P = 0.0756) (Fig. 2d).

Intriguingly, fecal IgA levels were significantly diminished in a large cohort (n = 44) of CD^rem patients (Fig. 2e). Since transcytosis of IgA across the intestinal epithelium and formation of SIgA depend on the basolateral pIgR (Fig. 2f), we studied its expression by RNA-seq and IHC. Levels of the colonic epithelial pIgR, however, were not different between CD^rem and non-IBD (Supplementary Fig. 2a, b). Further, SSCP indicated an increased abundance of CD19⁺CD27⁺CD38⁺CD20⁻CD138⁻ PBs in the colonic mucosa of CD^rem patients (Fig. 2g and

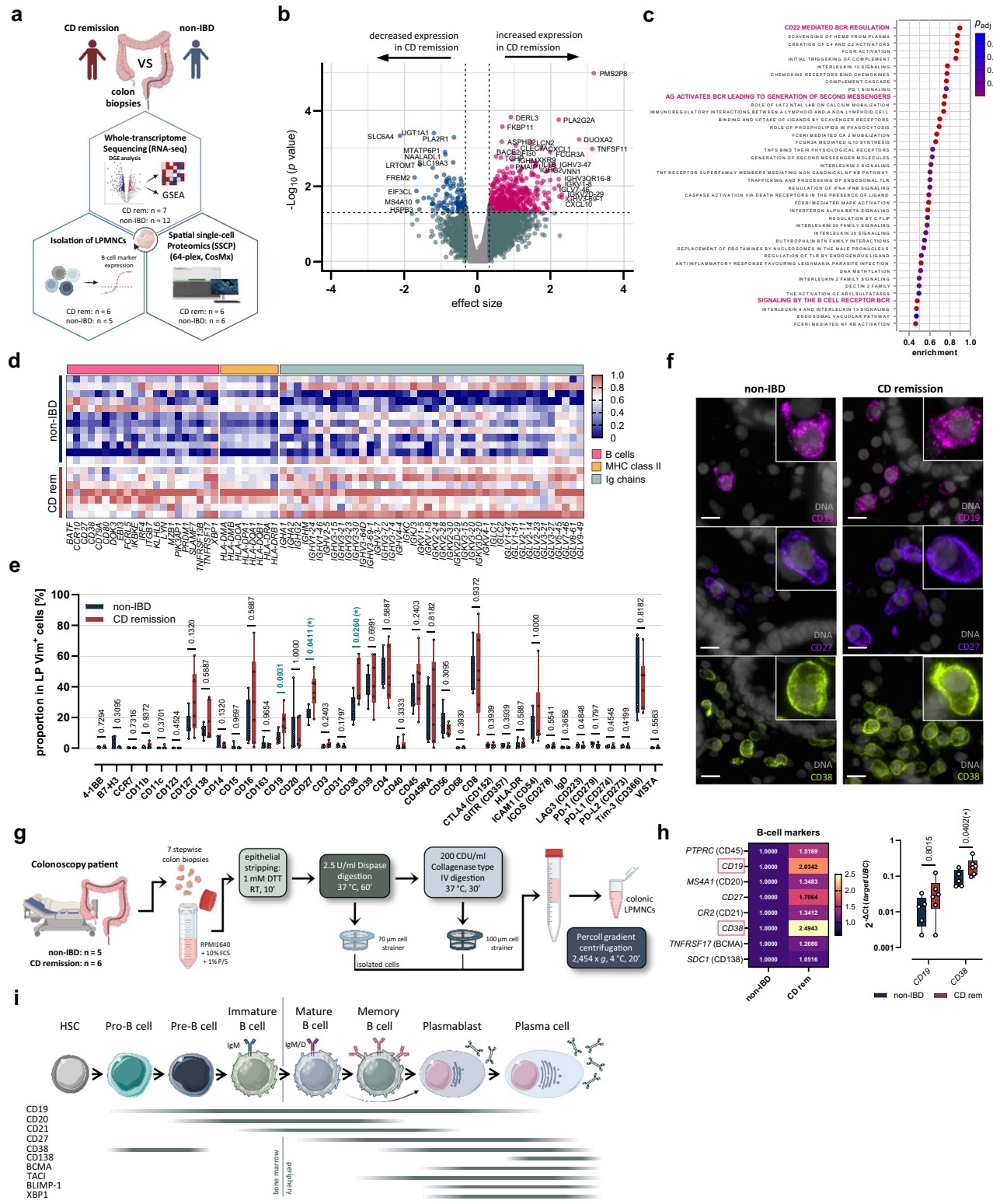

Supplementary Fig. 2c). Nevertheless, the proportion of colonic CD138[+] PCs was not upregulated in CD[rem], pointing to an impairment in local PC differentiation.

## Shift to proliferating CD19[+]CD45[+] PCs is associated with reduced dIgA secretion in CD

Assuming a defect in colonic PC maturation, we phenotyped colonic PCs using SSCP (Fig. 3a) to assess potential functional differences between CD[rem] and non-IBD in two independent experiments. The reclustering of 28,895 CD138[+] PCs identified within the pooled LP cells (Vim[+]) of CD[rem] and non-IBD (Fig. 3b) demonstrated a distinct phenotype of colonic PCs in CD[rem] (Fig. 3c and Supplementary Fig. 2d). It has been described that human intestinal CD138[+] PCs can be subdivided into three major subsets defined by selective expression of CD19 and CD45[4]. While CD19[+]CD45[+] PCs represent an early stage in the PC differentiation pathway and are described as immature and more

**Fig. 1 | Colonic B cells and PBs but not CD138⁺ PCs are expanded in CDʳᵉᵐ. a** RNA-seq, SSCP, and quantification of B-cell marker mRNA expression in isolated LPMNCs were performed utilizing colonic biopsies from age- and sex-matched CDʳᵉᵐ patients and non-IBD controls. **b** Volcano plot of differentially expressed genes (DEGs; pink: increased, blue: decreased) identified by RNA-seq (CDʳᵉᵐ *n* = 7, non-IBD *n* = 12) displaying unadjusted *P* values. **c** Top 40 upregulated REACTOME pathways according to gene set enrichment analysis (GSEA). The color of the dots corresponds to the adjusted *P* value. **d** Significantly upregulated DEGs linked to B cells (pink), MHC class II complex (orange), or immunoglobulin (Ig) chains (turquoise). **e** Percentage of cells positive for different immune cell markers in the colonic LP of CDʳᵉᵐ patients (*n* = 6) versus non-IBD controls (*n* = 6) was determined using SSCP. **f** Representative images of SSCP of colonic mucosa from CDʳᵉᵐ patients (*n* = 6, right) versus non-IBD controls (*n* = 6, left) for CD19 (magenta), CD27 (purple), and

CD38 (greenish yellow), and DAPI (gray). Scale bar: 10 μm. **g** LPMNCs were isolated from colon biopsies of CDʳᵉᵐ patients (*n* = 6) and non-IBD controls (*n* = 5) by enzymatic digestion. **h** Quantification of B-cell markers in CDʳᵉᵐ-derived (*n* = 6) colonic LPMNCs by RT-qPCR is displayed as median relative to non-IBD (*n* = 5) (left). *CD19* and *CD38* mRNA expression was increased >2-fold in colonic LPMNCs isolated from CDʳᵉᵐ patients (right). **i** Scheme of a simplified differentiation pathway of the B-cell lineage, including crucial surface markers. **e** Two-tailed multiple Mann−Whitney *U*-test. **h** Two-way ANOVA with Šídák's multiple comparisons test. Box plots depict the median and IQR with whiskers indicating the range. *P ≤ 0.05. IQR interquartile range, SSCP spatial single-cell proteomics. Parts of (**a, g, i**) were created in BioRender. Jäschke, S. (2026) https://BioRender.com/0mfac2x. Some elements in (**i**) were provided by Servier Medical Art (https://smart.servier.com).

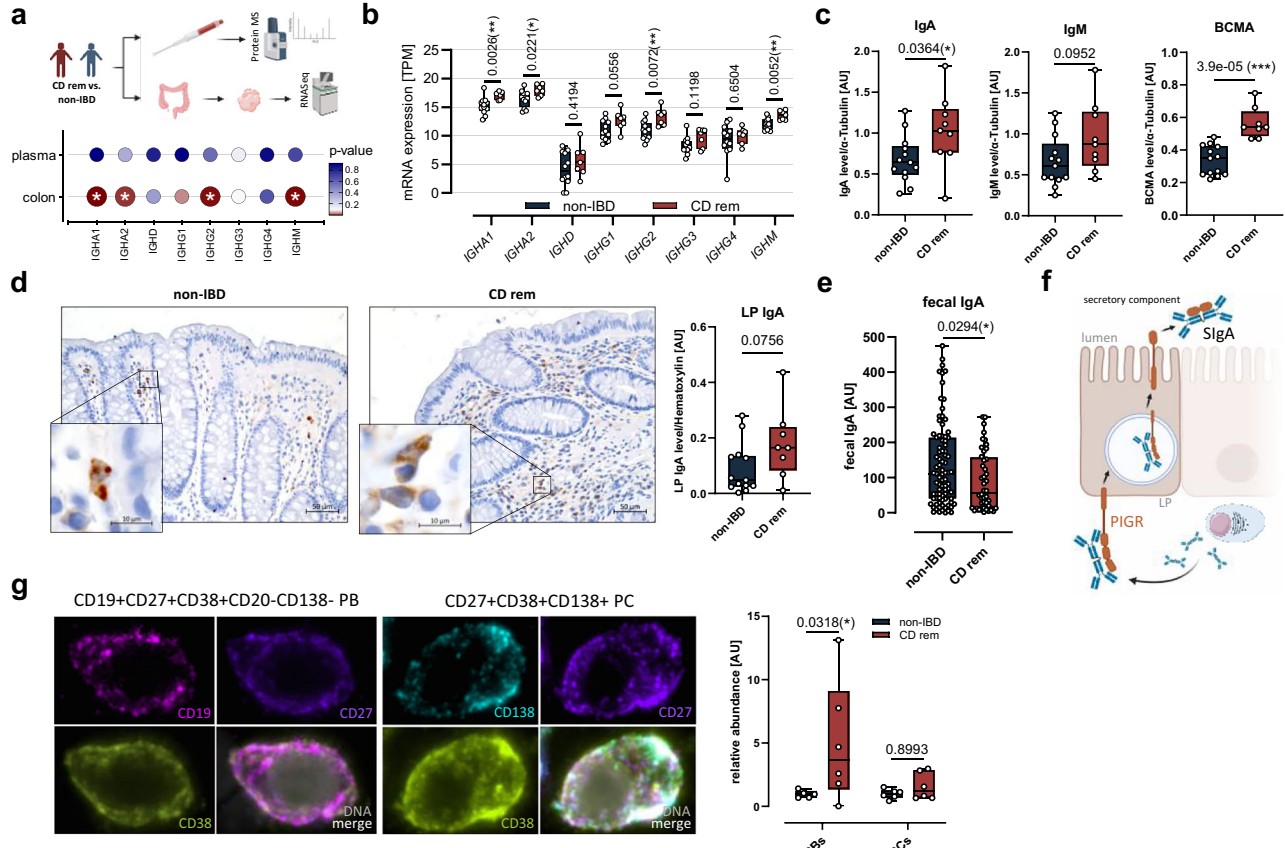

**Fig. 2 | CDʳᵉᵐ patients display a higher abundance of colonic CD19⁺CD27⁺CD38⁺CD138⁻ PBs but not CD138⁺ PCs and reduced fecal IgA levels. a** Immunoglobulin levels were quantified in blood plasma (CDʳᵉᵐ *n* = 10, non-IBD *n* = 21) and colonic tissue (CDʳᵉᵐ *n* = 7, non-IBD *n* = 12) using LC-MS proteomics or RNA-seq, respectively. Circle color reflects the *P* value of the Mann−Whitney *U*-test comparing CDʳᵉᵐ versus non-IBD, with larger circles indicating significant differences (*P ≤ 0.05). **b** Colonic RNA-seq data of Ig heavy chains of CDʳᵉᵐ patients (*n* = 7) and non-IBD controls (*n* = 12) are displayed as transcripts per million (TPM). **c** Quantification of colonic IgA, IgM, and BCMA in CDʳᵉᵐ patients (*n* = 9) versus non-IBD controls (*n* = 13) via Western blotting. **d** Representative images of IHC staining of colonic IgA (CDʳᵉᵐ *n* = 8, non-IBD *n* = 13) (left). Total IgA staining in the LP was normalized to nucleic hematoxylin staining (right). **e** Fecal SIgA levels were

quantified by IgA-specific ELISA experiments (CDʳᵉᵐ *n* = 44, non-IBD *n* = 85). **f** Schematic representation of epithelial IgA transcytosis and SIgA formation. **g** Representative images of SSCP of a plasmablast (PB) and a PC showing CD19 (magenta), CD27 (purple), CD38 (greenish yellow), CD138 (cyan), and DAPI (gray) (left). Relative abundance of CD19⁺CD27⁺CD38⁺CD138⁻ PBs, and CD138⁺ PCs in the colonic LP of CDʳᵉᵐ patients (*n* = 6) compared to non-IBD (*n* = 6; median set to 1) (right). **a–e** Two-tailed (Multiple) Mann−Whitney *U*-test. **g** Two-way ANOVA with Šídák's multiple comparisons test. Box plots depict the median and IQR with whiskers indicating the range. *P ≤ 0.05, ***P ≤ 0.001. IQR interquartile range, SSCP spatial single-cell proteomics. Parts of (**a, f**) were created in BioRender. Jäschke, S. (2026) https://BioRender.com/0mfac2x.

proliferative with a lower antibody secretion activity and a short lifespan[4,22], mature, long-lived CD19⁻CD45⁻ PCs display a high capacity for antibody secretion[4,22] (e.g., IgA) (Fig. 3d). Using SSCP, we validated the presence of these PC subsets in the human colonic mucosa of non-IBD controls (Fig. 3d). While CD138 expression was only slightly reduced in CD19⁻CD45⁺ and CD19⁻CD45⁻ PCs, the proportion of Ki-67⁺

cells was significantly reduced in CD19⁻CD45⁺ PCs compared to CD19⁺CD45⁺ PCs (Fig. 3e). In line with this, NF-κB p65 as well as pan-RAS, which are involved in activating signaling pathways, were downregulated stepwise during PC differentiation in CD19⁻CD45⁺ and CD19⁻CD45⁻ PCs, respectively (Fig. 3e). The cell size tended to increase slightly from CD19⁺CD45⁺ to CD19⁻CD45⁻ PCs, which might reflect the

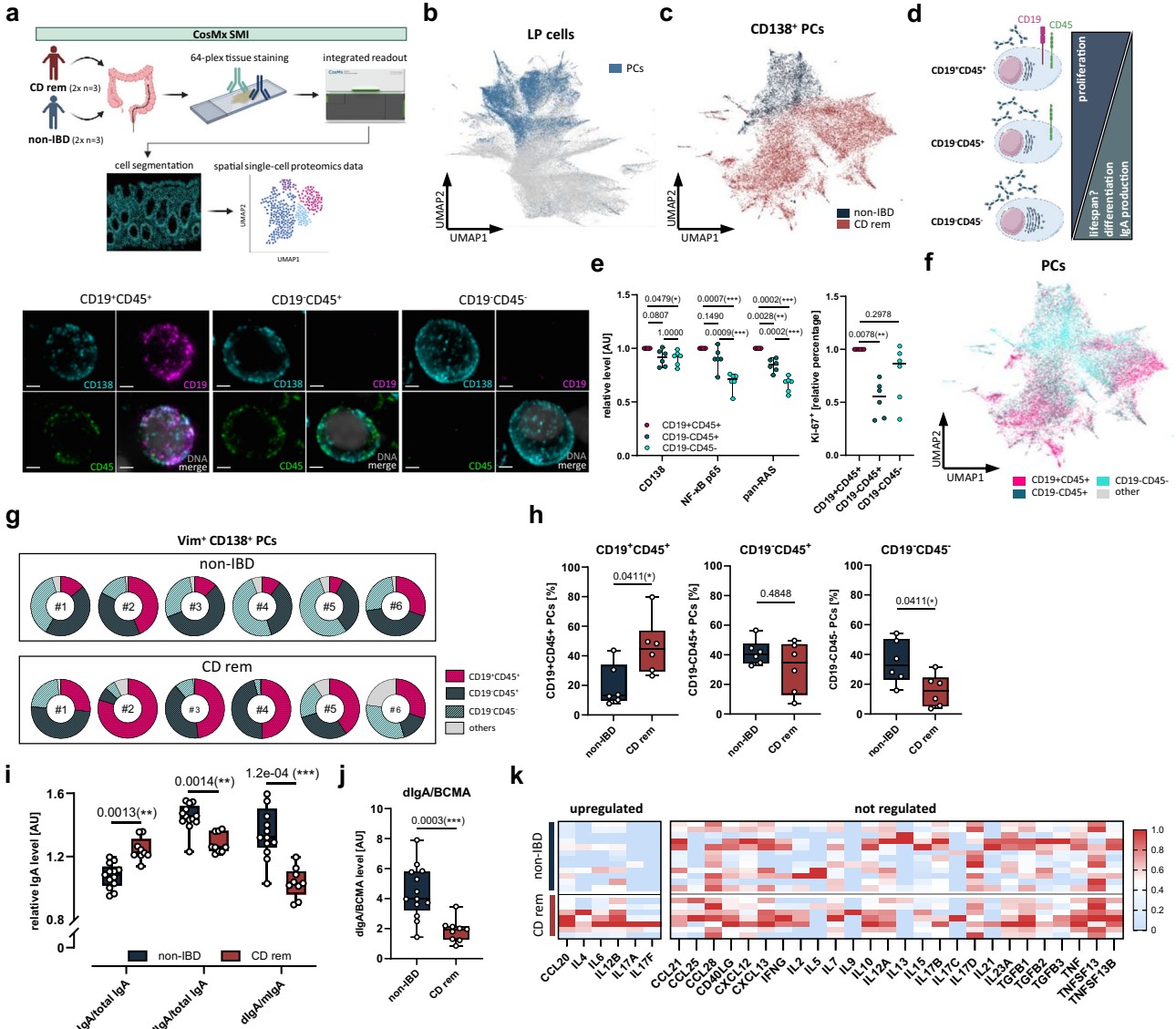

**Fig. 3 | Colonic low dIgA-secreting CD19⁺CD45⁺CD138⁺ PCs dominate over well-differentiated CD19⁻CD45⁻CD138⁺ PCs in CDʳᵉᵐ patients. a** Colonic PCs of CDʳᵉᵐ patients (n = 6) and non-IBD controls (n = 6) were characterized using the 64-plex Immuno-Oncology protein panel on the CosMx™ spatial molecular imager (SMI) in two independent experiments. **b** A total of 28,895 CD138⁺ PCs (blue) were identified in a UMAP of all colonic lamina propria (LP) cells (Vim⁺) of the first spatial single-cell proteomics experiment and **c** reclustered in a UMAP with coloring for non-IBD controls (n = 3) and CDʳᵉᵐ patients (n = 3). **d** Scheme comparing characteristics of intestinal IgA-secreting CD138⁺ PC subpopulations[4,22,34]. Representative images of mucosal CD19⁺CD45⁺ (left), CD19⁻CD45⁺ (middle), and CD19⁻CD45⁻ PCs (right) (CD138: cyan, CD19: magenta, CD45: green, DAPI: gray). Scale bar: 2 µm. **e** Relative levels of CD138, NF-κB p65, pan-RAS, and relative percentage of Ki-67⁺ cells in mucosal PC subpopulations of six non-IBD controls, related to the CD19⁺CD45⁺ PC population per patient. **f** UMAP displaying the CD138⁺ PC subtype distribution in colon biopsies from non-IBD controls (n = 3) and CDʳᵉᵐ patients (n = 3). **g** Fractions of CD19⁺CD45⁺ (magenta), CD19⁻CD45⁺ (dark green), CD19⁻CD45⁻ (turquoise) and other (gray) Vim⁺CD138⁺ PCs in colon biopsies of non-

IBD controls (n = 6; top) and CDʳᵉᵐ patients (n = 6; bottom). **h** Percentages of colonic PC subpopulations in CDʳᵉᵐ patients (n = 6) compared to non-IBD controls (n = 6). **i** Quantification of mIgA/total IgA, dIgA/total IgA, and dIgA/mIgA ratios in colonic biopsies of CDʳᵉᵐ patients (n = 9) and non-IBD controls (n = 13) via Western blotting. **j** The ratio of dIgA/BCMA in the colonic mucosa of CDʳᵉᵐ (n = 9) versus non-IBD (n = 13) was quantified using Western blotting. **k** Upregulated (left; P ≤ 0.05) and nonregulated (right) cytokines and chemokines crucial for B-cell/PC development were quantified using RNA-seq and are displayed as scaled transcripts per million (TPM) in the colonic mucosa of CDʳᵉᵐ patients (n = 7) versus non-IBD controls (n = 12). **e** Two-way ANOVA with Tukey's multiple comparisons test (left) and Friedman test with Dunn's multiple comparisons test (right), **h**–**k** Two-tailed (multiple) Mann–Whitney U-test. *P ≤ 0.05, **P ≤ 0.01, ***P ≤ 0.001. Box plots depict the median and IQR with whiskers indicating the range. AU arbitrary units, IQR interquartile range, UMAP uniform manifold approximation and projection. Parts of (**a**) were created in BioRender. Jäschke, S. (2026) https://BioRender.com/0mfac2x.

expansion of organelles necessary for high-level antibody secretion, such as the rough endoplasmic reticulum (rER) and the Golgi apparatus (Supplementary Fig. 2e). Moreover, the fully mature CD19⁻CD45⁻ PCs were found in closer proximity to the epithelium compared to the CD19⁺CD45⁺ and CD19⁻CD45⁺ populations, with no significant differences between CDʳᵉᵐ patients and non-IBD controls (Supplementary

Fig. 2f). Of note, SSCP identified a clear shift in these three phenotypically different colonic PC populations in CDʳᵉᵐ (Fig. 3f, g and Supplementary Fig. 2g, h). In CD patients, the presence of colonic CD19⁺CD45⁺ PCs was significantly increased (44.6%, IQR: 29.4–57.1%) in comparison to non-IBD controls (13.2%, IQR: 9.6–33.8%), while the percentage of colonic CD19⁻CD45⁻ PCs was decreased (15.5%, IQR:

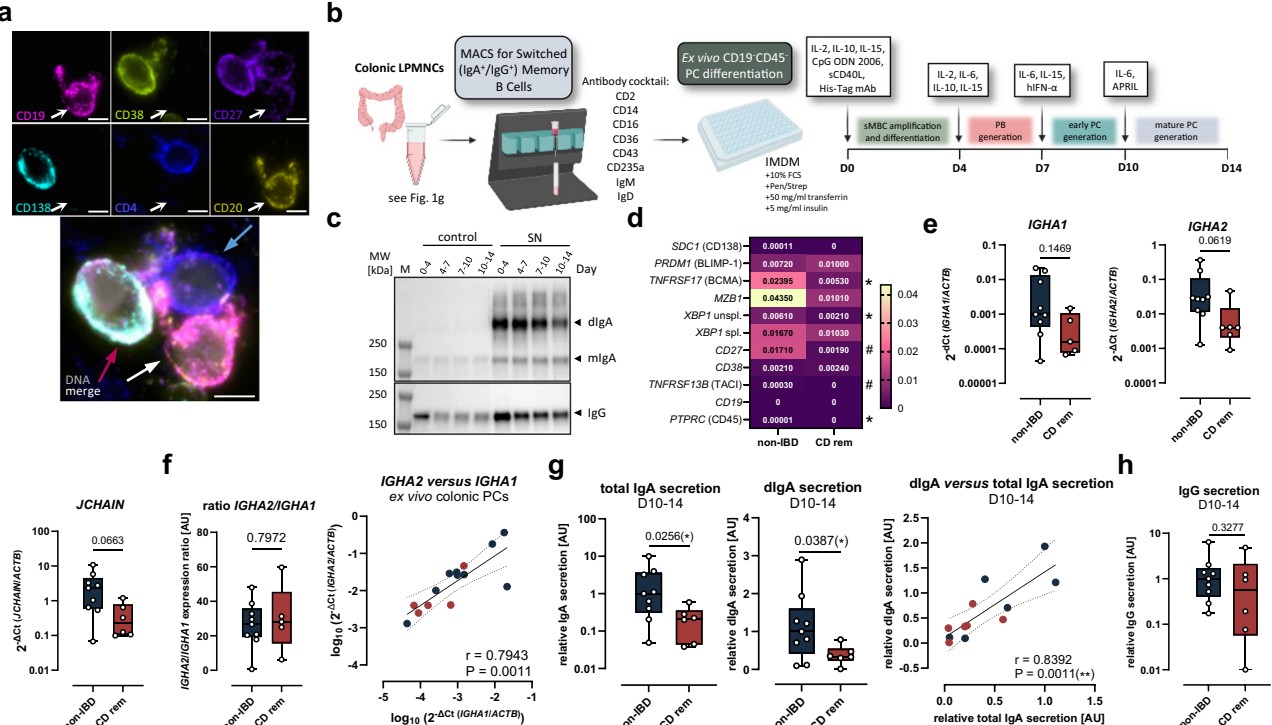

**Fig. 4 | Ex vivo IgA-secreting PC differentiation is impaired for CD^rem-derived colonic switched memory B cells (sMBCs). a** Representative images from six patients each in two independent SSCP experiments, showing a PC (red arrow) and a memory B cell (white arrow) making contact to a CD4⁺ T cell (blue arrow) in the colonic mucosa displaying CD19 (magenta), CD38 (greenish yellow), CD27 (purple), CD138 (cyan), CD4 (blue), CD20 (yellow) and DAPI (gray). Scale bar: 5 μm. **b** SMBCs were enriched from patient-derived colonic LPMNCs (see Fig. 1g; CD^rem patients *n* = 6, non-IBD controls *n* = 9) using negative selection magnetic-activated cell sorting (MACS) and were ex vivo differentiated into mature PCs for 14 days. **c** Secretion of dIgA, mIgA, and IgG by ex vivo differentiating PCs during Day 0–4, Day 4–7, Day 7–10, and Day 10–14 of cultivation (lanes 5-8) was demonstrated by Western blot experiments with respective IMDM samples (+cytokines) (lanes 1–4) as controls. **d** Expression of PC differentiation markers in patient-derived ex vivo differentiated colonic PCs on Day 14 was quantified using RT-qPCR and depicted in a heatmap as median values for CD^rem (*SDC1n* = 2, *PRDM1/TNFRSF17/XBP1* unspl./ *XBP1* spl./*CD27/CD38/CD19/PTPRCn* = 5, *MZB1n* = 6, *TNFRSF13Bn* = 4) and non-IBD (*SDC1/CD27/TNFRSF13B/PTPRCn* = 7, *PRDM1/TNFRSF17/XBP1* unspl./*XBP1* spl./

*CD38/CD19n* = 8, *MZB1n* = 9). **e** Expression of *IGHA1*, *IGHA2*, and *JCHAIN* for CD^rem- (*IGHA1n* = 5, *IGHA2/JCHAINn* = 6) versus non-IBD-derived (*n* = 9) ex vivo differentiated colonic PCs. **f** Ratio (left) and correlation (right) of *IGHA1* and *IGHA2* expression in CD^rem- (*n* = 5) and non-IBD-derived (*n* = 9) ex vivo differentiated colonic PCs on Day 14. **g** Normalized quantification (left) and correlation (right) of total and dimeric IgA secretion by ex vivo differentiated colonic PCs from CD^rem patients (*n* = 6) and non-IBD controls (*n* = 9; median set to 1), measured between Day 10–14 using ELISA experiments. **h** Normalized IgG secretion by CD^rem-derived ex vivo differentiated colonic PCs (*n* = 6) between Day 10–14 compared to non-IBD-derived PCs (*n* = 9; median set to 1) was quantified by Western blotting. **d**–**f**, **h** Two-tailed (multiple) Mann–Whitney *U*-test. **f**, **g** Spearman correlation and linear regression with 95% confidence interval (dashed lines). **g** Two-tailed Mann–Whitney *U*-test (total IgA) and *t*-test with Welch's correction (dIgA). Box plots depict the median and IQR with whiskers indicating the range. ^#*P* ≤ 0.1, **P* ≤ 0.05. IQR interquartile range, SSCP spatial single-cell proteomics. Parts of (**b**) were created in BioRender. Jäschke, S. (2026) https://BioRender.com/0mfac2x.

5.1–24.4%) compared to non-IBD controls (32.7%, IQR: 23.0–50.5%) (Fig. 3h). In accordance with the observation that the less mature CD19⁺CD45⁺ PCs dominate over the fully differentiated CD19⁻CD45⁻ PCs in CD^rem, the proportion of dimeric IgA (dIgA) relative to total IgA or monomeric IgA (mIgA) was reduced in the colonic mucosa of these patients (Fig. 3i and Supplementary Fig. 2i). In contrast, the proportion of mIgA was increased, likely reflecting an expansion of IgA⁺ B cells. Further, the significantly reduced dIgA/BCMA ratio indicated a decreased secretion of dIgA per PB/PC in the colonic mucosa of CD^rem (Fig. 3j). Next, we studied the colonic tissue microenvironment of these PCs by bulk RNA-seq regarding the expression of cytokines known to promote PC development and survival. Here, *IL4* and *IL6* were upregulated with respect to non-IBD, while *IL10*, *IL21*, *TGFB1* (TGF-β1), and *TNFSF13* (APRIL) were not significantly altered in CD^rem (Fig. 3k and Supplementary Fig. 2j). Similarly, chemokines such as *CCL25*, *CCL28*, and *CXCL12* that enable homing or retention of primed IgA⁺ PBs in the colon[23,24] were not altered in these patients, suggesting that PC differentiation may rather be impaired intrinsically and independently of the mucosal microenvironment. Taken together, our analyses demonstrated a compromised PC differentiation in CD^rem that

is reflected by an increase in immature colonic CD19⁺CD45⁺ PCs at the expense of terminally differentiated CD19⁻CD45⁻ PCs and diminished intestinal secretion of dIgA.

### CD patient-derived switched memory B cells display an intrinsic defect in ex vivo IgA⁺ PC differentiation

Conceptualizing that the local differentiation of colonic PCs may be intrinsically affected in CD^rem, we employed an ex vivo model of PC differentiation. Since the majority of intestinal PCs originate from memory B cells[25] (Fig. 4a), we enriched class-switched IgA⁺/IgG⁺ memory B cells (sMBCs) from isolated colonic LPMNCs (Fig. 4b and Supplementary Fig. 3a). Subsequently, the sMBC-enriched population, mainly expressing B-cell lineage markers like *MS4A1* (CD20), *CD27*, *CD38*, and *TNFRSF17* (Supplementary Fig. 3b), was cultivated for 14 days and differentiated into terminal PCs according to a protocol published by ref. 26 (Fig. 4b). The presence of antibody-secreting PBs and PCs was verified by the detection of high amounts of dIgA as well as moderate levels of mIgA and IgG in the supernatant over the time of cultivation (Fig. 4c). Lower levels of mIgA and IgG detected in the medium control (IMDM + cytokines) likely result from non-specific

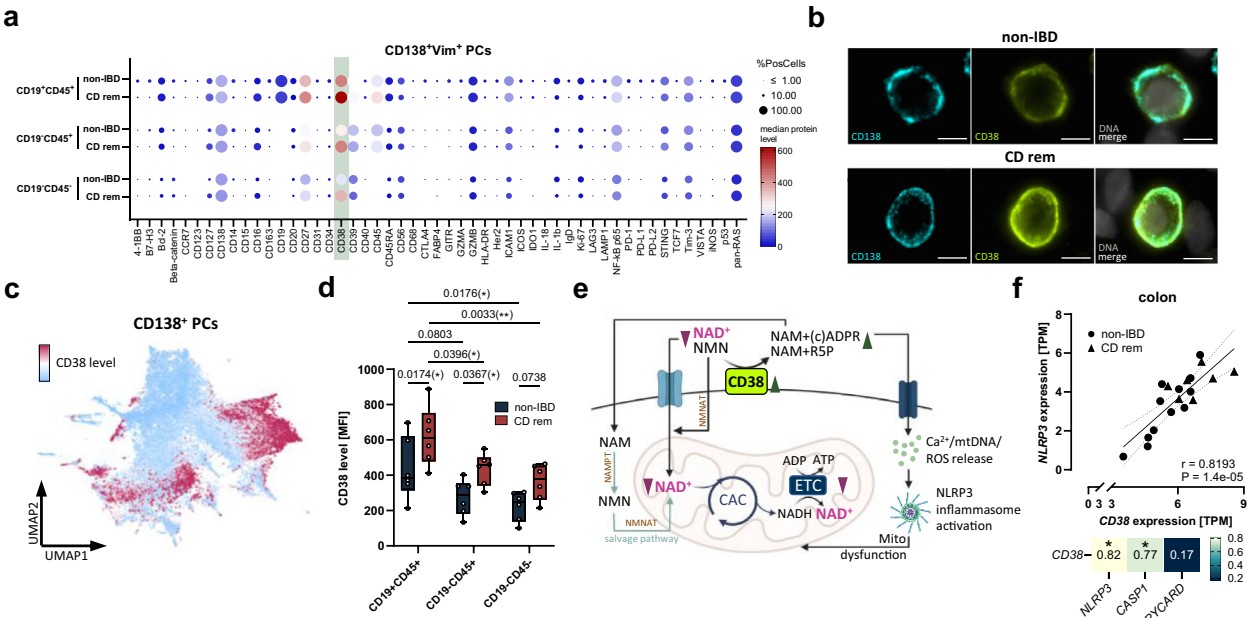

**Fig. 5 | Colonic PCs of CD^rem patients display increased surface levels of the NADase CD38. a** SSCP data of colonic CD19⁺CD45⁺, CD19⁻CD45⁺, and CD19⁻CD45⁻ PCs (CD^rem patients *n* = 6, non-IBD controls *n* = 6). Results are presented in a dotplot as median protein levels per cell (circle color) and percentage of positive cells (circle size). **b** Representative images of CD38 staining (greenish yellow) in a colonic CD138⁺ PC of CD^rem (*n* = 6) versus non-IBD (*n* = 6). Scale bar: 5 μm. **c** UMAP displaying the minimum (blue) and maximum (magenta) CD38 levels within all colonic PCs of the first SSCP experiment (CD^rem patients *n* = 3, non-IBD controls *n* = 3). **d** Quantification of CD38 levels in CD19⁺CD45⁺, CD19⁻CD45⁺, and CD19⁻CD45⁻ colonic PCs of CD^rem patients (*n* = 6) compared to non-IBD controls (*n* = 6) using SSCP. Data were presented as median CD38 level per cell. **e** Schematic representation of mitochondrial NAD⁺ metabolism, showing how increased NAD⁺-hydrolyzing activity of surface CD38 contributes to mitochondrial dysfunction.

**f** Correlations of *CD38* with *NLRP3*, *CASP1*, and *PYCARD* expression in the colon of CD^rem patients (*n* = 7) and non-IBD controls (*n* = 12) displaying Spearman correlation coefficients. **d** Two-way ANOVA with Tukey's multiple comparisons test, **f** Spearman correlation and linear regression with 95% confidence interval (dashed lines). Box plots depict the median and IQR with whiskers indicating the range. *$P \le 0.05$, **$P \le 0.01$. CAC citric acid cycle, (c)ADPR (cyclic) ADP-ribose, ETC electron transport chain, IQR interquartile range, MFI median fluorescence intensity, NAD⁺/NADH nicotinamide adenine dinucleotide, NAM nicotinamide (a form of vitamin B3), NAMPT Nicotinamide phosphoribosyltransferase, NMN nicotinamide mononucleotide, NMNAT Nicotinamide mononucleotide adenylyltransferase, R5P ribose 5-phosphate, ROS reactive oxygen species, SSCP spatial single-cell proteomics, UMAP uniform manifold approximation and projection. Parts of (**e**) were created in BioRender. Jäschke, S. (2026) https://BioRender.com/0mfac2x.

cross-reaction with mIgA and IgG in the fetal bovine serum (FBS) or with the murine IgG1 His-Tag antibody added on Day 0 for cross-linking the histidine-tagged CD40 ligand. On Day 14, the CD^rem- and non-IBD-derived PCs were compared regarding their expression of common PC markers (Fig. 4d and Supplementary Fig. 3c, d). In contrast to non-IBD-derived PCs, PCs derived from CD^rem lacked detectable expression of *SDC1*, indicating that they were unable to develop into fully mature PCs. Additionally, the expression of *TNFSRF17*, X-box binding protein 1 (*XBP1*), *CD27* (*P* = 0.0732), and *PTPRC* (CD45) was reduced in patient-derived ex vivo-generated PCs (Fig. 4d and Supplementary Fig. 3c). Unlike PCs of most non-IBD controls, expression of *TNFRSF13B*, encoding the Transmembrane activator and CAML interactor (TACI), could not be detected in PCs of CD^rem origin. While *IGHM* was not affected (Supplementary Fig. 3e) and *IGHA1* tended to be slightly reduced (Fig. 4e), the expression of *IGHA2* and the J chain (*JCHAIN*), which is crucial for the formation of dIgA, approached statistical significance in its reduction in CD^rem-derived PCs (Fig. 4e). The similar *IGHA2/IGHA1* ratio and the strong positive correlation between both isoforms (Fig. 4f) are consistent with a preserved IgA subclass composition in CD^rem- compared to non-IBD-derived colonic PCs. In line with these results, CD patient-derived PCs displayed significantly decreased secretion of total and dimeric IgA into the supernatant (Fig. 4g). The comparable dIgA/total IgA ratio (Supplementary Fig. 3f) excludes a compensatory increase in mIgA secretion by CD^rem- versus non-IBD-derived PCs, which is further supported by the positive correlation between dIgA and total IgA (Fig. 4g). The expression of *IGHG1*, *IGHG2*, *IGHG3*, and *IGHG4* tended to be slightly lower in patient-derived PCs with no significant differences (Supplementary Fig. 3g). However,

IgG secretion did not differ between PCs originating from non-IBD or CD^rem (Fig. 4h). Concisely, the data indicate an intrinsic defect in the differentiation of IgA⁺ colonic PCs from sMBCs of CD^rem patients, leading to an impaired capacity for dIgA secretion, particularly dIgA2.

## PCs in CD patients show increased levels of the NADase CD38 and mitochondrial dysfunction

To unravel underlying molecular mechanisms of impaired colonic PC differentiation in CD patients, we further phenotyped distinct PC subtypes (CD19⁺CD45⁺, CD19⁻CD45⁺, and CD19⁻CD45⁻) by SSCP using FFPE tissue slides of six CD^rem patients and six non-IBD controls. Three proteins were specifically regulated during PC maturation and differed between CD patients and controls. A slight increase in CD27 and CD45 was observed, while CD38 displayed the strongest upregulation in all three colonic PC subtypes in CD^rem (Fig. 5a, b). In line with this, the CD38^high PCs were found primarily in the CD^rem PC cluster identified previously (Fig. 5c and Supplementary Fig. 4a). Although nearly all PCs were CD38⁺ (Fig. 5a) and CD38 expression decreased during PC differentiation ranging from CD19⁺CD45⁺ to CD19⁻CD45⁻ PCs, median CD38 expression per cell was increased in CD^rem versus non-IBD for each PC subpopulation (Fig. 5d). Moreover, CD38 levels positively correlated with levels of the activating signaling proteins NF-κB and pan-RAS across all PC subtypes (Supplementary Fig. 4b). CD38 is known to act as a cell surface NADase/ADP-ribosyl cyclase as well as a cyclic ADPR (cADPR) hydrolase that catalyzes the synthesis of nicotinamide (NAM) and (c)ADPR using nicotinamide adenine dinucleotide (NAD⁺) as a substrate[27] (Fig. 5e). NAD⁺, which can be synthesized from tryptophan or various vitamin B3 derivatives, is crucial for

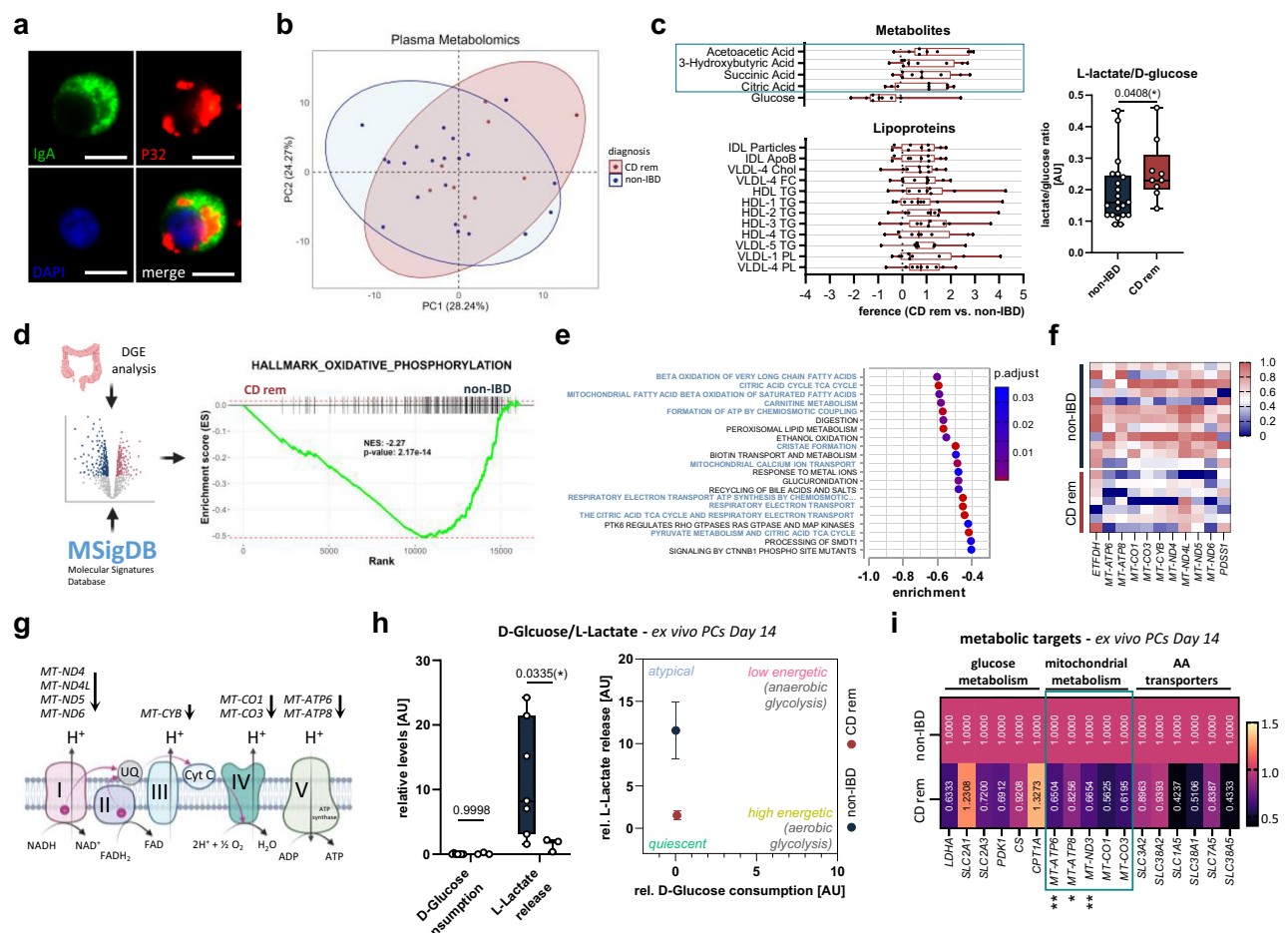

**Fig. 6 | Mitochondrial OXPHOS is impaired in CD patient-derived colonic PCs.**
**a** Representative image from three biological replicates showing immunofluorescence co-staining of a colonic PC for IgA and OXPHOS-promoting mitochondrial P32. Scale bar: 5 μm. **b** PCA of metabolites measured in plasma samples of CD[rem] (red, *n* = 9) and non-IBD (blue, *n* = 22) individuals using in vitro diagnostic research (IVDr) proton nuclear magnetic resonance (¹H NMR) spectroscopy.
**c** Significantly regulated plasma metabolites (top left) and lipoproteins (bottom left) in CD[rem] (*n* = 9; bars) compared to the reference average of non-IBD controls (*n* = 22; dotted center line), with upregulated ketone bodies and CAC intermediates highlighted in a blue box. Plasma L-lactate/D-glucose ratio was determined for CD[rem] (*n* = 9) versus non-IBD (*n* = 22) (right). **d** Gene set enrichment analysis (GSEA) of colonic RNA-seq data for the MSigDB "HALLMARK_OXIDATIVE_PHOSPHORYLATION" gene set in CD[rem] (*n* = 7) versus non-IBD (*n* = 12). NES normalized enrichment score.
**e** Downregulated REACTOME pathways in the colon of CD[rem] patients (*n* = 7) versus non-IBD controls (*n* = 12) according to GSEA. The color of the dots corresponds to the adjusted *P* value. **f** Significantly downregulated colonic gene transcripts linked to mitochondrial function in CD[rem] (*n* = 7) compared to non-IBD (*n* = 12) are displayed as scaled transcripts per million (TPM). **g** Schematic overview of all significantly downregulated colonic transcripts encoding subunits of the mitochondrial electron transport chain (ETC) complexes I, III, IV, and V in CD[rem]. **h** Normalized D-glucose consumption and L-lactate release were quantified in the supernatant of CD[rem]- (*n* = 3) and non-IBD-derived ex vivo differentiated colonic PCs between Day 10–14.
**i** Expression of metabolic enzyme(s) (subunits) and nutrient transporters in non-IBD- (*n* = 9) and CD[rem]-derived (*n* = 5) ex vivo differentiated colonic PCs on Day 14 was quantified using RT-qPCR and depicted as median relative to non-IBD, with mitochondrial genes highlighted in a green box. **c, f** Two-tailed (multiple) Mann–Whitney *U*-test, **h, i** Two-way ANOVA with Šídák's multiple comparisons test (**h**) or Fisher's least significant difference (LSD) test (**i**). Box plots depict the median and IQR with whiskers indicating the range. *P ≤ 0.05. IQR interquartile range. Parts of (**d, g**) were created in BioRender. Jäschke, S. (2026) https://BioRender.com/0mfac2x.

---

mitochondrial energy metabolism. Accordingly, several studies linked CD38 overexpression to the depletion of intracellular NAD⁺ levels and subsequent mitochondrial dysfunction[28,29]. Moreover, the formation of cADPR can induce a Ca²⁺ release-mediated activation of the NOD-like receptor family pyrin domain containing 3 (NLRP3) inflammasome, further exacerbating mitochondrial dysfunction[30]. In support of this, colonic *CD38* expression correlated significantly with the expression of the inflammasome components *NLRP3* and *CASP1* (Fig. 5f).

As mitochondrial respiration - supported by high levels of mitochondrially localized, OXPHOS-promoting P32 (Fig. 6a)—is essential for proper differentiation and function of PCs[12], we investigated the systemic as well as the local energy metabolism in the colonic mucosa of CD[rem] patients. Proton nuclear magnetic resonance (¹H NMR)-based

metabolomics detected no difference in the global metabolic profile of plasma samples from CD[rem] and non-IBD (Fig. 6b). However, elevated levels of ketone bodies and the significantly increased L-lactate/D-glucose ratio (Fig. 6c and Supplementary Fig. 4c) may indicate an upregulation of ketogenesis and glycolytic activity in CD[rem], respectively, both of which are known compensatory mechanisms in response to OXPHOS deficiency[31]. Higher succinic acid and citric acid levels might reflect a blockage of the citric acid cycle (CAC). Moreover, CD patients displayed a dyslipidemia with several upregulated lipoprotein fractions (Fig. 6c), which has often been linked to mitochondrial dysfunction[32,33]. In accordance with these systemic metabolic observations, GSEA of the "HALLMARK_OXIDATIVE_PHOSPHORYLATION" gene set using colonic tissue RNA-seq data demonstrated a significant depletion of OXPHOS-related genes in CD[rem] (Fig. 6d). In detail, GSEA

identified enrichment of various REACTOME pathways involved in mitochondrial ATP synthesis, β-oxidation, CAC, and electron transport chain (ETC) among the downregulated gene transcripts (Fig. 6e). Differential gene expression analysis uncovered multiple transcripts related to mitochondrial function that were significantly reduced in the colonic mucosa of CD[rem] (Fig. 6f) and correlated negatively with *CD38* expression (Supplementary Fig. 4d). The majority of these targets are mitochondrially encoded subunits of the complexes I, III, IV, and V of the ETC (Fig. 6g). Further transcriptional analyses revealed ex vivo differentiated PCs to display low expression of the glucose transporters *SLC2A1* and *SLC2A3*, but higher levels of the amino acid transporters *SLC3A2* and *SLC38A2* (Supplementary Fig. 4e, f), which facilitate the uptake of neutral amino acids, including branched-chain amino acids and glutamine—both of which serve as critical substrates for PC metabolism[34]. These findings were validated in functional metabolic analyses, where CD[rem] patient-derived PCs exhibited a very low consumption of D-glucose and a significantly reduced L-lactate secretion compared to non-IBD-derived ex vivo differentiated colonic PCs (Fig. 6h). Consistent with a significantly decreased expression of the mitochondrially encoded genes *MT-ATP6*, *MT-ATP8*, and *MT-ND3* (Fig. 6i and Supplementary Fig. 4g), these findings highlight a metabolically quiescent state of CD[rem] patient-derived PCs. Together, our data indicate a dysfunction in mitochondrial energy metabolism in the colonic mucosa of CD[rem] patients that potentially plays a causative role in defective colonic terminal PC differentiation and dIgA production in these patients.

## Mitochondrial dysfunction is associated with impaired IgA⁺ PC maturation in mice

To examine whether mitochondrial dysfunction may causally impede PC differentiation in vivo, we investigated the maturation of PBs and PCs in conplastic respiratory chain complex V mutant mice. These mice carry a mutation in the mitochondrially encoded ATP synthase membrane subunit 8 (ATP8), resulting in diminished respiratory capacity and ATP production with a compensatory increase in glycolysis in various cell entities, including lymphocytes[35] (Fig. 7a), and display an impaired intestinal barrier homeostasis[36,37]. Splenic resting B cells isolated from adult *Atp8*-mutant versus B6-WT (wild type) mice ($n = 3$ per group) were ex vivo differentiated into IgG1⁺ PBs under stimulation with lipopolysaccharide (LPS) and IL4 for four days (Fig. 7b and Supplementary Fig. 5a, b). Using the mitochondrial membrane potential-sensitive dye tetramethylrhodamine ethyl ester (TMRE), flow cytometry revealed a significant increase in mitochondrial activity in LPS + IL4-stimulated PBs (mean: 54.2% TMRE[high]) compared to resting B cells (mean: 17.1% TMRE[high]) obtained from B6-WT mice. As expected, the percentage of TMRE[high] PBs under LPS + IL4 stimulation was significantly lower in *Atp8*-mutant-derived B cells (Fig. 7c and Supplementary Fig. 5a). However, *Atp8*-mutant B cells displayed a significant boost in proliferation upon activation and PB induction in contrast to B6-WT B cells (Fig. 7d). While murine PBs exhibit high levels of surface B220, B220 expression is downregulated in short-lived PCs (SLPCs) and lost completely in most fully mature PCs[22]. Hence, the trend towards elevated B220 expression in *Atp8*-mutant PBs may indicate a less differentiated phenotype compared to B6-WT PBs (Fig. 7e). In line with this, splenic IgG levels were increased in *Atp8*-mutant mice under chronic DSS-induced colitis (Supplementary Fig. 5c), whereas expression of *Sdc1* (CD138) was significantly diminished compared to B6-WT mice (Supplementary Fig. 5d).

Additionally, we isolated cells from mesenteric lymph nodes (MLNs) and PPs for flow cytometry analysis from three animals per group (Supplementary Fig. 5e). While the percentages of B220⁺ B cells and CD138⁺ PCs in MLNs and PPs did not differ between *Atp8*-mutant and B6-WT mice (Supplementary Fig. 5f), the proportion of B220⁻ PCs was decreased in MLNs and PPs of *Atp8*-mutant mice (MLNs: 21.6%, IQR: 20.3–22.3% and PPs: 24.3%, IQR: 18.0–31.3%) compared to B6-WT

mice (MLNs: 30.8%, IQR: 21.4–52.3% and PPs: 34.2%, IQR: 28.6–42.1%) (Fig. 7f and Supplementary Fig. 5g). Consistent with our previous observations, colonic expression of B-cell and PB/PC differentiation markers, including *Cd19*, *Igha*, *Tnfrsf17*, *Prdm1*, and *Sdc1*, was significantly reduced in *Atp8*-mutant mice (Fig. 7g and Supplementary Fig. 5h). Notably, expression of *complement component 1q binding protein* (*C1qbp*), encoding p32, which is essential for maintaining mitochondrial OXPHOS[38], was decreased in *Atp8*-mutant mice (Supplementary Fig. 5d) and significantly correlated with colonic expression of *Igha*, *Prdm1*, and *Sdc1* (Fig. 7h, i). In line with this, the percentage of colonic IgA⁺ cells tended to be decreased in *Atp8*-mutant (0.5%, IQR: 0.5–0.7%, $n = 3$) compared to B6-WT mice (3.0%, IQR: 2.2–4.5%, $n = 3$) (Fig. 7j), accompanied by significantly diminished levels of mIgA and dIgA in the colonic mucosa (Supplementary Fig. 5i).

In a previous study[36], we demonstrated that removal of glucose from the diet and isocaloric replacement by the protein casein promotes colonic p32 levels and mitochondrial OXPHOS in mice (Fig. 7k). Indeed, adult C57BL/6 mice that were fed a glucose-free, high-protein (GFHP) diet for an average of 70 days displayed significantly upregulated colonic expression of *Cd19*, *Cd79a*, *Ighm*, and *Tnfrsf17* (Supplementary Fig. 5j), higher percentages of colonic IgA⁺ cells (2.7%, IQR: 2.1–3.3%) than isocaloric chow diet-fed mice (1.9%, IQR: 0.7–2.3%) (Fig. 7l), and increased fecal IgA levels (Supplementary Fig. 5k). In summary, these findings demonstrate that PC maturation and colonic humoral immunity are highly dependent on mitochondrial function and could potentially be promoted by nutritional intervention boosting OXPHOS efficiency.

## Discussion

Despite significant advances in IBD research, the multifactorial etiology of CD remains largely elusive, which highlights the unmet need for new concepts to develop causal therapies. SIgA is critical for shaping the microbial composition and maintaining homeostasis with the intestinal immune system to prevent chronic inflammation. Consistent with this, several cases of rituximab (anti-CD20)-induced ileocolonic CD suggest an anti-inflammatory role for B lymphocytes in the gut[39]. In recent decades, several publications have indicated alterations in B cells and humoral immunity in CD patients[14–19]. However, since most studies focused solely on inflamed intestinal tissue or did not distinguish between affected and unaffected areas, it is still unclear whether the observed dysregulation of the humoral immune system is a consequence or a cause of inflammation in CD. To overcome this issue, we here focused on patients with CD in remission to identify potential pathogenic changes in colonic humoral immunity underlying the disease.

CD patients have been previously described to display a hyperactive systemic[15,40] and intestinal (ileal/colonic) B-cell compartment[18,19], a finding confirmed by our observation of increased B-cell lineage targets, IgA⁺ cells, and PBs in the colonic mucosa. Unexpectedly, we detected decreased fecal IgA levels in CD[rem] patients, which is in line with small and large intestinal MNCs mainly from active IBD patients exhibiting reduced spontaneous IgA secretion[15] and diminished SIgA found in the fluid of the histologically non-involved jejunum of CD patients[17]. Multiple studies have indicated that murine strains deficient in IgA production or function (*e.g.*, AID, IgA, or pIgR KO) consistently develop intestinal dysbiosis and more severe spontaneous or trigger-induced inflammation[41–43]. Hence, reduced luminal SIgA levels would render CD patients more susceptible to mucosal invasion by microbiota due to compromised barrier integrity and exacerbate inflammation by failing to match the microbial antigenic load with a commensurate SIgA response. However, the comparable levels of both colonic PCs and epithelial pIgR led to the hypothesis that IgA-secreting PC differentiation may be impaired.

Landsverk et al. first described the distinction between early CD19⁺CD45⁺, intermediate CD19⁻CD45⁺, and fully mature late

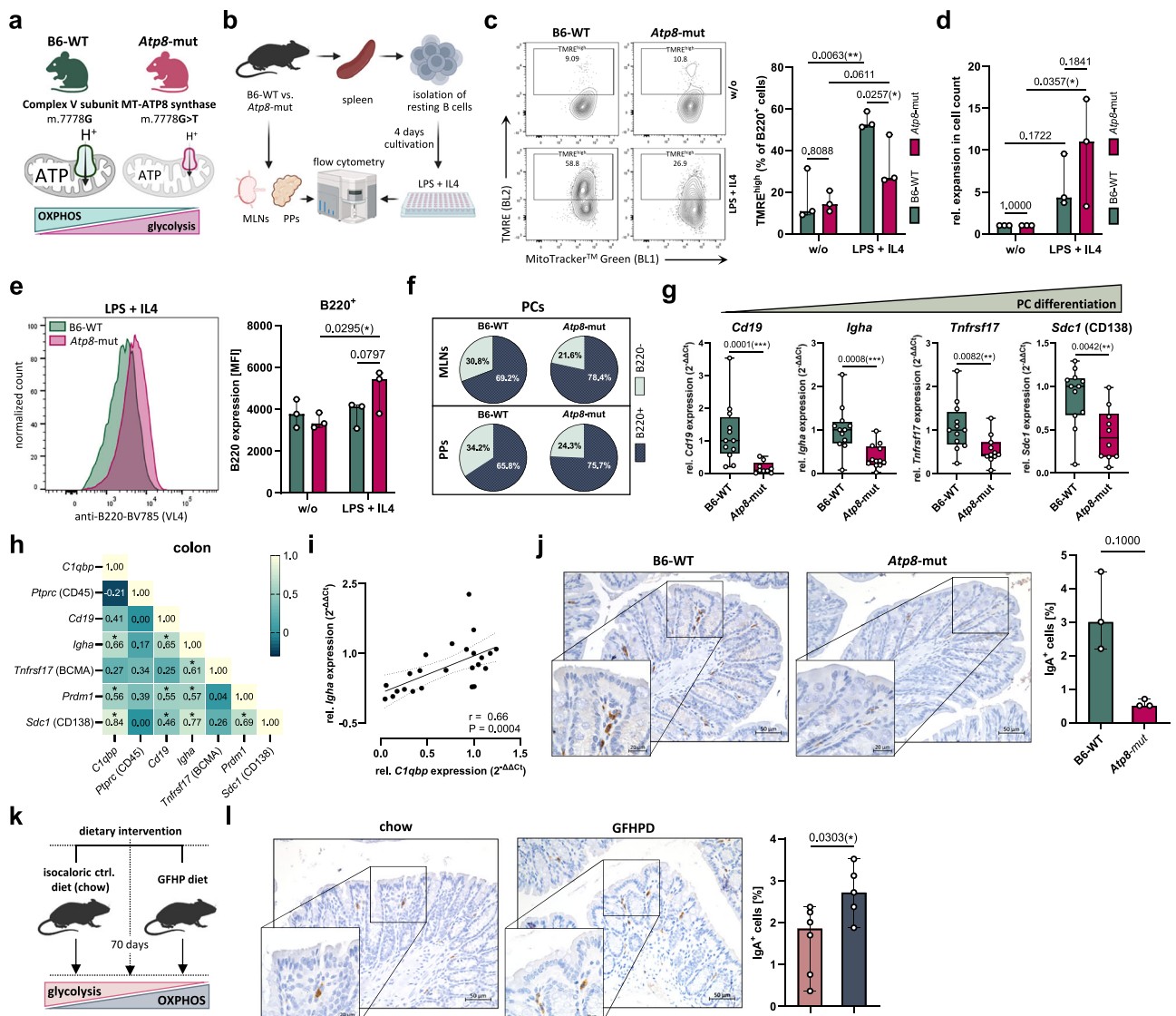

**Fig. 7 | Mitochondrial dysfunction impairs colonic PC differentiation and lowers IgA in mice. a** Mutation in subunit 8 of the ATP synthase and published metabolic imbalance[35,61]. **b** Schematic of flow cytometry of lymphocytes from mesenteric lymph nodes (MLNs) and Peyer's patches (PPs) and LPS + IL4-stimulated splenic B cells of male and female *Atp8*-mutant and B6-WT mice. **c** Representative contour plots of mitochondrial mass (MitoTracker Green) and membrane potential (TMRE, tetramethylrhodamine ethyl ester) (left). Percentage of TMRE^high cells in LPS + IL4-stimulated or unstimulated B220^+ cells from *Atp8*-mutant (*n* = 3) versus B6-WT mice (*n* = 3) (right) from two independent experiments. **d** Proliferation of *Atp8*-mutant (*n* = 3) and B6-WT (*n* = 3) splenic B cells under LPS + IL4 stimulation. **e** B220 level of ex vivo differentiated PBs from *Atp8*-mutant mice (*n* = 3) versus B6-WT-derived PBs (*n* = 3). **f** Median percentages of B220^+ (dark blue) and B220^- (turquoise) CD138^+ PCs isolated from MLNs and PPs of *Atp8*-mutant (*n* = 3) and B6-WT mice (*n* = 3) in two experiments. **g** Colonic expression of B-cell/PC markers was quantified for male *Atp8*-mutant (*Cd19n* = 9, *Ighan* = 12, *Tnfrsf17n* = 11, *Sdc1n* = 10), and B6-WT mice (*n* = 12) via RT-qPCR. Data were displayed relative to B6-WT mice for three sampling rounds, with box plots

representing median and IQR and whiskers indicating the range. **h** Pairwise correlation matrix of colonic *C1qbp* and B-cell/PC marker expression for *Atp8*-mutant (*n* = 9–12) and B6-WT mice (*n* = 12) displaying Spearman correlation coefficients. **i** Correlation of colonic *Igha* and *C1qbp* expression in Atp8-mutant (*n* = 12) and B6-WT mice (*n* = 12). **j** Representative colonic IgA IHC staining of male *Atp8*-mutant (*n* = 3) and B6-WT mice (*n* = 3) (left) and quantification of IgA^+ cells (right), with mice sampled on two days. **k** Hypothesized metabolic switch upon glucose-free high-protein (GFHP) dietary intervention. **l** Representative colonic IgA IHC staining of female GFHP diet (*n* = 5) versus isocaloric chow diet-fed mice (*n* = 6) (left) and quantification of IgA^+ cells (right). from two independent experiments. **c**–**e** Two-way ANOVA with Tukey's multiple comparisons test (**c**, **d**) or Fisher's least significant difference (LSD) test (**e**). (**g**, **j**, **l**) Two-tailed Mann−Whitney *U*-test, **h** Spearman correlation, **i** Spearman correlation and linear regression with 95% confidence interval (dashed lines). Bar graphs display median with range. *$P \le 0.05$, **$P \le 0.01$, ***$P \le 0.001$. MFI mean fluorescence intensity. Parts of (**a**, **b**, and **k** were created in BioRender. Jäschke, S. (2026) https://BioRender.com/0mfac2x.

CD19^-CD45^- PCs in the human intestine[4]. CD19^-CD45^- PCs are often referred to as long-lived (LLPCs) with a higher capacity for antibody secretion compared to less differentiated short-lived CD19^+CD45^+ PCs (SLPCs)[22]. Recently, Vaquero et al. showed an increase in actively proliferating Ki-67^+ IgA^+ PBs and early short-lived CD19^+ IgA^+ PCs at the expense of long-lived late and terminal CD19^- IgA^+ and CD45^- IgA^+ PCs, respectively, in the inflamed colonic mucosa of active IBD patients[21].

Using SSCP, we here describe a similar expansion of PBs and CD19^+ PCs and a proportional decline in CD45^- PCs in the colonic LP of patients with CD in remission, suggesting that this shift may be pathogenic rather than a consequence of inflammation. In line with the observation that intestinal MNCs from CD patients secreted a higher percentage of mIgA than control intestinal MNCs[15], our findings indicate a decreased level of dIgA in the colonic mucosa of CD^rem. Potential

additional changes in IgA diversity and their impact on host-microbiome interactions warrant further investigation in subsequent studies. The local intrinsic defect in PC maturation is corroborated by our data showing strongly reduced expression of PC differentiation markers, such as *TNFRSF17*, *XBP1*, *CD27*, *TNFRSF13B*, *IGHA2*, and *JCHAIN*, in CD[rem]-derived ex vivo-generated colonic PCs, accompanied by diminished IgA secretion.

Moreover, SSCP revealed a significant upregulation of the cell surface NADase CD38 on mucosal PCs in CD[rem]. To rule out the contribution of age and senescence, which have been associated with elevated CD38 expression[29,44], we ensured that the analyzed patients and controls were age-matched. In addition, it is well established that CD38 is upregulated during intestinal inflammation[45], however, our analyses were confined to non-inflamed tissue. Overexpression of CD38 has been associated with depleted intracellular NAD$^+$/NADH levels and mitochondrial dysfunction in different in vitro[28,29] and ex vivo[46] cell models. Additionally, tryptophan catabolism is skewed in IBD patients, resulting in inefficient de novo NAD$^+$ synthesis[47]. Blood-derived CD38$^{hi}$CD8$^+$ T cells from systemic lupus erythematosus (SLE) patients, in parallel, also showed an increased percentage of depolarized mitochondria and a diminished basal and maximum oxygen consumption rate (OCR), disrupting cytotoxic effector functions[46]. In contrast, CD38 KO mice displayed elevated cellular and mitochondrial NAD$^+$ levels along with an increased OCR in multiple tissues[29] and alleviated dextran sodium sulfate (DSS)-induced colitis[45], similar to conplastic mice carrying OXPHOS-promoting mtDNA variants[48]. Accordingly, our findings indicate a downregulation of various mitochondria- and OXPHOS-related pathways in the colonic mucosa of CD[rem], in line with a reduced expression of many subunits of the ETC complexes in the corresponding ex vivo differentiated PCs. Moreover, generation of the second messenger cADPR during CD38-mediated NAD$^+$ cleavage could indirectly impair mitochondrial energy metabolism by activating the NLRP3 inflammasome–caspase-1 pathway, which can cleave the OXPHOS-maintaining protein P32 and boost aerobic glycolysis-driven cell proliferation[49]. Yet, further studies are needed to identify the mechanisms regulating CD38 expression on PCs.

Activation of naïve B cells, proliferation, and subsequent differentiation into PBs/PCs is accompanied by a progressive increase in oxidative metabolism[12], which we confirmed in murine splenic B cells ex vivo differentiated into PBs under LPS and IL4 stimulation. While fully mature LLPCs and early SLPCs show remarkable transcriptional similarity[34], increasing evidence indicates that metabolic pathways may impact PC differentiation and maintenance. Lam et al. demonstrated that LLPCs have a higher pyruvate-dependent mitochondrial respiratory capacity than SLPCs in vivo, which is required for maximal Ig secretion[22]. In line with this, treatment of ex vivo-generated PBs with the ATP synthase inhibitor oligomycin resulted in a decrease in antibody secretion[12]. Conversely, dichloroacetate-driven OXPHOS promotion in ex vivo activated B cells increased the frequency of PBs and decreased cell division[12]. Transfer experiments of mtDNA-depleted B cells into wild-type mice showed that these cells were outcompeted during germinal center (GC) development and PC differentiation, with the strongest effect in the most mature PC population[50]. Accordingly, preliminary data from *Atp8*-mutant mice (*n* = 3) suggested a reduction in mature B220$^-$CD138$^+$ PCs in MLNs and PPs, consistent with the impaired IgA-mediated humoral immunity observed in the colon. Decreased expression of mitochondrial OXPHOS-related genes, accompanied by reduced L-lactate secretion, suggests a quiescent metabolic state in CD[rem]-derived colonic PCs, highlighting the need to investigate their metabolic profile in more detail. Conversely, we observed that nutritional intervention with an isocaloric GFHP diet fostering OXPHOS activity enhanced IgA-mediated humoral immune responses in the colon of mice. Overall, the influence of therapeutic regimens on the dysregulated humoral immune response and metabolism in CD patients requires further investigation in follow-up studies involving larger patient cohorts.

In conclusion, our data indicate that defective terminal differentiation of IgA-secreting PCs is already present in non-inflamed colonic mucosa of CD in remission, presumably rendering these patients more susceptible to chronic inflammation. Furthermore, we deciphered CD38-mediated mitochondrial dysfunction as a potential mechanism driving impaired PC differentiation, paving the way for novel therapeutic approaches, including metabolic interventions.

## Methods

### Study cohort and sample collection
Protocols were approved by the ethics committees of the University of Lübeck (AZ 12-138; AZ 19-233; AZ 21-415), the University of Münster (AZ 2016-305-b-S), the University Hospital Regensburg (No. 21-2390-101) and the Hannover Medical School (No. 8738_B0_S_2019 (DRKS00019082); No. 10847_BO_S_2023 (DRKS00032771)) and written informed consent was obtained from participants before sample collection.

Blood samples were collected at the University Hospital Schleswig-Holstein, Campus Lübeck, Germany. Blood was drawn from fasting patients into a hirudin blood collection tube, centrifuged at 4122×*g* for 15 min, and plasma was stored at −80 °C. Stool samples were obtained from the University Hospital Schleswig-Holstein, Campus Lübeck, Germany; the University Hospital Münster, North Rhine-Westphalia, Germany; the University Hospital Regensburg, Bavaria, Germany, and the Hannover Medical School Hospital, Lower Saxony, Germany and stored at −80 °C. Stepwise colon biopsies were taken during endoscopy as part of regular patient management in the Department of Medicine I, University Hospital Schleswig-Holstein, Campus Lübeck, Germany. Biopsies were immediately fixed in 4.5% formaldehyde, snap-frozen in liquid nitrogen, and stored at −80 °C or placed in ice-cold RPMI for cell isolation. Clinical characteristics of CD patients and non-IBD controls included in this study are listed in Supplementary Tables 1–4. Due to the limited number of patients, analyses were not stratified by sex.

Patients with a confirmed diagnosis of CD and colonic involvement were classified as being in remission or active disease based on clinical and endoscopic parameters. CD patients for stool sample collection were defined as being in remission if the available parameters met the following criteria: Harvey–Bradshaw index (HBI) ≤3, CRP ≤5 mg/l, and fecal calprotectin <100 μg/g ±1 month around stool sampling. CD patients for blood draw and endoscopic removal of colon biopsies were classified as being in remission if they were in clinical remission, the colon presented without macroscopic evidence of mucosal inflammation, and the ileum showed no or at most minor inflammatory signs during endoscopy. The age- and sex-matched control group included patients without a diagnosis of IBD and who presented without macroscopic or histological evidence of mucosal inflammation if endoscopy had been performed ±1 month around sample collection. Non-IBD controls with autoimmune diseases such as primary sclerosing cholangitis (PSC) or multiple sclerosis (MS), prior organ transplantation, immunosuppressive medication, diverticulitis, or non-specific intestinal inflammation were excluded from data analysis.

Due to limitations in available patient sample material, experimental methods were applied to different, overlapping patient subgroups (Supplementary Fig. 6).

### Animal models
All animal experiments were approved by the Ministry of Energy, Agriculture, the Environment, Nature and Digitalization Schleswig-Holstein, Germany (mitochondrial mutation: V 242-63560/2017 [5-1/18] and V 312-72241.122-4 [93-10/10]; dietary intervention: V 242-

27664/2018 [64-5/17]) and performed in accordance with the ARRIVE guidelines. Mice were housed under specific pathogen-free conditions at 21 ± 2 °C and 55 ± 10% humidity on a regular 12-h light-dark cycle with ad libitum access to food (Altromin #1324, unless otherwise specified) and water.

C57BL/6J mice were purchased from the Jackson Laboratory and bred in the animal facility of the University of Lübeck. The conplastic strain C57BL/6J-mt[FVB/NJ], which carries a mutation in the mitochondrial gene encoding the ATP synthase membrane subunit 8 (*Atp8*), was generated as described previously[51] and maintained by repeatedly backcrossing female conplastic offspring with male C57BL/6J mice. Here, 3- to 6-month-old male and female *Atp8*-mutant mice (median age 14.9 weeks, IQR 13.3–16.0 weeks) and corresponding C57BL/6J control animals (B6-WT; median age 17.5 weeks, IQR 15.0–18.0 weeks) were euthanized for organ collection in independent experimental rounds. Due to variations in baseline mRNA expression levels of targets of interest, data were normalized to the median B6-WT target expression for each of three experiments. In a separate experiment, chronic colitis was induced in 3- to 4-month-old male and female *Atp8*-mutant and B6-WT mice (median age at organ harvest – *Atp8*-mutant: 13.3 weeks, IQR 13.3–13.3 weeks/B6-WT: 18.0 weeks, IQR 18.0–18.0 weeks) using three cycles of 5 days of 2% dextran sodium sulfate (DSS) in the drinking water, each followed by 5 days of regular drinking water for recovery. Due to the limited number of mice available from in-house breeding, sex-specific stratification was not performed.

Glucose-free high-protein (GFHP) and isocaloric control diets were obtained from Ssniff (Soest, Germany), with the respective compositions specified in Supplementary Table 5. Female C57BL/6 mice, aged 7 to 8 weeks, were purchased from Charles River Laboratories (Wilmington, MA) and were allowed to acclimate on a standard chow diet. At 20 weeks, mice were randomly assigned to groups receiving the GFHP diet or the isocaloric control diet. Mice were maintained on the respective diet for an average of 10 weeks before organ sampling. The dietary intervention was conducted in two independent experimental runs.

### Isolation of colonic lamina propria mononuclear cells (LPMNCs)
Seven stepwise biopsies were collected from the cecum to rectum during colonoscopy and immediately placed in ice-cold RPMI 1640 supplemented with 10% (v/v) fetal bovine serum (FBS), 100 U/ml penicillin, and 100 μg/ml streptomycin. Biopsies were centrifuged at 300×*g* and 4 °C for 5 min and treated with 1 mM dithiothreitol (DTT) at room temperature for 10 min to disintegrate the mucus layer and the extracellular matrix. Then, biopsies were centrifuged again (300×*g*, 4 °C, 5 min), digested with 2.5 U/ml Dispase (STEMCELL Technologies #07913) and 1% (w/v) BSA in PBS at 37 °C for 1 h, and isolated cells were collected by passage through a 70-μm cell strainer and placed on ice. For further digestion, the remaining tissue was incubated in pre-warmed 200 CDU/ml collagenase type IV (Sigma-Aldrich #C5138) in supplemented RPMI 1640 at 37 °C for 30 min, and isolated cells were separated using a 100 μm cell strainer. Both isolated cell fractions were pooled, centrifuged (300×*g*, 4 °C, 5 min), and resuspended in 5 ml of supplemented RPMI 1640. To purify lamina propria mononuclear cells (LPMNCs) by density gradient centrifugation, the cell suspension was layered onto 6 ml of 63% (v/v) Percoll solution and centrifuged at 2,454×*g* and 4 °C for 20 min without brake. The turbid layer of MNCs was removed, centrifuged (300×*g*, 4 °C, 5 min), resuspended in 1 ml supplemented RPMI 1640 and the number of living cells was determined in a counting chamber using trypan blue staining.

### Enrichment of colonic IgA+ and IgG+ switched memory B cells
IgA+ and IgG+ switched memory B cells (sMBCs) were enriched from colonic LPMNCs using negative selection magnetic-activated cell sorting (MACS; a cocktail of biotin-conjugated monoclonal antibodies against CD2, CD14, CD16, CD36, CD43, CD235a, IgM, and IgD) following the manufacturer's instructions of the Switched Memory B Cell Isolation Kit (Miltenyi #130-093-617).

### Ex vivo differentiation of colonic switched MBCs into PCs
The sMBC-enriched cell population in the flow-through was centrifuged at 300×*g* and 4 °C for 5 min, resuspended in 120 μl Iscove's Modified Dulbecco's Medium (IMDM) supplemented with 10% (v/v) FBS, 100 U/ml penicillin, 100 μg/ml streptomycin, 5 mg/ml human insulin, and 50 mg/ml human transferrin[26] and seeded in one round-bottom well of a 96-well suspension cell culture plate. Cells were cultivated at 37 °C and 5% $CO_2$ in a humidified incubator for 14 days, and differentiation into plasmablasts (PBs) and PCs was induced by sequentially adding different combinations of cytokines as established by ref. 26. SMBCs were initially cultured from Day 0 to 4 with IL-2 (20 U/ml, R&D Systems #202-IL), IL-10 (50 ng/ml, R&D Systems #217-IL), IL-15 (10 ng/ml, Thermo Fisher Scientific #200-15), CpG oligo-deoxynucleotide (ODN) 2006 (10 μg/ml, InvivoGen #tlrl-2006), histidine-tagged soluble recombinant human CD40L (50 ng/ml, R&D Systems #2706-CL), and anti-polyhistidine mAb (5 μg/ml, R&D Systems #MAB050) to induce B-cell activation. Between Day 4 and 7, PBs were generated in the presence of IL-2 (20 U/ml), IL-6 (50 ng/ml, Thermo Fisher Scientific, #200-06), and IL-15 (10 ng/ml), followed by PC differentiation from Day 7 to 10 induced by IL-6 (50 ng/ml), IL-15 (10 ng/ml), and IFNα (500 U/ml, Sigma-Aldrich #IF007). Stimulation with IL-6 (50 ng/ml) and APRIL (200 ng/ml, R&D Systems #5860-AP) between Day 10 and 14 promoted PC survival and terminal differentiation. Supernatants were collected on Day 4, 7, 10, and 14 after centrifugation at 300×*g* and 4 °C for 5 min and cells were washed with 100 μl of IMDM before fresh supplemented IMDM and cytokines were added.

### D-glucose assay
D-glucose levels were measured in supernatants of Day 10–14 (1:10 v/v) from patient-derived ex vivo differentiated colonic PCs using the D-glucose assay kit (Megazyme #K-GLUC) according to the manufacturer's instructions.

### L-lactate assay
L-lactate levels were quantified in supernatants of Day 10–14 (1:4 v/v) from patient-derived ex vivo differentiated colonic PCs following the manufacturer's instructions of the L-lactate assay kit (Megazyme #K-LATE).

### RNA extraction and RT-qPCR
Total RNA was extracted from isolated colonic LPMNCs, sMBCs, or ex vivo differentiated PCs (Day 14) following the manufacturer's instructions using the Single Cell RNA Purification Kit (Norgen Biotek #51800). Native frozen murine colon tissue was homogenized, and total RNA was isolated according to the innuPREP RNA Mini Kit 2.0 (IST Innuscreen #845-KS-2040010). The RNA was on-column treated with DNase I (Sigma-Aldrich #AMPD1-1KT), and its concentration was determined using the NanoDrop™ One[C] spectrophotometer. CDNA synthesis was performed using Oligo(dT)$_{18}$ primers and the RevertAid H Minus reverse transcriptase (Thermo Fisher Scientific #EP0451). Real-time quantitative polymerase chain reaction (RT-qPCR) was performed on the StepOnePlus™ or the QuantStudio™ 1 system (Thermo Fisher Scientific) using the PerfeCTa™ SYBR® Green SuperMix (Quantabio #95055-02 K) and target-specific oligonucleotides (Supplementary Table 6). Target expression levels were normalized to *UBC* or *ACTB/Actb* and displayed as $2^{-\Delta Ct}$ or $2^{-\Delta\Delta Ct}$ (relative expression to the control group).

### Bulk RNA sequencing of patient-derived colonic tissue
Native frozen colon biopsies were homogenized and total RNA was isolated following the manufacturer's instructions of the innuPREP

RNA Mini Kit 2.0 (IST Innuscreen #845-KS-2040010) including on-column DNA digestion using DNase I. RNA was quantified using the NanoDrop™ One$^C$ spectrophotometer (Thermo Fisher Scientific) and verified to have an RNA integrity number (RIN) >5 using the RNA 6000 Nano Kit and the 2100 Bioanalyzer instrument (Agilent Technologies #5067-1511). Bulk mRNA sequencing was performed at Novogene Corporation Inc. in Cambridge, United Kingdom. Following poly(A) enrichment and non-directional mRNA library preparation, paired-end 150 bp sequencing was performed to a depth of 30 million reads per sample on the NovaSeq X Plus platform (Illumina).

Demultiplexed FastQ files were mapped onto the human reference genome (GRCh38) using kallisto (v0.46.1) with sequence bias correction and 30 bootstraps per sample. Differentially expressed genes were identified by applying the R package sleuth (v0.30.1) on kallisto quantifications using a likelihood ratio test to calculate $P$ values. Effect sizes, which approximate $\log_2$ fold changes, were calculated using the Wald test implemented in sleuth. Genes with $P \leq 0.05$ and absolute effect sizes >1 were considered significantly different between conditions. Gene set enrichment analysis (GSEA) against REACTOME pathways (MSigDB v2023.2, Oct 2023) was performed on effect sizes using the R package mitch (v1.16.0). Pathways with $q$ values <0.001 and absolute enrichment scores >0.4 were considered significant. GSEA for the gene set HALLMARK_OXIDATIVE_PHOSPHORYLATION (MSigDB v2023.2) was conducted using the R package fgsea (v1.24.0).

### Protein isolation from stool samples and colon biopsies

For fecal protein extraction, stool samples were homogenized in a tenfold volume of native protein buffer (20 mM $Na_2HPO_4$, 650 mM NaCl, 1 mM EDTA, 1 mM PMSF; pH 7.4) and cleared by centrifugation at 20,000×$g$ and 4 °C for 10 min.

To isolate proteins from colon tissue, native frozen biopsies were homogenized in denaturing lysis buffer containing 1% (w/v) sodium dodecyl sulfate (SDS), 10 mM Tris (pH 7.4), 2% (v/v) phosphatase inhibitor 2 and 3 (Sigma-Aldrich #P5726 and #P0044), and 1% (v/v) protease inhibitor (Sigma-Aldrich #P8340). The homogenate was heated at 100 °C for 5 min, treated with ultrasonication, and cleared by centrifugation at 12,000×$g$ and 4 °C for 15 min. Whole protein concentration was determined using the Roti® Quant universal assay (Carl Roth #0120.1).

### Sodium dodecyl sulfate-polyacrylamide gel electrophoresis (SDS-PAGE) and immunoblotting

For SDS-PAGE, whole protein extracts were separated on a Criterion™ TGX™ 4–15% polyacrylamide gel (Bio-Rad #5671085). To differentiate between monomeric and dimeric IgA, SDS-PAGE was performed under non-reducing conditions using a 3–8% Criterion™ XT Tris-Acetate polyacrylamide gel (Bio-Rad #3450131) and XT Tricine Running Buffer (Bio-Rad #1610790). The amount of IgG secreted into the SN by ex vivo PCs was quantified by SDS-PAGE under non-reducing conditions on a 4–15% precast polyacrylamide gel (Bio-Rad #5671085). Separated proteins were blotted to a PVDF membrane by semi-dry transfer.

After blocking in 5% (w/v) non-fat milk in Tween20-tris-buffered saline (T-TBS), membranes were probed with specific primary antibodies, followed by respective horseradish peroxidase (HRP)-conjugated secondary antibodies (Supplementary Table 7). Specific proteins were detected using chemiluminescent HRP substrate on the ChemiDoc™ XRS+ (Bio-Rad) or the iBright 1500 imaging system (Thermo Fisher Scientific) and quantified in ImageJ. To ensure similar transfer and equal loading of proteins, the expression of target proteins was normalized to α-Tubulin or β-Actin. The amount of secreted IgG by ex vivo-generated PCs was normalized to the RNA concentration, a measure of viable cell mass, isolated from respective cells.

### IgA-specific enzyme-linked immunosorbent assay (ELISA)

For total IgA quantification, fecal protein samples or SNs from ex vivo-generated PCs were coated onto Nunc™ MaxiSorp 96-well plates (Thermo Fisher Scientific #442404) at 4 °C overnight, followed by three washing steps and blocking. A serial dilution of purified human IgA kappa (SouthernBiotech #0155K-01) served as a standard curve for quantification.

For the detection of dimeric IgA secreted by ex vivo differentiated PCs, we established a dIgA-specific ELISA, which was validated to exclusively detect binding of dIgA to the polymeric immunoglobulin receptor (pIgR) using human monomeric IgA from serum (Sigma-Aldrich #I4036), secretory IgA from colostrum (Sigma-Aldrich #I2636), IgG1 from myeloma (Sigma-Aldrich #I5154) and recombinant anti-CD38 mIgA2 as negative controls (Supplementary Fig. 7). For coating, 150 ng/well of pIgR (R&D Systems #2717-PG) was incubated on a Half-Area High-Bind microplate (Corning #3690) at 4 °C overnight. Following three washing steps and blocking, the supernatant was incubated for 1.5 h at room temperature.

After an additional three washes, wells were incubated with a human IgA-specific antibody conjugated to HRP (Supplementary Table 7). After six washing steps, 3,3',5,5'-tetramethylbenzidine (TMB) substrate was added, and the enzymatic reaction was stopped using 3 M HCl. Optical density (OD) was measured at 450 nm on the SpectraMax® iD3 microplate reader (Molecular Devices) and normalized to a reference wavelength of 540 nm. OD detected in control samples (only supplemented IMDM from Day 10–14) was subtracted from signals of ex vivo-generated PC SNs for background correction. Quantified amount of secreted IgA was normalized to the RNA concentration, a measure for viable cell mass, of respective ex vivo generated PCs isolated on Day 14.

### Immunohistochemical (IHC) and immunofluorescence (IF) tissue staining

Immunohistochemical staining of 2 μm tissue sections of formalin-fixed and paraffin-embedded (FFPE) colon samples was performed according to standard protocols. After deparaffinization, rehydration, antigen retrieval, endogenous peroxidase inhibition, and blocking, the tissue was probed with specific primary antibodies against IgA or pIgR, followed by respective HRP-labeled secondary antibodies (Supplementary Table 7). Subsequently, tissue sections were incubated with 3,3'-diaminobenzidine (DAB) substrate solution, counterstained with Mayer's hemalum solution, dehydrated, and mounted in Entellan™. In the case of immunofluorescence, tissue was permeabilized using 0.1% Triton X-100 and incubated with primary antibodies against human IgA and P32, followed by respective fluorochrome-labeled IgG secondary antibodies (Supplementary Table 7). Afterwards, nuclei were counterstained with 4,6'-diamidino-2-phenylindole (DAPI).

Images were obtained on an Axio Scope.A1 microscope (Zeiss) using the ZEN lite imaging software (v2.3; Zeiss). The area fraction of the epithelium or the lamina propria stained for human pIgR or IgA, respectively, was quantified by applying the Color Deconvolution 2 plugin in ImageJ (v1.53t; National Institutes of Health). To account for differences in cell density, IgA staining was normalized by hematoxylin nuclear staining. For murine colonic tissue, the percentage of IgA$^+$ cells was determined by normalization to the total number of nuclei.

### Spatial single-cell proteomics (SSCP)

Spatial single-cell proteomics of 5 μm FFPE colonic biopsy sections was performed on the CosMx™ Spatial Molecular Imager (SMI, NanoString/Bruker) using the 64-plex CosMx™ Human Immuno-Oncology Protein Panel (NanoString/Bruker #121500010) according to the manufacturer's protocol. Two independent experiments were performed, each including $n = 3$ CD patients in remission and $n = 3$ non-IBD controls. Cell segmentation was performed based on the markers

B2M/CD298, PanCK, CD45, and CD3, as well as nuclear staining with DAPI.

An expression matrix with raw fluorescence intensities was exported from the AtoMx™ platform. The positivity cut-off was set to 30 or 100, using mouse IgG1 and rabbit IgG isotype controls, respectively. All fields of view (FOVs, $0.5 \times 0.5$ mm) containing (part of) a lymphoid follicle/aggregate were excluded from the data analysis of total immune cells. To account for differences in area and cell density of the LP, immune cell counts were normalized to the number of total Vimentin$^+$ LP cells. Following isotype fluorescence subtraction from each target per cell (negative values set to zero), Uniform Manifold Approximation and Projection (UMAP) of total Vim$^+$ LP cells or CD138$^+$Vim$^+$ PCs was performed using the Qlucore Omics Explorer (v3.9; Qlucore, Lund, Sweden). Thirteen targets unrelated to the B-cell compartment (β-catenin, CD16, CD3, CD4, CD56, CD68, CD8, EGFR, EpCAM, fibronectin, FOXP3, Her2, and SMA) were excluded from the phenotyping of PCs (UMAPs and dotplot). This approach allowed for the analysis of a median of 16,444 (IQR: 11,947–30,849) cells per patient, of which 1,943 (IQR: 920–2,722) were PCs. The median distance of each PC subtype to the nearest EpCAM$^+$ epithelial cell was calculated using R (v4.2.3).

## Plasma proteomics using liquid chromatography-coupled mass spectrometry (LC-MS)

Untargeted label-free proteome profiling was performed by liquid chromatography-coupled mass spectrometry (LC-MS). Plasma proteins were precipitated by incubation in ice-cold acetone (100%) followed by centrifugation. The protein pellet was dried and reconstituted in DOC buffer (1% (w/v) sodium deoxycholate (DOC) and 100 mM ammonium bicarbonate) to determine the protein concentration according to the Roti® Quant universal assay. Fifty micrograms of protein were incubated with dithiothreitol (1 M) and iodoacetamide (0.5 M). Proteins were digested using trypsin (Promega #V5280) overnight. After treatment with formic acid and centrifugation, peptides were lyophilized, and pellets were reconstituted in 0.1% (v/v) formic acid in water for mass spectrometric analysis.

Extracted peptides were loaded onto C18 EvoTip disposable trap columns and separated by chromatography on the EvoSep One (Evo-Sep) using a C18 Performance column (EV1137, 15 cm × 150 μm, 1.5 μm) paired with the extended 15 SPD method (88 min gradient, 220 nl/min). Peptides were analyzed on the TIMS (trapped ion mobility spectroscopy) quadrupole TOF (time-of-flight) mass spectrometer (timsTOF Flex HT, Bruker), including a CaptiveSpray nano-electrospray ion source and operating in the diaPASEF mode[52]. The acquired diaPASEF mass spectra were processed using the DIA-NN software tool (v1.8.1)[53], employing an optimized window design by using py_diAID[54].

A unique protein matrix containing only proteins that were identified and quantified by at least two proteotypic peptides and that passed the FDR cut-off of 0.01 was created using the DIA-NN package in R. Protein abundances resulting from the MaxLFQ algorithm[55] were preprocessed using the R package DEP[56]. Following the exclusion of proteins identified in less than 70% of the samples, a variance stabilizing normalization[57], including a log$_2$ transformation, was applied to the data. Missing values were imputed by employing the k-nearest neighbor method, and the observed batch effect (two batches) was corrected using the ComBat function[58] of the Surrogate Variable Analysis (sva) R package[59].

## Analysis of plasma metabolites using $^1$H NMR spectroscopy

Plasma metabolomics was performed using proton nuclear magnetic resonance ($^1$H NMR) spectroscopy, applying Bruker's standardized in vitro Diagnostic Research (IVDr) procedure (Bruker BioSpin). Plasma samples were mixed 1:1 (v/v) with buffer consisting of 75 mM sodium phosphate (pH 7.4), 0.04% (w/v) sodium azide (NaN$_3$), 4.6 mM 3-trimethyl-silyl-[2,2,3,3-H$_4$] propionic acid (TSP-d$_4$), and 20% (v/v) deuterium oxide (D$_2$O) by manual panning for one minute and 600 μl of sample were transferred to a 5 mm NMR tube. NMR spectrometry was carried out on a Bruker Avance III HD 600 MHz NMR spectrometer with a TXI probe at 310 K. The spectrometer was equipped with a SampleJet™ (Bruker) for automation, in which samples were kept at 279 K until measurement. Prior to analysis, a standard operating procedure was performed to check temperature calibration, quantification, and water suppression performance. A one-dimensional (1D) nuclear Overhauser effect spectroscopy (NOESY) experiment (pulse program: noesygppr1d) and a 1D Carr−Purcell−Meiboom−Gill (CPMG) spin-echo experiment (pulse program: cpmgpr1d) for the suppression of protein and other macromolecular signals were recorded per sample. Quantification in plasma/serum (B.I.QUANT-PS™) and lipoprotein subclass analysis (B.I.LISA™) of the Bruker IVDr platform were used to automatically quantify 39 metabolites (+2 technical additives) and 112 lipoprotein parameters, including several subfractions of apolipoproteins (Apo), cholesterol (Chol), free cholesterol (FC), phospholipids (PL), and triglycerides (TG). Concentrations below the limit of detection were assigned a value of zero. For principal component analysis (PCA), metabolites identified in fewer than 40% of the samples were excluded, and data were transformed using the Yeo−Johnson method.

## Isolation of primary murine cells and culture of splenic B cells

Murine MLNs, PPs, and splenic tissue were processed through a 70-μm cell strainer to obtain single-cell suspensions.

Splenic resting B cells were enriched using negative selection MACS (a cocktail of biotin-conjugated monoclonal antibodies against CD43, CD4, and Ter-119) following the manufacturer's instructions of the B Cell Isolation Kit (Miltenyi #130-090-862). Subsequently, the isolated splenic B cells were cultivated in RPMI medium with 10% (v/v) FBS, 100 μM 2-mercaptoethanol, 1 mM HEPES, 100 U/ml penicillin, and 100 μg/ml streptomycin in the presence or absence of 3 μg/ml LPS-EB ultrapure (InvivoGen #tlrl-3pelps), and 10 ng/ml IL4 (Thermo Fisher Scientific #214-14) for 4 days. After cultivation, live cells were counted in the Attune NxT flow cytometer (Thermo Fisher Scientific) using trypan blue staining and prepared for flow cytometry as described below.

## Flow cytometry

For mitochondrial staining, cultivated splenic B cells were incubated with 150 nM tetramethylrhodamine ethyl ester (TMRE, Thermo Fisher Scientific #T669) and 100 nM MitoTracker™ Green (Thermo Fisher Scientific #M7514) at 37 °C for 15 min.

Following Fc receptor blocking with an anti-CD16/32 antibody (Supplementary Table 7) at 4 °C for 10 min, splenic B cells, as well as cells from MLNs and PPs, were stained with fluorochrome-labeled primary antibodies against B220, CD138, or IgG1 (Supplementary Table 7) on ice for 30 min. Subsequently, cells were washed, resuspended in PBS containing 0.5% (w/v) BSA, and measured immediately using the Attune NxT flow cytometer. Data analysis was performed using the FlowJo software (v10.10; FlowJo, Ashland, OR) and cells were selected using a lymphocyte gate, with debris and doublets excluded. Gating strategies for splenic B cells and cells isolated from MLNs/PPs are specified in Supplementary Fig. 5a and e, respectively.

## Statistical analysis

Statistical analyses were performed using GraphPad Prism (v10.0.2; GraphPad Software, San Diego, CA). No statistical method was used to predetermine sample sizes. Outliers were identified and removed by the ROUT test (Q = 1%, for stool IgA ELISA Q = 2%). Normality was assessed using either the D'Agostino−Pearson or the Kolmogorov−Smirnov test, followed by an unpaired two-tailed t-test or a two-tailed Mann−Whitney U-test for the comparison of two groups. Statistical differences between more than two groups were analyzed by multiple Mann−Whitney U-test or Kruskal−Wallis test with Dunn's

post hoc test. Two-way analysis of variance (ANOVA) with Bonferroni's, Dunnett's, Šídák's, Tukey's, or Fisher's least significant difference (LSD) post hoc test was applied for datasets with two variables. The correlation between variables was assessed by determining the Spearman correlation coefficient. $P$ values were calculated, and null hypotheses were rejected when $P \leq 0.05$. Data were shown as median with range. For data visualization, GraphPad Prism, Qlucore Omics Explorer (v3.9; UMAPs), and R (v4.2.3) in RStudio (v2024.12.0 + 467; metabolomics PCA and enrichment plot) were employed. All schemes were created using BioRender (Toronto, Canada).

### Reporting summary

Further information on research design is available in the Nature Portfolio Reporting Summary linked to this article.

## Data availability

The mass spectrometry proteomics data generated in this study have been deposited at the ProteomeXchange Consortium via the PRIDE[60] partner repository with the dataset identifier PXD069842. The bulk RNA-seq data were available at the European Nucleotide Archive (ENA) under the accession number PRJEB98848. Spatial single-cell proteomics data generated using the CosMx™ platform are accessible on figshare [https://doi.org/10.6084/m9.figshare.31057111]. The processed $^1$H NMR metabolomics data are provided in the Source Data file. Source data are provided with this paper.

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

## Acknowledgements

This work was supported by the Federal Ministry of Education and Research (research grant 01KD2103A in the course of the OUTLIVE-CRC consortium) and internal funding from the University of Lübeck (research grant J04-2023 to L.N.d.A.). The BioRender license used for creating figures was funded by the Research Training Group (RTG) 2633 of the University of Lübeck. The authors thank all the participating patients for contributing to this study and Dr. Misa Hirose and Prof. Saleh Ibrahim for generously providing the conplastic C57BL/6J[FVB/NJ] mouse line. A.K. and H.B. acknowledge computational support from the OMICS compute cluster at the University of Lübeck.

## Author contributions

S.D. designed the concept of the study and supervised it. P.S., M.M.K., J.U.M., A.R., and H.S. collected and provided human biopsy and blood samples. M.W., D.B., H.C.T., L.C., and the OUTLIVE-CRC consortium collected and provided human stool samples. A.R., L.N.d.A., F.S., T.G., T.S., M.R., H.S., and M.H. performed the experiments and acquired the data. A.K. and H.B. supported bioinformatic data analysis. M.P. and T.V. produced and provided the recombinant human anti-CD38 IgA2 antibody. R.A.M., L.A., M.E., P.R., U.L.G., and C.S. provided expertise in the field and offered critical feedback on the discussion of the study results. A.R. and S.D. analyzed and interpreted the data and drafted the article. All authors read and approved the final manuscript.

## Funding

## Competing interests

The authors declare no competing interests.

## Additional information

[1]Institute of Nutritional Medicine, University Hospital Schleswig-Holstein, Campus Lübeck, Lübeck, Germany. [2]1st Department of Medicine, University Hospital Schleswig-Holstein, Campus Lübeck, Lübeck, Germany. [3]Institute of Chemistry and Metabolomics, University of Lübeck, Lübeck, Germany. [4]Department of Gastroenterology, Hepatology, Infectious Diseases and Endocrinology, Hannover Medical School, Hannover, Germany. [5]Gastroenterology, Hepatology, Endocrinology, Rheumatology and Infectious Diseases, Department of Internal Medicine I, University Hospital Regensburg, Regensburg, Germany. [6]CED Schwerpunktpraxis, Münster, Germany. [7]Medical Faculty, University of Münster, Münster, Germany. [8]Division of Antibody-Based Immunotherapy, Department of Internal Medicine II, University Hospital Schleswig-Holstein and Christian-Albrechts-University Kiel, Kiel, Germany. [9]Division of Stem Cell Transplantation and Immunotherapy, Department of Medicine II, University Hospital Schleswig-Holstein and Christian-Albrechts-University Kiel, Kiel, Germany. [10]Department of Internal Medicine II, Rechts der Isar Hospital, TUM School of Medicine and Health, Technical University of Munich, Munich, Germany. [11]Section for Translational Surgical Oncology and Biobanking, Department of Surgery, University Hospital Schleswig-Holstein, Campus Lübeck, Lübeck, Germany. [12]Airway Research Center North, University of Lübeck, German Center for Lung Research, Lübeck, Germany. [13]Institute of Clinical Molecular Biology, University of Kiel and University Hospital Schleswig-Holstein, Kiel, Germany. [14]Institute for Systemic Inflammation Research, University of Lübeck, Lübeck, Germany. [15]Medical Systems Biology Group, Lübeck Institute of Experimental Dermatology, University of Lübeck, Lübeck, Germany. [16]University Cancer Center Schleswig-Holstein, University Hospital Schleswig-Holstein, Campus Lübeck, Lübeck, Germany.
✉e-mail: Annika.Raschdorf@uksh.de; Stefanie.Derer@uksh.de

## OUTLIVE-CRC consortium

**Alexander Katalinic[17], Martina Oberländer[18], Ruth Deck[17], Stefanie Derer ⬥ [1]✉, Holger Sültmann[19,20], Nikolas von Bubnoff[21], Timo Gemoll[11], Hauke Busch[15,16], Tobias Hutzenlaub[22,23], Peter Jülg[22,23], Christian Sina[1,2] & Dominik Burziwoda[24]**

[17]Institute for Social Medicine and Epidemiology, University of Lübeck, Lübeck, Germany. [18]Interdisciplinary Center for Biobanking-Lübeck (ICB-L), University of Lübeck, Lübeck, Germany. [19]Division of Cancer Genome Research, German Cancer Research Center (DKFZ), German Cancer Consortium (DKTK), National Center for Tumor Diseases (NCT), Heidelberg, Germany. [20]German Center for Lung Research (DZL), TLRC Heidelberg, Heidelberg, Germany. [21]Department of Hematology and Oncology, University Hospital and University Cancer Center Schleswig-Holstein, Campus Lübeck, Lübeck, Germany. [22]Hahn-Schickard Society, Freiburg, Germany. [23]Department of Microsystems Engineering (IMTEK), University of Freiburg, Freiburg, Germany. [24]Perfood Laboratories GmbH, Lübeck, Germany.

