## [Transparent Peer Review file · Nature Communications]

Colonic spatial single-cell proteomics and murine models link mitochondrial dysfunction to dimeric IgA-secreting plasma cell deficiency in Crohn's disease

Corresponding Author: Ms Annika Raschdorf

Version 0:

Reviewer comments:

Reviewer #1

(Remarks to the Author)

The article by Raschdorf and colleagues examines the phenotype and functionality of plasma cells and plasmablasts in the colon of Chron's disease patients in remission. The authors find that CDrem patients had a higher proportion of B cells and plasmablasts in the colonic mucosa, but an overall reduction in secretory IgA compared to healthy controls. They hypothesise that the lack of IgA may be due to incomplete or exacerbated differentiation of B cells/PBs into fully mature plasma cells (PCs), which they confirm by ex vivo differentiation. Moreover, they propose a mechanisms by which the B cell lineage in CD shows a reduced capacity for oxidative phosphorylation, correlating to increased CD38 expression, leading to a reduction in mature PC development. The authors further link OxPhos and PC development by showing that Atp8 mutant mice, which have impaired respiratory capacity, demonstrate a lack of intestinal PC development.

Overall, the manuscript is clear and well-presented and contains exhaustive and interesting data from human samples that suggest a molecular mechanism which may explain changes in the B cell lineage in the colonic mucosa. The analysis of human material is very thorough and I only have some minor concerns about the experimental methods and some of the interpretation of the data. The last section of the paper, using a mouse model, is very interesting but here I have some more pressing concerns regarding the reproducibility of the data and some missing readouts which would be essential in assessing the importance and relevance of the mouse experiments. However, if the authors address these concerns I would be happy to support the publication of this manuscript in Nature Communications.

- The authors note an increase in CD19+ CD138- B cell in the colonic mucosa of CDrem patients, but the localisation of these cells is not clear. Are the B cells the authors observe in colonic biopsies localised in the lamina propria or in lymphoid aggregates (lymphoid follicles, inflammation-induced tertiary lymphoid tissue etc.)? Is the increased proportion of B cells in the colon of CDrem due to a higher number of lymphoid follicles and/or lymphoid tissues? This is especially important for the interpretation of the data since an increased number of follicles would disproportionately affect B cells counts.

- The authors demonstrate an increased proportion of B cells and plasmablasts in the colons of CDrem patients but no overall change in the proportion of plasma cells. However, it is unclear if there is an overall increase in infiltrate in the colons of these patients, i.e. whether there are more leukocytes overall. If yes, this would actually indicate an increase in the overall number of PCs, albeit disproportionately when compared to PBs. Could the authors provide any indication whether the numbers of leukocytes is increased in CD patients (e.g. quantified per unit area)?

- When discussing figure 3E, the authors state "As expected, CD138 expression was similar across the 2 populations, while the proportion of Ki-67+ cells was reduced in CD19-CD45+ and CD19-CD45- PCs compared to CD19+CD45+ PCs". However the data in 3E show no significant difference between the latter two subpopulations. Please amend the text accordingly. Moreover, it would be helpful to show whether the Ki67 pattern seen in CD patients is also present in healthy colonic PC subpopulations.

- In 3I the authors quantify dIgA/BCMA ratios by western blot and show a reduced ratio in CD compared to healthy controls and state that "... the significantly reduced dIgA/BCMA ratio indicated a decreased secretion of dIgA per PB/PC in the colonic mucosa of CDrem" However, it is unclear if the reduced ratio is primarily due to more BCMA or less dIgA? The authors already show an increase in the number of colonic B cells, which would presumably also increase the BCMA signal and lead to a skewed ratio (Less dIgA per B/PB/PC would then be tautological). A more informative quantification might be a ratio of dIgA/CD138 to get a measure of dIgA production per PC instead.

- Related to this, could a reduction in dIgA production be due to an increase in monomeric IgA production? Can the method

used accurately quantify mIgA?

- The data in figure 4 is very informative and convincingly shows an intrinsic defect in CD B cell differentiation into IgA-secreting PCs. Could the authors specify whether the ELISA used for quantification of IgA secretion in 4G detects only dIgA or whether it would also measure mIgA? If not, could the western blot quantification (from 4C) be utilised instead? I think this could be an important point in understanding the defect of CD PCs, as a reduction of J chain production could indicate more mIgA secretion which would also account for total reduced SIgA. Similarly, does the ELISA used discriminate between IgA1 and IgA2, as the latter seems to be disproportionately affected in CD?
- Could the authors provide a more detailed explanation of what the images in Figure 6A are supposed to illustrate?
- In figure 7, the overall number of mice used seems very low with only two mice per group in 7E and three mice in 7F and J. The authors should increase the number of biological replicates and perform all experiments on at least two occasions to demonstrate reproducibility. In addition, could the authors indicate why some experiments were performed only on male mice (qPCR) and some only on female mice (GFHP diet).
- The authors use in vitro differentiation of splenic B cells to show a defect in the capacity of Atp8 mutant cells to develop into IgG-producing PCs, however the relevance of these data is unclear. Since these experiments are supposed to provide a mechanistic explanation for the human data, which shows a selective defect in IgA-PC induction, the authors should utilise conditions that favour induction of IgA producing PCs (i.e. IL-6, IL-21, TGF β) for their in vitro assay.
- Similarly, the relevance of Figure 7F is uncertain. Since the main phenotype observed in human concerns colonic PC in the lamina propria, it is unclear why the authors chose to analyse lymphoid tissues associated with the small intestine (MLN and PP). This analysis should be complemented by FACS (or multi-coloured immunohistochemistry) of PCs in the colonic LP. This has the added benefit that it would allow for a much more detailed phenotypic description of these cells than the rather uninformative "IgA+ cells" obtained from the histology in 7J and 7I.

Reviewer #2

(Remarks to the Author)

In this manuscript Raschdorf and colleagues study the B cell compartment in Crohn's disease patients in remission. Several recent studies evaluated the composition and activity of B cells in inflammatory bowel disease. As stated by the authors, the main difference with the current study is the fact that in the earlier studies, only patients with active disease were included, or disease activity was not addressed specifically. I agree with the authors that disease in remission is an interesting subject in its own right, but I do have several concerns regarding this study.

Major general concerns:

- The authors emphasize the fact that they want to study patients in remission. However, when assessing their patient characteristics, a considerable proportion of the 'remission' patients had macroscopic inflammation in the ileum at the time of sampling. Although I realize the samples were obtained from the colon, this does stress these patients were in fact not in remission. For any systemic measurements (stool/serum) this is a problem in itself, and for the specific colonic analyses, the study is actually not different from the 'non-inflamed' samples which have in fact been included in earlier studies (e.g doi.org/10.1038/s41591-022-01680-y). As this concerns up to 33% of patients in many analysis, this is the difference between showing activity in actual remission, or just repeating earlier studies.

- On an analytical level, this study lacks rigor. The biggest problem is an unacceptable level of cherry picking in the statistics. In several panels, effects which they deem relevant are described as 'increased' or 'decreased' without acknowledgement of the fact that the difference lacks statistical significance. At the same time, effects apparently deemed unwanted are described as 'not different' and mentioned without any p-values, just as 'NS'. An example is figure 2c, where the change in IgM is described as 'increased' (p value 0.1), while in figure 2g the proportion of PC is 'not upregulated', even though there is a clear trend as well, and no p value is provided. This is just one example, but occurs throughout the manuscript. The authors need to be consistent in their description (it is fine to describe non-significant effects as well, but then for all) and provide p values for all comparisons.

Other general concerns:

1) the use of low sample numbers, which for sure has underpowered several analyses. Already in the first paragraph, the authors mention there was no clustering of samples in the PCA plot based on medication. However, most medications only included a single patient, and therefore can not possibly cluster. RNASeq analysis with primary material using 12 vs 7 samples is very limited etc. This is also evident from the volcano plot (fig 1b) which depicts p values rather than p values adjusted for high level multiple testing (Bonferroni or False Discovery Rate) as should be done for RNASeq. Using the appropriate metric would show no significant DEG.

2) Use of less than conventional methods. Determination of immune cell subsets is usually performed by surface marker evaluation rather than transcription, in particular bulk expression. For example in figure 2g/h, immune cells were actually isolated from the colon, but rather than perform flow cytometry, were assessed by bulk expression. Production and secretion of antibodies is consistently measured by Western Blot, while this is mostly done by more specific ELISA assay (which can be done on the sample sample type). This makes quantification much more reliable, as the western blots are not all convincing (eg Suppl figure 2g really does not show reliable bands, and thus quantifications is not very meaningful).

3) The role of metabolic changes seems to be an overreach. Metabolomics are performed on plasma, and can therefore not be correlated to local B cells. Bulk RNASeq similarly does not indicate the involvement of B cells. The mechanism is mainly suggested based on RNA transcription (at best suggestive, not definitive for metabolic activity) and correlations. If they authors really want to state that decreased OXPHOS is involved in the impaired differentiation of PC in IBD, they need to

perform the actual experiment, which would be Seahorse assays on the B cells themselves.

4) I don't understand the use of Vimentin as a positive marker in this study. I understand the need to correct in some way for the amount of lamina propria included in the respective samples. However, this is commonly done using DAPI or another nuclear/DNA stain. Correcting against total vimentin as a LP marker is unusual but I can understand to some degree. However, the authors refer to the plasmablasts/cells (as well as T cells) as Vim+, which seems odd, as this is not a marker for these cells at all?

Other specific concerns

- Several graphs contain delta-Ct values as a measure of absolute expression. Delta-CT is a perfectly fine method for assessing relative expression of a particular gene, but can not be compared between different primers, and thus different genes. These values should not be in a single graph (eg Suppl fig 2e), and definitely not in a stacked bar such as figure 4e. The statement 'cells mainly expressed CD20, CD27 etc' is incorrect from this figure (I do believe they do, but the conclusion can not be drawn from this figure).
- Several figures are presented in a way which does not allow any statistical analyses, even though the data itself would. For example figure 2h shows a compound's alteration in expression, with no information regarding distribution and thus no insight into the robustness of the data. Fig 3j is used to suggest increased expression of IL6, which is highly doubtful based on the image (seems mainly driven by 1 donor).
- Since identification of cells in the spatial analysis is crucial, some images of the separation and analysis should be provided as supplementary information. Showing the staining of a single double/triple stained cell does not prove anything else than the fact one exists. For example, the number of T cells (or at least CD3+ cells) appears extremely low, and should be addressed.

Reviewer #3

(Remarks to the Author)

The manuscript by Raschdorf et al. provides a deep dive into the role of IgA+ plasma cells and blasts in Crohn's disease. Unconventionally, they target patients in remission (i.e., without active inflammation). They apply bulk RNA sequencing, fluorescent microscopy, CosMX, mass-spec, ex vivo profiling and in vitro mouse modelling to propose a cell-intrinsic defect in the terminal differentiation of CD19-CD45-CD138+ long-lived plasma cells. Increased CD38 expression and the employment of ketogenesis and glycolytic activity suggest OXPHOS deficiency in these cells. Together, the manuscript sheds light on altered IgA plasma cell profiles driving dysregulated barrier control in CD, rather than being simply in response to inflammation. The manuscript is well written and logical. The numbers of participants included in the study are not large (i.e., 7 vs 12, 6 vs 5 and 6 vs 6), but the chosen methods are appropriate and executed effectively. I have only minor comments.

1. To further support defective regulation of the microbiota in CD rem, the authors could consider B cell receptor repertoire analysis or analysis of IgA binding to faecal microbiota (e.g., by flow cytometry).
2. Methods page 17 line 18: Patients were classified as being in remission if the colon presented without macroscopic evidence of mucosal inflammation. Was this confirmed histologically?
3. Methods page 19 line 19: The 'different combinations of cytokines' used in the differentiation of PBs and PCs should be explicitly stated.
4. Please include catalogue numbers for kits included throughout the methods section.

Version 1:

Reviewer comments:

Reviewer #1

(Remarks to the Author)

The authors have made significant changes to the manuscript and have addressed or provided reasonable explanations for most of the concerns I raised in the initial review. Indeed, the authors clearly went to great lengths to seriously engage with the reviewers' comments, which should be commended. The development of a dIgA ELISA in such a short timeframe is especially impressive.

The one potential difficulty still remaining is the extremely limited number of biological replicates in the mouse experiments, which the authors were able to only partially address. My understanding is that this is due to unfortunate and unpredictable circumstances and certainly not due to any lack of willingness by the authors to engage with the problem. At this point, the possible solutions would be to either remove the mouse data or to proceed with the small number of replicates to publication. On the whole, I very much favour the latter option, providing that the limitation inherent in the small sample sizes is clearly indicated in the text. The mouse data is not the primary focus of the manuscript but is still informative and provides a potential mechanistic explanation for the observed phenotype that is novel and interesting.

Ultimately, the final decision lies with the editors and adherence to the editorial policy of Nature Communications. However, from the point of view of scientific peer review I am happy to endorse the publication of the manuscript.

Reviewer #2

(Remarks to the Author)

The authors have resolved most of my concerns in their rather lengthy rebuttal. Some concerns remain, which can probably be handled textually.

Concern 1

- The use of samples including those with inflammation in the small bowel

The authors extensively describe their samples in their response, which largely satisfies my concerns regarding their patient selection. The only thing I would like to confirm before assigning these patients as 'in remission' is whether these patients previously showed colonic disease. I agree that someone with previous colonic involvement and currently only aftous lesions on the ileocecal anastomosis is in remission. However, to differentiate their study from earlier work, it is crucial these patients did in fact have colonic involvement at some point in their disease history. I am willing to assume that is the case, but it would be important to confirm this in the patient characteristics.

Concern 2

- The use of low sample numbers, which for sure has underpowered several analyses

I appreciate the fact that the study was focused on techniques not suitable for high throughput analysis (ie spatial transcriptomics) and would not require to increase sample size for these experiments. However, I would be very careful in stating anything regarding the RNASeq, which clearly is underpowered. Representation of the data is now improved, and I can agree to that. However, still any mention regarding the effect of medication here should be removed to the small group sizes.

The additional analysis of IgA based on medication does unfortunately not resolve matters. These data show no significant differences, but include many different treatments grouped together or still in too small numbers to provide meaningful information. Although not significant, for example anti-TNF vs other biologicals does appear to show a trend to higher IgA in the 'other' group. This may very well be explained by differences in initial disease activity (patients receiving other biologicals almost always have failed anti-TNF and thus suffer more refractory disease), but still confounds the data. Additionally, no patient in this sub-cohort appears to be using no biologic at all, making this an unusual patient cohort. Since the authors do not focus on the effect of medication at all, I would remove most comments regarding this data, and at best state that although numbers were too small for statistical analysis, they did not find data suggesting an effect of the medication (if they want to mention medication at all cost).

Concern 3

- The data regarding metabolomics was mainly associative and lacked functional assays

I appreciate the initial focus of the study did not allow for the inclusion of for example Seahorse experiments and the added metabolomic analysis did add support for the claims of the authors. I would appreciate some statements that this requires further study etc in the discussion.

Reviewer #3

(Remarks to the Author)

The authors have satisfactorily addressed each of my comments to their original manuscript and I have no additional comments. The revised manuscript has been improved through the revision process.

Version 2:

Reviewer comments:

Reviewer #2

(Remarks to the Author)

The authors have now satisfied my remaining concerns.

The article by Raschdorf and colleagues examines the phenotype and functionality of plasma cells and plasmablasts in the colon of Crohn's disease patients in remission. The authors find that CDrem patients had a higher proportion of B cells and plasmablasts in the colonic mucosa, but an overall reduction in secretory IgA compared to healthy controls. They hypothesise that the lack of IgA may be due to incomplete or exacerbated differentiation of B cells/PBs into fully mature plasma cells (PCs), which they confirm by *ex vivo* differentiation. Moreover, they propose a mechanism by which the B cell lineage in CD shows a reduced capacity for oxidative phosphorylation, correlating to increased CD38 expression, leading to a reduction in mature PC development. The authors further link OxPhos and PC development by showing that *Atp8*-mutant mice, which have impaired respiratory capacity, demonstrate a lack of intestinal PC development.

Overall, the manuscript is clear and well-presented and contains exhaustive and interesting data from human samples that suggest a molecular mechanism which may explain changes in the B cell lineage in the colonic mucosa. The analysis of human material is very thorough and I only have some minor concerns about the experimental methods and some of the interpretation of the data. The last section of the paper, using a mouse model, is very interesting but here I have some more pressing concerns regarding the reproducibility of the data and some missing readouts which would be essential in assessing the importance and relevance of the mouse experiments. However, if the authors address these concerns I would be happy to support the publication of this manuscript in Nature Communications.

(1) The authors note an increase in CD19+ CD138- B cells in the colonic mucosa of CD^{rem} patients, but the localisation of these cells is not clear. Are the B cells the authors observe in colonic biopsies localised in the lamina propria or in lymphoid aggregates (lymphoid follicles, inflammation-induced tertiary lymphoid tissue etc.)? Is the increased proportion of B cells in the colon of CD^{rem} due to a higher number of lymphoid follicles and/or lymphoid tissues? This is especially important for the interpretation of the data since an increased number of follicles would disproportionately affect B-cell counts.

- The authors thank the reviewer for raising this comment and fully agree that the localization of, for example, B cells is very important for the interpretation of the data.

The main focus of our study was on investigating PC differentiation in the colonic mucosa of CD^{rem} patients, for instance using spatial single-cell proteomics on FFPE tissue slices (CosMxTM SMI). Since CD138⁺ PCs are not located in lymphoid follicles/aggregates but mainly in the LP (Figure 1), no regions were excluded from the PC-specific analyses in Figure 3c-h and Figure 5a-d of the original manuscript.

Fig. 1 | Representative image of a lymphoid aggregate in the colonic biopsy section of a CD^{rem} patient showing that CD138⁺ PCs (green) are almost absent in the lymphoid aggregate and are exclusively located in the LP.

In contrast, we observed that especially the percentages of B cells (CD19, CD20) and T cells (CD3, CD4, CD8) varied greatly between patients, depending on the presence of a lymphoid follicle/aggregate in the analyzed biopsy section. Further, the number of lymphoid follicles/aggregates shows interindividual variability. In our samples, we identified a lymphoid follicle/aggregate in 2 out of 6 non-IBD tissue sections and 4 out of 6 CD^{rem} tissue sections. Therefore, we excluded all fields of view (FOVs) containing (part of) a lymphoid follicle/aggregate (Figure 2) from the analysis of total immune cell percentages, depicted in Figure 1e of the original manuscript. To highlight the importance of excluding lymphoid follicles/aggregates for the interpretation of the data, the authors illustrated the high density of B and T lymphocytes within these structures in Figure 2. We can therefore conclude that the increased proportion of CD19⁺CD138⁺ B cells is present in the lamina propria (LP), independent of the number of lymphoid follicles/aggregates. These analyses validated the expanded B-cell compartment previously observed in bulk RNA-seq of colonic tissue and qPCR of isolated colonic LPMNCs, which did not allow for spatial differentiation. In follow-up studies involving a larger patient cohort, we aim to investigate intestinal lymphoid follicles/aggregates in CD patients in more detail.

In the first version of the manuscript, the exclusion of lymphoid follicles/aggregates from the analysis of total immune cells was only described in the methods section on page 22, lines 28-29 and page 23, line 1. Additionally, we edited a sentence in the results section of the revised manuscript file on page 7, line 21 (highlighted in yellow) to clarify that the analysis focuses on immune cell frequencies in the LP.

Fig. 2 | Representative image of a lymphoid aggregate in the colonic biopsy section of a CD^{rem} patient with strong B (CD19, CD20) and T cell signals (CD3, CD4, CD8) that was excluded from analysis of immune cell proportions.

(2) The authors demonstrate an increased proportion of B cells and plasmablasts in the colons of CD^{rem} patients but no overall change in the proportion of plasma cells. However, it is unclear if there is an overall increase in infiltrate in the colons of these patients, i.e. whether there are more leukocytes overall. If yes, this would actually indicate an increase in the overall number of PCs, albeit disproportionately when compared to PBs. Could the authors provide any indication whether the numbers of leukocytes is increased in CD patients (e.g. quantified per unit area)?

- We are grateful to the reviewer for pointing out this aspect and recognize the critical importance of distinguishing between relative and absolute immune cell numbers to interpret the data properly. Therefore, it is essential to determine the density of total leukocytes/LP cells in the colonic tissue. For this purpose, we determined cell numbers of lamina propria mononuclear cells (LPMNCs) isolated from seven sequential colonic biopsies per patient, which is presented in Supplementary Figure 1b of the original manuscript. In this context, we identified a small error in the graph, where one active CD patient without colitis was inadvertently included, while another CD patient with endoscopically confirmed

active colitis had been excluded. This has now been corrected in Supplementary Figure 1c and Supplementary Table 4 of the revised manuscript. Although absolute LPMNC counts show considerable inter-patient variability (mainly due to differences in biopsy size), it is evident that the CD^{rem} patients do not exhibit an increased LPMNC count when compared to non-IBD controls. In contrast, the number of colonic LPMNCs isolated from CD patients with active colitis was significantly increased compared to CD^{rem} patients.

Moreover, the reviewer's suggestion prompted us to also consider the LP cell density within the colonic biopsy sections analyzed using spatial single-cell proteomics. Therefore, we quantified the total number of Vimentin⁺ cells, effectively excluding all epithelial cells (Figure 3a), and normalized the counts to the tissue area [mm^2]. We preferred the quantification of Vim⁺ cells over CD45⁺ cells for two main reasons. First, vimentin is also expressed by non-immune cells such as fibroblasts, which increase during inflammation. This is supported by single-cell transcriptomics data of the Human Gut Cell Atlas, published by Elmentaite et al. in Nature (2021)¹. While all LP cells, such as immune cells, endothelial cells, neuronal cells, and mesenchymal cells, express VIM, it is absent in colonic epithelial cells (Figure 3b). Second, not all immune cells express CD45. For example, mature PCs, which are the focus of our study, typically lose surface CD45 during differentiation.

Fig. 3 | Vimentin serves as a marker for colonic lamina propria (LP) cells. (a) Representative image of Vimentin staining (yellowish green) of the colonic LP cells of a non-IBD control in the CosMx™ SMI. **(b)** VIM (Vimentin) expression in colonic immune, endothelial, neuronal, mesenchymal, and epithelial cells. Single-cell transcriptomics data originate from the publicly available Human Gut Cell Atlas¹.

Omitting FOVs comprising lymphoid follicles/aggregates, there is no significant difference in the number of Vim⁺ cells/ mm^2 between CD^{rem} patients and non-IBD controls (Figure 4). This observation rules out leukocyte/cell infiltration in these patients, as it would result in a higher density of LP cells.

Fig. 4 | Absolute number of Vim⁺ cells per tissue area [mm^2] quantified in colonic biopsy sections of CD^{rem} patients (n=6) and non-IBD controls (n=6). Mann-Whitney U test.

We decided to include this data in the revised manuscript file as Supplementary Figure 1b to clarify interpretation of the results. Respective changes in the text have been highlighted in yellow and added to the results section on page 7, lines 22-24.

Referring to the reviewer's comment, we found that the absolute number of B cells per mm^2 is significantly increased in CD^{rem} patients, PBs show a strong upward trend, while the absolute number of PCs per mm^2 is not significantly different from non-IBD controls (Figure 5).

Fig. 5 | Absolute numbers of B cells (a; CD19⁺), PBs (b; CD19⁺CD27⁺CD38⁺CD20⁺CD138⁺), and PCs (c; CD138⁺) per tissue area [mm^2] in CD^{rem} patients (n=6) compared to non-IBD controls (n=6).

(3) When discussing figure 3e, the authors state “As expected, CD138 expression was similar across the 2 populations, while the proportion of Ki-67+ cells was reduced in CD19⁻CD45⁺ and CD19⁻CD45⁻ PCs compared to CD19⁺CD45⁺ PCs”. However the data in 3E show no significant difference between the latter two subpopulations. Please amend the text accordingly. Moreover, it would be helpful to show whether the Ki67 pattern seen in CD patients is also present in healthy colonic PC subpopulations.

- The authors thank the reviewer for pointing out this mistake. We have amended the sentence on page 9 lines 10-12 to clarify that only CD19⁻CD45⁺ PCs, but not CD19⁻CD45⁻ PCs, displayed a significantly decreased percentage of Ki-67⁺ cells compared to CD19⁺CD45⁺ PCs.

Since the focus of Figure 3e is on the general characterization of the three PC subtypes, the data presented include only the six non-IBD controls. The authors have now added this information to the results section on page 9, line 10 (highlighted in yellow), in addition to the legend of Figure 3e.

Moreover, the dotplot in Figure 5a of the original manuscript demonstrates that there is no significant difference in the percentage of Ki-67⁺ cells between CD^{rem} patients and non-IBD controls in any of the PC subpopulations.

To further improve the general characterization of the three PC subpopulations, we have now added a spatial analysis to the revised manuscript that allowed determination of the distance between the three PC types and the colonic epithelial cells. While no differences were observed between CD^{rem} patients and non-IBD controls, the fully mature CD19⁻CD45⁻ PCs were found in closer proximity to the epithelium compared to the CD19⁺CD45⁺ and CD19⁻CD45⁺ PCs (Figure 6). The data were included as Supplementary Figure 2f in the revised Supplementary Material file and respective changes in the text have been highlighted in yellow and added to the results section on page 9, lines 17-19 and the methods section on page 25, lines 14-16.

Fig. 6 | Distance of CD19⁺CD45⁺, CD19⁻CD45⁺, and CD19⁻CD45⁻ PC subpopulations from the colonic epithelium in CD^{rem} patients (n=6) and non-IBD patients (n=6) was determined using spatial single-cell proteomics (CosMx™ SMI). Distance is presented as relative values compared to CD19⁺CD45⁺ PCs per patient. Friedman test with Dunn's multiple comparisons test. *P<.05. AU, arbitrary units.

(4) In 3I the authors quantify dIgA/BCMA ratios by western blot and show a reduced ratio in CD compared to healthy controls and state that "... the significantly reduced dIgA/BCMA ratio indicated a decreased secretion of dIgA per PB/PC in the colonic mucosa of CDrem". However, it is unclear if the reduced ratio is primarily due to more BCMA or less dIgA? The authors already show an increase in the number of colonic B cells, which would presumably also increase the BCMA signal and lead to a skewed ratio (Less dIgA per B/PB/PC would then be tautological). A more informative quantification might be a ratio of dIgA/CD138 to get a measure of dIgA production per PC instead.

Indeed, the reviewer raises an important point regarding the amount of colonic dIgA in relation to the number of mucosal B cells/PBs/PCs. The significantly decreased dIgA/BCMA ratio in CD^{rem} patients from Western blot experiments reflects a combined effect of increased BCMA, shown in Figure 2c of the original manuscript, and a strong trend towards reduced dIgA levels, depicted in Figure 7a. Additionally, decreased dIgA/total IgA and dIgA/mIgA ratios in CD^{rem} patients compared to non-IBD controls are displayed in Supplementary Figure 2g of the original manuscript. Increased levels of BCMA could be validated by upregulated expression of TNFRSF17, the gene encoding BCMA, in the colonic tissue of CD^{rem} patients, as shown in the RNA-seq data in Figure 1d of the original manuscript.

However, we respectfully disagree with the reviewer's assumption that the increased BCMA levels are caused by the expansion of colonic B cells in these patients. According to published data, BCMA/TNFRSF17 is marginally expressed on B cells but is highly expressed in later differentiation stages like PBs and PCs. This is supported, among others, by single-cell transcriptomics data of the Human Gut Cell Atlas (Figure 7b), published by Elmentaite et al. in Nature (2021)¹. Given that CosMx spatial single-cell proteomics revealed no increase in PCs but elevated PBs (Figure 2g of the original manuscript file), we conclude that the increased BCMA levels in these patients are primarily driven by mucosal PB expansion. We understand the reviewer's suggestion that normalizing dIgA levels to CD138 would be more informative in this context. However, while CD138 is a well-established marker for identifying PCs within immune cell populations, its interpretation in colonic tissue is confounded by its lower mRNA expression in epithelial cells (Figure 7c). In the CosMx analyses, selection of Vim⁺ cells allowed us to focus on CD138⁺ LP cells. However, in Western blot experiments, CD138⁺ PCs cannot be distinguished from CD138⁺ epithelial cells by the commercially available antibodies we have utilized. Considering the specific expression of BCMA on PBs and PCs, in contrast to B cells, and the fact that CD138 is also

expressed by epithelial cells in colonic tissue, we consider the dIgA/BCMA ratio as a valid indicator of the amount of dIgA secreted per PB/PC. We hope that the reviewer agrees with it.

To illustrate specific expression of TNFRSF17 (BCMA) in PBs and PCs, but not B cells, we have included the respective data from the Human Gut Cell Atlas as Supplementary Figure 1i in the revised manuscript and included this information on page 8 lines 15-16 (highlighted in yellow).

Fig. 7 | Colonic dIgA levels per PB/PC. (a) Dimeric IgA levels were quantified in colonic biopsies from CD^{rem} patients and non-IBD controls using Western blot experiments. (b) and (c) *TNFRSF17* (BCMA; b) and *SDC1* (CD138; c) expression in colonic B cells, PBs/PCs and epithelial cells. Single-cell transcriptomics data originate from the publicly available Human Gut Cell Atlas¹. (a) Mann-Whitney *U* test. One outlier was identified and removed by the Grubb's test ($\alpha = 0.05$).

(5) Related to this, could a reduction in dIgA production be due to an increase in monomeric IgA production? Can the method used accurately quantify mIgA?

The authors appreciate the reviewer's comment and can follow their line of thought. To our knowledge, there is no commercial ELISA available for the specific detection of monomeric or dimeric IgA². Considering this, along with the limited sample volume, we believe that non-reducing Western blotting represents the most suitable method to differentiate between the two IgA forms in this context. However, in response to a comment raised by reviewer 2, we optimized the Western blotting conditions, now using XT Tricine Running Buffer (Bio-Rad #1610790) in combination with a 3-8% CriterionTM XT Tris-Acetate polyacrylamide gel (Bio-Rad #3450131) to improve visualization of mIgA and dIgA (Figure 17a below). While the amount of dIgA relative to total IgA or mIgA was reduced in the colonic mucosa of CD^{rem} patients, the proportion of mIgA was increased (Figure 17b below). At this stage, it remains uncertain whether this reflects an increase of mIgA-secreting PCs or an expansion of IgA⁺ precursor cells. However, considering the broader context of our data, the latter appears more likely. Since the mIgA signal detected by Western blotting cannot differentiate between intracellular, membrane-bound, or secreted forms, the IgA levels measured in the supernatant of the ex vivo differentiated PCs provide a more accurate indication of secreted forms.

We substituted the original Western blot and corresponding quantifications in Supplementary Figure 2g with the optimized Western blot and quantification, which is now provided in Figure 3i and j and Supplementary Figure 2i in the revised manuscript. Additionally, we have addressed the increased proportion of mIgA relative to total IgA in the results section on page 9, lines 27-28.

(6) The data in figure 4 is very informative and convincingly shows an intrinsic defect in CD B cell differentiation into IgA-secreting PCs. Could the authors specify whether the ELISA used for quantification of IgA secretion in 4G detects only dIgA or whether it would also measure mIgA? If not, could the western blot quantification (from 4C) be utilised instead? I think this could be an important point in understanding the defect of CD PCs, as a reduction of J chain production could indicate more mIgA secretion which would also account for total reduced SIgA. Similarly, does the ELISA used discriminate between IgA1 and IgA2, as the latter seems to be disproportionately affected in CD?

The authors appreciate the reviewer's insightful comment regarding the experimental distinction of IgA forms in the supernatant of the ex vivo differentiated colonic PCs. The ELISA used for quantification of IgA secretion in Figure 4g of the original manuscript detects total IgA (monomeric and dimeric). To our knowledge, there is no commercial ELISA available that can differentiate between total IgA and dIgA².

In response, we have established an ELISA for the specific detection of dIgA during the course of the revision process. First, we characterized the IgA variants in different (purified) IgA samples via Western blotting (Figure 8a). Accordingly, we used monomeric IgA from human serum (lane 1), human IgA from colostrum (lane 2, mainly sIgA), human IgG1 from myeloma (lane 3), and recombinantly expressed human anti-CD38 mIgA2 (lane 6) as negative controls. Native human colon protein lysate (lane 5) contained all three IgA forms (mIgA, dIgA, and sIgA) and served as a positive control. As human IgA κ -UNLB (#0155K-01, SouthernBiotech) is mainly comprised of dIgA (lane 4), it was used as a standard in the ELISA. For specific detection of J-chain-bound dIgA, we coated the plate with increasing amounts of human recombinant pIgR protein or bovine serum albumin (BSA) and validated specific binding of human IgA κ -UNLB to pIgR (Figure 8b). Based on these results, we coated 150 ng pIgR/well and observed a concentration-dependent binding of human IgA κ -UNLB to pIgR, with saturation at higher concentrations (Figure 8c). The specificity of dIgA detection was validated by the absence of signal from mIgA and sIgA, whereas the native colon protein produced a strong signal (Figure 8d). Measuring the supernatant of ex vivo differentiated colonic PCs between Day 10-14 revealed significantly decreased dIgA levels for CD^{em}- versus non-IBD-derived PCs (Figure 8e), comparable to the total IgA ELISA depicted in Figure 4g of the original manuscript.

As we cannot directly quantify mIgA in the supernatant using ELISA, we calculated the ratio of dIgA levels measured by the newly established dIgA-specific ELISA to total IgA levels measured by ELISA in the original manuscript. The similar dIgA/total IgA ratio excludes a compensatory increase in mIgA secretion in CD^{em}- versus non-IBD-derived colonic PCs (Figure 8f). This is further supported by the observed positive correlation between dIgA and total IgA in the supernatant samples (Figure 8g).

Fig. 8 | Establishment of dimeric IgA-specific ELISA. (a) Profiling of different equimolar amounts of purified human IgA samples and native colon protein by Western blot experiment using the anti-human IgA antibody (Thermo Fisher Scientific, #PA1-74395). **(b)** Verification of binding specificity of human IgA κ UNLB (mainly dIgA) to human recombinant plgR. **(c)** Standard curve of human IgA κ UNLB with coating of 150 ng plgR/well. **(d)** Validation of binding specificity for dimeric IgA testing a positive and equimolar amounts of negative control samples. **(e)** Normalized dimeric IgA secretion by CD^{rem}-derived *ex vivo* differentiated colonic PCs between Day 10-14 compared to non-IBD-derived PCs (median set to 1) was quantified by ELISA. **(f)** Ratio of dIgA/total IgA in the supernatant of CD^{rem}-derived *versus* non-IBD-derived *ex vivo* differentiated colonic PCs between Day 10-14. **(g)** Correlation of dIgA and total IgA secretion by CD^{rem}- and non-IBD-derived *ex vivo* differentiated colonic PCs between Day 10-14 quantified using ELISA experiments. **(b)-(d)** Each data point represents the mean of technical duplicates with whiskers indicating the range. **(e)** T test with Welch's correction, **(f)** Mann-Whitney *U* test, **(g)** Spearman correlation and linear regression with 95% confidence interval (dashed lines).

As already depicted in Supplementary Figure 3e and Figure 4f of the original manuscript, we have quantified IGHA1 and IGHA2 mRNA expression in the *ex vivo* differentiated colonic PCs using qPCR. Although the effect was slightly less prominent compared to IGHA2 ($P=0.0619$), IGHA1 expression also tended to be decreased in CD^{rem}- compared to non-IBD-derived colonic PCs ($P=0.1469$). We here provide an additional analysis of the IGHA2/IGHA1 ratio, which was found to be comparable between CD^{rem} and non-IBD (Figure 9a). Based on these findings, we do not assume a shift in IgA subclass composition. Together, colonic PC-specific IGHA1 and IGHA2 expression levels display a strong positive correlation, with most CD^{rem}-derived colonic PCs expressing lower amounts of both, IGHA1 and IGHA2 (Figure 9b).

Fig. 9 | IGHA1/2 expression in ex vivo differentiated colonic PCs. (a) Ratio of IGHA2/IGHA1 expression and **(b)** correlation of IGHA1 and IGHA2 expression in CD^{rem}- and non-IBD-derived ex vivo differentiated colonic PCs at Day 14. **(a)** Mann-Whitney U test, **(b)** Spearman correlation and linear regression with 95% confidence interval (dashed lines).

As we are confident that it adds value to the manuscript, we included the quantified dIgA levels in the supernatant of the ex vivo differentiated PCs in Figure 4g of the revised version, together with the correlation between dimeric and total IgA secretion. The interpretation of the results is provided in the results section on page 11, lines 3-6 (highlighted in yellow). In the interest of reproducibility, we briefly described the procedure and validation of this newly established ELISA in the results section on page 23, lines 22-29 and added the respective data as Supplementary Figure 7 in the revised manuscript.

Further, we decided to include the IGHA2/IGHA1 ratio and the correlation of both isoforms as Figure 4f in the revised manuscript and added the corresponding interpretation in the results section on page 10, line 30 and page 11, line 1-3 (highlighted in yellow).

(7) Could the authors provide a more detailed explanation of what the images in Figure 6A are supposed to illustrate?

- The authors thank the reviewer for identifying the missing explanation of the immunofluorescence staining presented in Figure 6a of the manuscript. The images depict a co-staining of IgA and P32, a pivotal protein targeted to the mitochondria to facilitate mitoribosome assembly and enable the translation of mitochondrially encoded subunits of the electron transport chain^{3,4}. In previous studies, the authors could verify that mitochondrial translocation of P32 is essential for respiration in different cell models^{5,6}. Given the specificity of the antibody to mitochondrial P32, the strong signal observed in IgA⁺ LP cells underscores their high OXPHOS activity.

The authors added information on the function of P32 to the results section on page 12 lines 4-5 and to the legend of Figure 6a in the revised manuscript (highlighted in yellow) to provide a more comprehensive explanation.

(8) In figure 7, the overall number of mice used seems very low with only two mice per group in 7E and three mice in 7F and J. The authors should increase the number of biological replicates and perform all experiments on at least two occasions to demonstrate reproducibility. In addition, could the authors indicate why some experiments were performed only on male mice (qPCR) and some only on female mice (GFHP diet).

We appreciate the reviewer's careful reading and fully understand the concern regarding the limited sample size in Figure 7c-e (n=2 mice per group) and Figure 7f and j (n=3 mice per group).

Unfortunately, we are unable to further increase the number of biological replicates, as the breeding of the conplastic mouse strain C57BL/6J-mt^{FVB/NJ} has been unexpectedly discontinued and no living animals are currently available. Further details can be found in the separate file 'Official statement_mice availability' by PD Dr. Misa Hirose (breeding coordinator).

In addition to the four male mice (n=2 Atp8-mutant, n=2 B6-WT) in the splenic B-cell stimulation experiment (Figure 7c-e of the original manuscript), we also analyzed two female mice (n=1 Atp8-mutant, n=1 B6-WT). However, it is well established that mice exhibit sex-specific differences in immune system characteristics, with females generally displaying stronger adaptive immune responses^{7,8}. For this reason, we initially decided against including the data from the female mice in the analysis. However, several studies indicate that sex-related disparities in adaptive immunity become more evident only in middle-aged or older mice and are partially driven by hormonal factors⁹. Additionally, in an ongoing study using C57BL/6J WT mice, we compared the colonic proteome of males and females using liquid chromatography-coupled mass spectrometry (LC-MS) and observed that several adaptive immunity-associated proteins were upregulated in 17-month-old female mice compared to age-matched males (Figure 10a). Notably, such differences were absent in younger, 4-month-old animals (Figure 10a), which corresponds to the age of the Atp8-mutant and B6-WT mice used in our experiments. Data from the colonic compartment were validated by qPCR analyses performed on splenic samples from 4-month-old mice, which similarly showed no significant differences in Cd19, Sdc1 (CD138), or immunoglobulin heavy chain expression between females and males (Figure 10b).

Fig. 10 | Sex-specific differences in adaptive immunity in mice depend on age. (a) Colonic levels of humoral immunity-related proteins were quantified *via* liquid chromatography-coupled mass spectrometry (LC-MS) in 4-month-old (left) and 17-month-old (right) B6-WT mice and compared between sexes. (b) Comparison of expression levels of humoral immunity-associated transcripts in splenic tissue of 4-month-old male and female B6-WT mice using RT-qPCR. (a) and (b) Two-way ANOVA with Šidák's multiple comparisons test. *** $P \leq 0.001$.

Since the experiments presented in the manuscript were conducted using 3- to 4-month-old mice (median: 15.14 weeks, IQR: 14.86-18.00), we decided to include the data from the two female animals (n=1 Atp8-mutant, n=1 B6-WT) in Figure 7c-e and Supplementary Figure 5b of the revised manuscript in order to increase the statistical power (Figure 11a-d). Respective changes in the text have been highlighted in yellow and added to the methods section on page 19, line 14 and the results section on page 13, lines 13, 14, and 16.

As the percentages of CD138⁺ PCs in Atp8-mutant and B6-WT mice are very similar between MLNs and PPs (Supplementary Figure 5c of the original manuscript), we decided to combine the data of both organs into one graph to improve statistical power (Figure 11e) and hope that the reviewer agrees with it. PCs from MLNs and PPs are displayed as circles or squares, respectively, and the legend of Supplementary Figure 5g of the revised manuscript file has been updated accordingly (highlighted in

yellow). However, in Figure 7f of the revised manuscript, the data for MLNs and PPs are still presented separately.

Fig. 11 | Ex vivo IgG1⁺ PB differentiation of splenic B cells with mitochondrial dysfunction. (a) Generation of IgG1⁺ cells upon LPS+IL4 stimulation is shown for splenic B cells from three B6-WT mice. (b) Percentage of TMRE^{high} cells in LPS+IL4 stimulated or unstimulated splenic B220⁺ cells from *Atp8*-mutant (n=3) versus B6-WT mice (n=3). (c) Expansion of *Atp8*-mutant and B6-WT B cells under LPS+IL4 stimulation relative to the unstimulated controls. (d) B220 level of LPS+IL4 stimulated or unstimulated splenic B220⁺ cells from *Atp8*-mutant compared to B6-WT mice. (e) Percentages of B220⁺ and B220⁻ CD138⁺ PCs isolated from the MLNs (circles) and PPs (squares) of *Atp8*-mutant (n=3) and B6-WT mice (n=3). (a) Mann-Whitney *U* test, (b)-(e) Two-way ANOVA with Fisher's least significant difference (LSD) test. **P*≤.05, ***P*≤.01.

To validate the dysregulated B-cell compartment in a larger number of independent mice, we retrospectively analyzed spleen tissue samples from one of our previous studies involving n=8 *Atp8*-mutant and n=4 B6-WT mice, in which chronic colitis was induced by administering 2% DSS via the drinking water for three cycles of 5 days each, with recovery periods in between. Western blotting revealed significantly increased IgG levels in the spleen of *Atp8*-mutant compared to B6-WT mice (Figure 12a). Elevated IgG levels in the spleen of *Atp8*-mutant mice could reflect enhanced B-cell proliferation, as we also observed in isolated splenic B cells stimulated with LPS+IL4, shown in Figure 7d of the original manuscript. Consistent with the mitochondrial dysfunction confirmed by reduced *mt-Nd3* expression, PC numbers seem diminished, as indicated by significantly lower *Sdc1* mRNA levels in the spleen of *Atp8*-mutant mice (Figure 12b).

Fig. 12 | Dysregulated splenic B-cell compartment in *Atp8*-mutant mice under chronic DSS-induced colitis. Quantification of (a) splenic IgG levels by Western blot and (b) splenic expression of *mt-Nd3* and *Sdc1* mRNA via RT-qPCR for *Atp8*-mutant (n=8) versus B6-WT (n=4) mice under chronic DSS-induced colitis. Data are displayed as relative values to B6-WT mice. (a) Mann-Whitney *U* test, (b) Two-way ANOVA with Holm-Šidák's multiple comparisons test. **P*≤.05. AU, arbitrary units.

We decided to include these data as Supplementary Figure 5c and d of the revised manuscript and added the corresponding interpretation to the results section on page 13, lines 20-23 (highlighted in yellow). The approval number has been included in the methods section on page 19, line 5-6 and details of the animal experiment are described on page 19, lines 18-21 of the revised manuscript (highlighted in yellow).

However, the stimulation experiment of splenic B cells (Figure 7c-e of the original manuscript) was performed using six mice on two separate days. The six mice used for the analysis of MLNs and PPs (Figure 7f of the original manuscript) were also sampled on two different days, and the corresponding flow cytometry experiments were performed independently on each day to demonstrate reproducibility. The six mice used for colonic IgA IHC staining were sacrificed on two consecutive days, and the reduction of colonic IgA in *Atp8*-mutant mice was additionally confirmed by qPCR analysis in a substantially larger number of animals (Figure 7g of the original manuscript, n=12 per group). Moreover, it is important to note that different animals were used for the individual experiments (spleen, MLNs/PPs, colon), yet all consistently showed a compromised B-cell compartment in *Atp8*-mutant mice, supporting the reproducibility of this phenotype.

The predominant use of male or female mice in the *Atp8*-mutant or glucose-free high-protein diet (GFHPD) experiment, respectively, is attributable to technical reasons.

The predominant use of male *Atp8*-mutant and matching WT mice was not intentional, but resulted from the fact that almost exclusively male mice were available from the breeding at that time. However, for the analysis of mesenteric lymph nodes (MLNs) and Peyer's patches (PPs), female *Atp8*-mutant and B6-WT mice were used. To improve clarity for the readership, we now explicitly state the sex of the animals in the corresponding legends of Figure 7b, g, j, and l and Supplementary Figure 5i in the revised manuscript file (highlighted in yellow).

The female GFHPD- and isoenergetic chow diet-fed mice originate from the control arm of a large-scale mouse study¹⁰. In this study, colonic tumors were induced by AOM/DSS treatment in half of the mice. After 10 weeks, the mice were assigned to different intervention groups (GFHPD/chow diet; anti-EGFR mAb/isotype ctrl. mAb/PBS) in a way that ensured approximately equal body weight distribution across groups. Reorganization of cage groups, however, is only feasible with female mice, as male mice that were not raised together tend to fight when housed in the same cage.

(9) The authors use in vitro differentiation of splenic B cells to show a defect in the capacity of *Atp8*-mutant cells to develop into IgG-producing PCs, however the relevance of these data is unclear. Since these experiments are supposed to provide a mechanistic explanation for the human data, which shows a selective defect in IgA-PC induction, the authors should utilise conditions that favour induction of IgA producing PCs (i.e. IL-6, IL-21, TGFb) for their in vitro assay.

We understand the reviewer's perspective and would like to clarify that the purpose of the *Atp8*-mutant versus B6-WT mouse experiments was to demonstrate a general impairment in PC differentiation, which is evidently not restricted to mucosal IgA⁺ PCs. The defect in the colonic PC compartment is subsequently characterized in Figure 7g-j of the original manuscript. We also attempted to differentiate isolated splenic resting B cells into IgA⁺ PBs/PCs by stimulation with 10 ng/ml IL-6 and 0.1 μM retinoic acid (RA). However, this approach was not successful, as the cells did not proliferate, comparable to the unstimulated controls (Figure 13).

Under physiological conditions, IgA class switch recombination (CSR) does not typically occur in the spleen, but primarily takes place in mucosa-associated lymphoid tissues, such as PPs, MLNs, or LP. As a systemic lymphoid organ, the spleen predominantly supports CSR to IgG isotypes, especially IgG1 and

IgG2a. Therefore, we focused here on the *ex vivo* differentiation of LPS-activated B cells into IgG1⁺ PBs/PCs, which is specifically induced by IL-4 stimulation^{11,12}.

Fig. 13 | *Ex vivo* differentiation of splenic B cells from *Atp8*-mutant and B6-WT mice into PBs. (a) Expansion of *Atp8*-mutant and B6-WT splenic B cells under LPS+IL-4 or IL-6+retinoic acid (RA) stimulation for four days. **(b)** Representative contour plots of B220 and IgG1 (top) or IgA (bottom) on splenic B cells of a B6-WT mouse upon stimulation with LPS + IL-4 (top right) or IL-6+RA (bottom right). **(a)** Two-way ANOVA with Tukey's multiple comparisons test. ** $P \leq .01$, *** $P \leq .001$.

(10) Similarly, the relevance of Figure 7F is uncertain. Since the main phenotype observed in human concerns colonic PC in the lamina propria, it is unclear why the authors chose to analyse lymphoid tissues associated with the small intestine (MLN and PP). This analysis should be complemented by FACS (or multi-coloured immunohistochemistry) of PCs in the colonic LP. This has the added benefit that it would allow for a much more detailed phenotypic description of these cells than the rather uninformative “IgA+ cells” obtained from the histology in 7J and 7I.

Here, the authors would like to point out that the aim was to demonstrate a systemic impairment of B-cell/PC differentiation in *Atp8*-mutant compared to B6-WT mice. Accordingly, we included different organs and tissues for analysis, such as spleen, MLNs, PPs, and colonic mucosa.

Since we no longer have access to additional conplastic C57BL/6J-*mt*^{FVB/NJ} mice (see comment 8 above), we unfortunately are not in a position to carry out the experiments suggested by the reviewer at this time. However, we would like to draw the reviewer's attention to the qPCR data presented in Figure 7g and Supplementary Figure 5d of the original manuscript, which characterize the PC compartment in the colonic LP in more detail. Notably, the reduced expression of colonic B-cell and PC markers such as *Cd19*, *Igha*, *Tnfrsf17*, *Sdc1*, and *Prdm1* in *Atp8*-mutant animals is reflected in the decreased number of IgA⁺ cells observed by IHC (Figure 7j of the original manuscript).

Regarding the glucose-free high-protein diet (GFHPD)- versus isocaloric chow diet-fed mice, we fully agree with the reviewer that quantifying IgA⁺ cells in the colonic mucosa alone (Figure 7I of the original manuscript) is not sufficiently informative. Therefore, we performed the following additional experiments to further characterize the effect of OXPHOS-promoting dietary intervention on colonic humoral immunity in mice. First, we quantified the expression of various markers along the B cell to PC lineage in colonic tissue from the same animals analyzed by IHC using qPCR. We observed that both, B-

cell markers, such as *Cd19*, *Cd79A*, and *Ighm*, and the PB/PC-expressed *Tnfrsf17*, which encodes BCMA, were significantly upregulated in GFHPD-fed compared to chow diet-fed mice (Figure 14a). These findings suggest that the dietary intervention not only promotes PC differentiation but also affects the entire mucosal B-cell compartment. Furthermore, we performed non-reducing Western blotting using fecal protein samples collected after 10 weeks on the respective diets to quantify IgA. In line with our qPCR and IHC data, we detected significantly increased total IgA and dIgA levels in fecal samples from GFHPD-fed mice compared to controls (Figure 14b), indicating a promoting effect on IgA⁺ PC maturation. We decided to include these data in the revised manuscript file as Supplementary Figures 5j and k. Respective changes in the results section and discussion have been highlighted in yellow and can be found on page 14, lines 11-14 and on page 17, lines 13-14, respectively.

Fig. 14 | Effect of glucose-free high-protein diet (GFHPD) on colonic humoral immunity in mice. (a) Colonic expression of different B-cell and PC markers was quantified for GFHPD-fed (n=6) and chow diet-fed mice (n=5) *via* RT-qPCR. Data are displayed as relative values to chow diet-fed mice for two independent experiments. **(b)** Quantification of fecal IgA levels in mice after 10 weeks of nutritional intervention (chow diet n=3, GFHPD n=3) using non-reducing Western blotting. **(a)** Mixed-effects model with Fisher's least significant difference (LSD) test, **(b)** Two-way ANOVA with Fisher's LSD test. * $P \leq .05$, ** $P \leq .01$.

In this manuscript Raschdorf and colleagues study the B cell compartment in Crohn's disease patients in remission. Several recent studies evaluated the composition and activity of B cells in inflammatory bowel disease. As stated by the authors, the main difference with the current study is the fact that in the earlier studies, only patients with active disease were included, or disease activity was not addressed specifically. I agree with the authors that disease in remission is an interesting subject in its own right, but I do have several concerns regarding this study.

Major general concerns:

(1) The authors emphasize the fact that they want to study patients in remission. However, when assessing their patient characteristics, a considerable proportion of the 'remission' patients had macroscopic inflammation in the ileum at the time of sampling. Although I realize the samples were obtained from the colon, this does stress these patients were in fact not in remission. For any systemic measurements (stool/serum) this is a problem in itself, and for the specific colonic analyses, the study is actually not different from the 'non-inflamed' samples which have in fact been included in earlier studies (e.g doi.org/10.1038/s41591-022-01680-y). As this concerns up to 33% of patients in many analyses, this is the difference between showing activity in actual remission, or just repeating earlier studies.

We thank the reviewer for drawing attention to this critical aspect. At this point, we would like to emphasize that our study focuses on CD patients in clinical remission, which refers to a state with minimal or no patient-reported clinical symptoms¹³⁻¹⁵. In addition, all patients included had confirmed endoscopic remission in the colon, defined as the absence of macroscopic evidence of inflammation during colonoscopy. We have clarified this accordingly in the methods section on page 18 line 15 of the revised manuscript file (highlighted in yellow).

We have reported comparable numbers of isolated colonic LPMNCs from CD^{rem} patients and non-IBD controls in Supplementary Figure 1b of the original manuscript. To further support the absence of local inflammation in the colonic mucosa, we now additionally included histological assessment of LP cell density as Supplementary Figure 1b of the revised manuscript file. In contrast to previous studies that typically rely solely on the non-inflamed appearance of the analyzed biopsy¹⁶ – while adjacent areas may still be inflamed – we confirmed endoscopic remission throughout the entire colon at the time of sampling.

As correctly noted by the reviewer, three of the CD patients in this study showed macroscopic signs of inflammation in the ileum during endoscopy. However, this mostly represented low-grade inflammation at the ileocolonic anastomosis in patients with prior ileocecal resection. In two patients, inflammatory signs were classified as Rutgeerts score 1 or 2, respectively, whereas the third patient displayed only some aphthous lesions on the ileocecal valve, with no inflammation in the ileum itself. The reviewer is indeed correct that active ileal inflammation in clinical remission could potentially influence systemic plasma analyses. However, as these three patients did not show differences in plasma CRP levels, immunoglobulin levels, or metabolomics compared to patients without endoscopic signs of ileal inflammation (Figure 15a-c), we assume that this minor inflammation of the anastomosis does not exert a systemic effect. Particularly, we can exclude that differences observed in plasma metabolomics between CD^{rem} patients and non-IBD controls are driven or biased by these three patients.

Fig. 15 | Plasma CRP, immunoglobulin levels, and metabolites in CD^{rem} patients. CRP levels **(a)** and immunoglobulins **(b)** were quantified in plasma samples from CD^{rem} patients and non-IBD controls using liquid chromatography-coupled mass spectrometry (LC-MS). **(c)** Significantly regulated plasma metabolites (top) and lipoproteins (bottom) in CD^{rem} patients (bars) compared to the reference average of non-IBD controls (dotted center line) were determined *via* ¹H NMR spectroscopy. Patients with macroscopic signs of minor ileal inflammation are indicated in green. **(a)** Kruskal-Wallis test with Dunn's multiple comparisons test, **(b)** and **(c)** Multiple Mann-Whitney *U* test. AU, arbitrary units.

To provide a more detailed definition of remission, we changed the sentence in methods section on page 18, lines 17-20 as follows:

'CD patients for blood draw and endoscopic removal of colon biopsies were classified as being in remission if they were in clinical remission, the colon presented without macroscopic evidence of mucosal inflammation, and the ileum showed no or at most minor inflammatory signs during endoscopy.'

Moreover, two patients with active CD were inadvertently omitted from the initial analysis of plasma CRP levels in the original manuscript and have now been included in Supplementary Figure 1d and Supplementary Table 4 of the revised manuscript.

The quantification of IgA levels in stool samples does not include these patients, as it was performed in an independent cohort of CD patients who were classified as being in clinical remission based on the following criteria: Harvey-Bradshaw Index (HBI) ≤ 3 , CRP ≤ 5 mg/l and fecal calprotectin ≤ 100 μ g/g (methods section page 17 lines 15-17 of the original manuscript).

Since our tissue-level analyses focused specifically on the colon, we are convinced that our study provides novel insights into the humoral immune response in the colonic mucosa of CD patients in the absence of local inflammation. This distinguishes our study from previous works where inflamed and non-inflamed biopsies from patients with active colonic inflammation were compared to non-IBD control biopsies. This approach neglects the fact that 'non-inflamed' biopsies may have been affected by neighbouring inflamed areas, for example via migratory immune cells.

(2) On an analytical level, this study lacks rigor. The biggest problem is an unacceptable level of cherry picking in the statistics. In several panels, effects which they deem relevant are described as 'increased' or 'decreased' without acknowledgement of the fact that the difference lacks statistical significance. At the same time, effects apparently deemed unwanted are described as 'not different' and mentioned without any p-values, just as 'NS'. An example is figure 2c, where the change in IgM is described as 'increased' (p value 0.1), while in figure 2g the proportion of PC is 'not upregulated', even though there is a clear trend as well, and no p value is provided. This is just one example, but occurs throughout the manuscript. The authors need to be consistent in their description (it is

fine to describe non-significant effects as well, but then for all) and provide p values for all comparisons.

We thank the reviewer for pointing out this inconsistency in the description of differences and effects, that are not statistically significant. The authors fully agree with the reviewer that only statistically significant effects should be described as 'increased' or 'decreased', whereas non-significant comparisons showing an apparent tendency toward one direction should be interpreted as trends. In response, we have carefully revised the results section and highlighted changes in yellow (page 7 lines 17-19, page 8, lines 11, 13, and 18, page 9 lines 14-15, page 10 lines 25 and 30, page 13 line 19, page 14, line 5).

We acknowledge that a visual difference in the number of PCs in Figure 2g of the original manuscript may be perceived. Nevertheless, we decided not to describe this tendency as a trend, as it is driven by only two higher data points, and the P value from the Two-way ANOVA with Šidák's post hoc test is 0.8995 (see Figure 2g of the revised manuscript file). Applying a Mann-Whitney U test to the comparison of PC numbers between CD^{rem} and non-IBD samples would still yield a P value of 0.5887. The authors hope that the reviewer agrees with this decision.

In the original manuscript, we indicated statistically significant differences ($P \leq 0.05$) with an asterisk (), reported P values between 0.05 and 0.1 explicitly, and marked comparisons with $P \geq 0.1$ as 'ns' (not significant). However, we agree with the reviewer that providing P values for all comparisons increases transparency and interpretability. We have therefore updated the figure panels of all main and supplementary figures to include P values for all differences, regardless of their significance level.*

Other general concerns:

(1) the use of low sample numbers, which for sure has underpowered several analyses. Already in the first paragraph, the authors mention there was no clustering of samples in the PCA plot based on medication. However, most medications only included a single patient, and therefore can not possibly cluster. RNASeq analysis with primary material using 12 vs 7 samples is very limited etc. This is also evident from the volcano plot (fig 1b) which depicts p values rather than p values adjusted for high level multiple testing (Bonferroni or False Discovery Rate) as should be done for RNASeq. Using the appropriate metric would show no significant DEG.

Of course, we share the reviewer's opinion that increasing the number of patients would likely enhance statistical power. At this point, we would like to emphasize that the focus of our study was on in-depth phenotyping of CD patients in remission, involving technically demanding and resource-intensive approaches such as spatial single-cell proteomics, which currently do not allow for high-throughput analysis. In addition, we validated our results by ex vivo differentiation of patient-derived switched memory B cells, isolated from colonic biopsy samples. Moreover, due to the fact that such clear differences observed between CD^{rem} patients and non-IBD controls are already evident in a relatively small cohort, we are convinced of the strength and robustness of the identified effects.

We agree with the reviewer that RNA-seq analyses should be corrected for multiple testing. In our analysis, using the Benjamini-Hochberg FDR method, no genes reached the conventional threshold of $FDR < 0.05$, likely due to the small sample size (7 versus 12 patients) and the associated limited statistical power. For visualization in Figure 1b of the manuscript, we presented unadjusted P values to illustrate the distribution of differential expression across the transcriptome, a practice also used in exploratory RNA-seq studies. To ensure transparency and accuracy, we have now explicitly stated in the legend of Figure 1b of the revised manuscript that the volcano plot displays unadjusted P values (highlighted in yellow). Importantly, the finding of an expanded B-cell compartment in the colonic mucosa of CD^{rem} patients using RNA-seq was orthogonally validated in subsequent experiments

involving qPCR of isolated patient-derived colonic LPMNCs, Western blotting, and spatial single-cell proteomics. These follow-up analyses confirmed the increased expression of B cell-related targets in the colonic mucosa of CD^{rem} patients despite the lack of FDR significance in the discovery dataset.

The authors fully agree with the reviewer's comment that the RNA-seq data presented in Supplementary Figure 1a of the manuscript cannot meaningful cluster based on medication, given that the respective treatment groups only include 1-2 patients each. Hence, we edited the respective sentence in the results section on page 7, lines 7-8 as follows:

'Although principal component analysis (PCA) did not reveal a clear separation of CD^{rem} patients under different medications from non-IBD controls (Supplementary Figure 1a),...'

Moreover, the reviewer's valuable comment prompted us to stratify fecal IgA levels in CD^{rem} patients by medication, as this analysis was conducted in a larger cohort, comprising n=47 CD^{rem} patients. Stratification revealed that fecal IgA levels did not differ between patient medication groups (Figure 16). We decided to include the graph as Supplementary Figure 1j in the revised Supplementary Material file and hope that the reviewer agrees with it. Respective changes in the text have been highlighted in yellow and added to the results section on page 8, lines 19-20.

Fig. 16 | Stratification of fecal IgA levels in CD^{rem} patients (n=43) according to medication. Only treatment groups with ≥ 3 patients are displayed. Kruskal-Wallis test with Dunn's multiple comparisons test. * $P \leq .05$. 5-ASA, Mesalazine; AZA, Azathioprine; BUD, Budesonide; PRED, Prednisolone; RZB, Risankizumab; UST, Ustekinumab; VDZ, Vedolizumab.

Of course, the effect of medication on the dysregulated humoral immune response in CD patients needs to be investigated in more detail in follow-up studies including larger cohorts, which is now addressed in the revised discussion on page 17, lines 14-16 (highlighted in yellow).

(2) Use of less than conventional methods. Determination of immune cell subsets is usually performed by surface marker evaluation rather than transcription, in particular bulk expression. For example in figure 2g/h, immune cells were actually isolated from the colon, but rather than perform flow cytometry, were assessed by bulk expression. Production and secretion of antibodies is consistently measured by Western Blot, while this is mostly done by more specific ELISA assay (which can be done on the same sample type). This makes quantification much more reliable, as the western blots are not all convincing (eg Suppl figure 2g really does not show reliable bands, and thus quantification is not very meaningful).

We appreciate the reviewer's insightful comment regarding some of the methods used in this study. Since there is no Figure 2h in the manuscript and Figure 2g does not match the context of the comment, we have interpreted the reviewer's remarks as referring to Figure 1g/h. Should this assumption be incorrect, we kindly ask for clarification.

The authors understand the reviewer's concern regarding our choice to use RNA-seq for the initial profiling of colonic immune cell markers (Figure 1d of the original manuscript), as well as qPCR for the characterization of B-cell markers in isolated colonic LPMNCs (Figure 1h of the original manuscript), instead of the more conventional method of flow cytometry based on surface marker expression. However, since the majority of the colonic biopsies and isolated LPMNCs were allocated to the enrichment of sMBCs and subsequent ex vivo PC differentiation, the remaining cell numbers were insufficient to allow for reliable flow cytometric analyses with appropriate controls. As an estimate, we isolated on average approximately 1.5 million LPMNCs from seven biopsies per patient. Based on the data from the Human Gut Cell Atlas¹, memory B cells represent about 3-4% of colonic immune cells – corresponding to roughly 50,000 memory B cells per sample, of which only the switched subsets were enriched in our experiments. At this point, we would further like to emphasize that the finding of an expanded colonic B-cell compartment in CD^{em} patients, as identified on the transcriptional level, has been orthogonally validated and further refined by spatial single-cell proteomics.

We agree with the reviewer that ELISA is typically the preferred method for quantitative analysis of antibody secretion. In our study, however, the primary reason for using Western blotting was the ability to distinguish between monomeric and dimeric forms of IgA in the colonic tissue based on their molecular weight. To our knowledge, there is no commercial ELISA available that can specifically differentiate between mIgA and dIgA². In response to comment 6 from reviewer 1, we established an ELISA for the specific detection of dimeric IgA in the supernatants of the patient-derived ex vivo PCs in the course of this revision process. Since the protein samples from colonic biopsies used for Western blotting were prepared under denaturing conditions, we are not able to use these protein extracts in our newly established dIgA-specific ELISA, which is based on the binding of native dIgA to the polymeric immunoglobulin receptor (pIgR).

The authors can to some extent understand the reviewer's view that the Western blot in Supplementary Figure 2g of the original manuscript may not appear entirely convincing. In response, we optimized the Western blotting conditions, now using XT Tricine Running Buffer (Bio-Rad #1610790) in combination with a 3-8% CriterionTM XT Tris-Acetate polyacrylamide gel (Bio-Rad #3450131) to improve visualization of mIgA and dIgA. While the amount of dIgA relative to total IgA or mIgA was reduced in the colonic mucosa of CD^{em} patients, the proportion of mIgA was increased. At this stage, it remains uncertain whether this reflects an increase of mIgA-secreting PCs or an expansion of IgA⁺ precursor cells. However, considering the broader context of our data, the latter appears more likely.

The non-reduced Western blot of human colonic samples shown in Supplementary Figure 2g of the original manuscript has now been replaced with the optimized Western blot (Fig. 17a) in Supplementary Figure 2i of the revised version. In addition, the quantifications of IgA (Fig. 17a and b) in Figure 3i and j of the revised manuscript have been updated accordingly. Respective changes in the text have been highlighted in yellow and added to the results section on page 9, lines 27-28 and the methods section on page 23, line 5. However, due to the extensive glycosylation of IgA molecules¹⁷, discrete bands are not expected on non-reducing Western blots of denatured proteins. As distinguishing between monomeric and dimeric IgA in the tissue was important for our analysis, we consider the non-

reducing Western blot to be the most appropriate method for this purpose and hope that the reviewer can appreciate this rationale.

Fig. 17 | Monomeric and dimeric IgA levels in the colonic mucosa of CD^{rem} patients. (a) Detection of non-reduced mIgA and dIgA in colonic biopsies of CD^{rem} patients (n=9) and non-IBD controls (n=13) *via* Western blotting (left) and densitometric quantification of relative monomeric and dimeric IgA levels (right). **(b)** DlgA/total IgA and dlgA/mIgA ratios quantified *via* Western blotting. **(a)** Multiple Mann-Whitney *U* test, **(b)** Two-way ANOVA with Šidák's multiple comparisons test. ***P*≤.01, ****P*≤.001. AU, arbitrary units.

(3) The role of metabolic changes seems to be an overreach. Metabolomics are performed on plasma, and can therefore not be correlated to local B cells. Bulk RNASeq similarly does not indicate the involvement of B cells. The mechanism is mainly suggested based on RNA transcription (at best suggestive, not definitive for metabolic activity) and correlations. If the authors really want to state that decreased OXPHOS is involved in the impaired differentiation of PC in IBD, they need to perform the actual experiment, which would be Seahorse assays on the B cells themselves.

We fully agree with the reviewer that plasma metabolomics cannot be directly correlated with metabolism in the colon or with the metabolic state of local B cells. However, systemic and local metabolism are interconnected to some extent, and changes in circulating metabolite levels may indirectly reflect or influence local metabolic processes.

In our study, analyses were performed starting with systemic NMR-based plasma metabolomics, followed by assessment of mucosal metabolism and finally metabolic gene expression analysis of ex vivo differentiated colonic PCs. In line with the increased CD38 levels on PCs, which we quantified in situ using spatial single-cell proteomics and which have been described in the literature as reducing OXPHOS¹⁸⁻²⁰, all these experiments suggest a reduced mitochondrial activity in the colon and in ex vivo PCs, albeit indirectly, as indicated by decreased expression of various electron transport chain (ETC) transcripts. Conversely, we have shown that a mitochondrial defect in conplastic Atp8-mutant mice resulted in a disturbed B-cell compartment and reduced humoral immune responses.

The authors agree with the reviewer that the Seahorse assay represents the most suitable method for directly assessing OXPHOS and glycolytic activity in cells. However, it is important to note that Seahorse analyses are performed under in vitro cell culture conditions, which do not fully reflect the physiological environment of the tissue. Therefore, spatial single-cell-based metabolic analyses are a main focus of our group in ongoing studies. As mentioned above, the limited number of cells prevented us from performing Seahorse analyses on the patient-derived ex vivo PCs. As an alternative, we have now quantified D-glucose consumption and L-lactate release in respective cell culture supernatants, providing additional insights into the metabolic activity of these cells. While D-glucose uptake was below the detection limit in nearly all samples, CD^{rem}-derived PCs displayed reduced L-lactate release compared to non-IBD-derived PCs (Figure 18a). In combination with the suggested mitochondrial

dysfunction, these findings indicate that the CD^{rem}-derived ex vivo PCs are likely in a quiescent metabolic state (Figure 18b).

Fig. 18 | Glucose uptake and lactate release of ex vivo differentiated colonic PCs. (a) Normalized D-glucose consumption and L-lactate release were quantified in the supernatant of CD^{rem}- and non-IBD-derived ex vivo differentiated colonic PCs between Day 10-14. (c) Metabolic plot displaying mean \pm SEM of D-glucose consumption versus L-lactate production by ex vivo PCs. (a) Two-way ANOVA with Šídák's multiple comparisons test. * $P \leq .05$. AU, arbitrary units.

The fact that PCs from both groups lacked glucose consumption suggests that they may rely on alternative energy sources, such as amino acids (AA). This is supported by single-cell transcriptomics data of the Human Gut Cell Atlas, published by Elmentaite et al. in Nature (2021)¹. While SLC2A1 and SLC2A3 expression, encoding the glucose transporters GLUT1 and GLUT3, is nearly absent in colonic IgA⁺ PCs, AA transporters, encoded by SLC3A2, SLC38A2, SLC1A5, SLC38A1, SLC7A5, and SLC38A5, are expressed by different subpopulations (Figure 19a). In qPCR analyses utilizing non-IBD-derived colonic ex vivo PCs, glucose transporters displayed lower mRNA expression levels compared to the AA transporter (subunits) SLC3A2 and SLC38A2 (Figure 19b). SLC3A2 encodes a subunit of the AA transporters LAT1 and LAT2, which primarily mediate the uptake of large neutral AAs, including the branched-chain AAs leucine, isoleucine, and valine, which are crucial for cellular energy metabolism. Similarly, SLC38A2 encodes the AA transporter SNAT2, which facilitates the uptake of small neutral AAs, including glutamine – another key metabolic substrate.

Fig. 19 | Expression of glucose and amino acid (AA) transporters in colonic PCs. (a) Glucose transporter (magenta) and AA transporter (blue) expression in single-cell transcriptomics data of colonic IgA⁺ PCs from healthy adults, publicly available in the Human Gut Cell Atlas¹. (b) Expression of the different glucose and AA transporters, shown in (a), was quantified in non-IBD-derived ex vivo differentiated colonic PCs (n=9) using RT-qPCR.

However, no statistically significant differences were observed in AA transporter mRNA expression between CD^{rem} and non-IBD-derived colonic PCs (Figure 20a and b).

Fig. 20 | Expression of metabolic transcripts in patient-derived ex vivo differentiated colonic PCs. (a) Levels of metabolic enzyme(s) (subunits) and nutrient transporters in CD^{rem}- and non-IBD-derived ex vivo differentiated colonic PCs on Day 14 were quantified using RT-qPCR and depicted as median relative to non-IBD with mitochondrially encoded genes highlighted in a green box. **(b)** 2^{-ΔCt} values for metabolic enzyme(s) (subunits) and nutrient transporter expression in ex vivo differentiated colonic PCs from CD^{rem} patients and non-IBD controls. **(a)** and **(b)** Two-way ANOVA with Fisher's least significant difference (LSD) test. *P≤.05, **P≤.01.

To provide a more comprehensive metabolic characterization of the ex vivo differentiated PCs, we decided to include the graphs displaying glucose consumption and lactate release (Figure 18) as Figure 6h in the revised manuscript. The corresponding interpretation has been added to the revised results section on page 12, lines 22-30 and page 13, line 1. The D-glucose and L-lactate assay can be found in the methods section on page 21, lines 10-13 and lines 14-17, respectively, of the revised manuscript file (highlighted in yellow). Instead, the correlation matrix of CD38 mRNA expression with the downregulated mitochondrial transcripts has now been moved from Figure 6g of the original manuscript to Supplementary Figure 4d. Moreover, we added the expression levels of AA transporters to the heatmap in Figure 6i and to the boxplot in Supplementary Figure 4g of the revised manuscript file. The additional primer sequences used for RT-qPCR are now provided in Supplementary Table 5. More extensive metabolic phenotyping of colonic PCs in both healthy individuals and CD patients is planned as part of follow-up studies by our group.

(4) I don't understand the use of Vimentin as a positive marker in this study. I understand the need to correct in some way for the amount of lamina propria included in the respective samples. However, this is commonly done using DAPI or another nuclear/DNA stain. Correcting against total Vimentin as a LP marker is unusual but I can understand to some degree. However, the authors refer to the plasmablasts/cells (as well as T cells) as Vim+, which seems odd, as this is not a marker for these cells at all?

We appreciate the reviewer's thoughtful comment regarding the normalization of immune cell counts in the spatial single-cell proteomics analysis. We agree with the reviewer that normalization of cell counts within tissues is commonly done using a nuclear stain. However, using DAPI in the CosMx™ analyses would cover total cells of the FFPE section, including LP cells and epithelial cells. As already outlined in our response to comment 2 from reviewer 1, Vimentin staining is suitable in this context as it specifically labels cells of the LP, while effectively excluding all epithelial cells (Figure 21a). This is supported by single-cell transcriptomics data of the Human Gut Cell Atlas, published by Elmentaite et al. in Nature (2021)¹. While all LP cells, such as immune cells, endothelial cells, neuronal cells, and mesenchymal cells, express VIM, it is absent in colonic epithelial cells (Figure 21b). To account for varying proportions of LP in the different biopsy sections, we normalized the immune cell counts to the

number of Vim^+ cells, as depicted in Figure 1e of the original manuscript. This is also described in the methods section on page 23, lines 1-2 of the original manuscript file. We hope the reviewer appreciates the rationale behind this. Supplementary Figure 1b of the revised manuscript now contains a representative image from the CosMx™ SMI, displaying the specific Vimentin staining of the colonic LP and the quantification of Vim^+ cells per tissue area.

Fig. 21 | Vimentin serves as a marker for colonic lamina propria (LP) cells. (a) Representative image of Vimentin staining (yellowish green) of the colonic LP cells of a non-IBD control in the CosMx™ SMI. **(b)** *VIM* (Vimentin) expression in colonic immune, endothelial, neuronal, mesenchymal, and epithelial cells. Single-cell transcriptomics data originate from the publicly available Human Gut Cell Atlas¹.

Other specific concerns:

(1) Several graphs contain delta-Ct values as a measure of absolute expression. Delta-CT is a perfectly fine method for assing relative expression of a particular gene, but can not be compared between different primers, and thus different genes. These values should not be in a single graph (e.g. Suppl fig 2e), and definitely not in a stacked bar such as figure 4e. The statement ‘cells mainly expressed CD20, CD27 etc’ is incorrect from this figure (I do believe they do, but the conclusion can not be drawn from this figure).

*The authors thank the reviewer for their comment. However, as Supplementary Figure 2e does not show any qPCR data, we assume the reviewer may have intended to refer to Supplementary Figure 3b instead. We agree with the reviewer that $2^{-\Delta Ct}$ values are not suitable for direct comparison of absolute expression levels between different genes due to varying primer efficiencies and amplification kinetics. We would like to point out that our intention in Supplementary Figure 3b was not to quantify or rank absolute expression levels across genes. Rather, we aimed to demonstrate that B-cell-specific transcripts, such as *MS4A1*, *TNFRSF17*, *CD27*, and *CD38*, were clearly expressed, while targets commonly associated with T cells, natural killer (NK) cells, dendritic cells, monocytes, macrophages, granulocytes, and erythrocytes were close to zero or undetectable in the qPCR data, with the exception of *CD16* and *CD56*, which showed a signal. Due to the very limited number of enriched colonic switched Memory B cells (sMBCs), flow cytometry analysis was not feasible, and qPCR remained the only available approach to characterize this cell population. To ensure robust identification, we typically quantified multiple target genes per immune cell type.*

While we acknowledge that this does not guarantee identical primer efficiencies, we aimed to minimize technical variability across genes to the extent possible. All reactions were performed on the same qPCR instrument using the same program with an annealing temperature of 55 °C and the same detection

threshold. Moreover, primers were ordered from the same supplier (Metabion international AG) and were designed with similar parameters (Table 1).

Tab. 1 | Parameters of primers used for characterization of enriched colonic switched Memory B cells (sMBCs) via RT-qPCR in Supplementary Figure 3b. Median (range) values are displayed for primer length, amplified product length, GC content, melting temperature (T_m), and difference between T_m of forward and reverse primer.

primer length [bp]	product length [bp]	GC content [%]	T_m [°C]	ΔT_m (forward/reverse)
20 (19 - 23)	212.0 (146 - 382)	53.82 (39.13 - 57.9)	58.94 (57.31 - 60.06)	0.26 (0.12 - 2.3)

Further, we would like to highlight that in Figure 1h and 4d and Supplementary Figure 1d, 3d, and 4d of the original manuscript, which also display expression levels of multiple transcripts, comparisons are made only between CD^{rem} and non-IBD within the same gene, not between different genes.

Although we did not aim to directly compare the expression levels of different immunoglobulin genes in Figure 4e and h, we understand the reviewer's concern that the use of a stacked bar graph might imply such comparability, which could indeed be misleading. As these stacked bar graphs did not provide additional information and may lead to misinterpretation, we have decided to remove them from Figure 4e and h. The underlying qPCR data for IGHA1 and IGHA2 are presented in separate box plots in Figure 4e, while the data for IGHM and the IGHG subclasses can be found in Supplementary Figures 3e and g, respectively, in the revised manuscript file.

(2) Several figures are presented in a way which does not allow any statistical analyses, even though the data itself would. For example figure 2h shows a compounds alteration in expression, with no information regarding distribution and thus no insight into the robustness of the data. Fig 3j is used to suggest increased expression of IL6, which is highly doubtful based on the image (seems mainly driven by 1 donor).

We thank the reviewer for their attentive comment. However, we would like to point out that there is no Figure 2h in the manuscript. Based on the context of the comment, we assume that the intended reference was to Figure 1h. If this interpretation is incorrect, we would greatly appreciate clarification regarding the figure panel.

In Figure 1h, the relative expression of B-cell transcripts in the isolated colonic LPMNCs of CD^{rem} patients is displayed compared to non-IBD controls. We here chose the heatmap representation to make group-level differences more accessible and visually intuitive for the readership. The two targets with a fold change greater than 2 (CD19 and CD38) are shown separately as individual data points next to the heatmap. The individual expression values of all less strongly regulated transcripts, including statistical comparisons, are provided in Supplementary Figure 1e of the original manuscript. In the revised manuscript, we have also added the individual P values for all transcripts in Supplementary Figure 1e. We suppose that the reviewer's comment may also refer to other heatmaps (Figure 4d and 6i of the original manuscript) and dot plots (Figure 2a of the original manuscript), in which individual values and statistical analyses are not immediately visible. We would like to emphasize that the corresponding individual data points are presented as box plots in Figure 2b and Supplementary Figures 1f, 3c, 3d, and 4g, in order to make data distribution and P values transparent to the reader.

With the heatmap in Figure 3j, our main intention was to demonstrate that the transcripts of cytokines and chemokines known to be important for B-cell differentiation and PC survival, such as IL4, IL6, IL10, IL21, TGFB1, and TNFRSF13, were not decreased in the colon of CD^{rem} patients compared to non-IBD controls. Rather, the majority of total quantified cytokines/chemokines was not differentially expressed, and six transcripts were even upregulated, suggesting that the observed defect in PC

maturation is not caused by a lack of microenvironmental cytokines. We understand the reviewer's impression that the elevated IL6 expression observed in the RNA-seq data may be driven primarily by a single patient, based on the appearance of the heatmap. However, the upregulation of IL6 in CD^{rem} compared to non-IBD colonic tissue is statistically significant, as depicted in Figure 22a. When excluding the patient with the highest IL6 expression (Figure 22b), there is still a strong trend towards upregulation with a P value of 0.0736. The reviewer's comment inspired us to include the individual data points for all six significantly elevated cytokines/chemokines from Figure 3j of the manuscript as box plots with corresponding P values (Figure 22c) in Supplementary Figure 2j in the revised manuscript

Fig. 22 | Upregulated cytokine and chemokine expression in the colonic mucosa of CD^{rem} patients. (a) Increased expression of IL6 in colonic biopsies from CD^{rem} compared to non-IBD individuals was determined using RNA-seq. (b) Excluding the highest value from the CD^{rem} group in (a) still shows a strong trend towards upregulation. (c) Significantly elevated expression of six cytokines/chemokines in the colon of CD^{rem} patients (n=7) versus non-IBD controls (n=12), as depicted in the heatmap in Figure 3j of the manuscript. (a)-(c) (Multiple) Mann-Whitney U test. *P≤.05. TPM, transcripts per million.

file. Respective changes in the text have been highlighted in yellow and added to the figure legend and the results section on page 10, line 2. We hope that the reviewer appreciates these changes to the manuscript.

(3) Since identification of cells in the spatial analysis is crucial, some images of the separation and analysis should be provided as supplementary information. Showing the staining of a single double/triple stained cells does not prove anything else than the fact one exists. For example, the number of T cells (or at least CD3+ cells) appears extremely low, and should be addressed.

We appreciate the reviewer's suggestion that a more detailed depiction of cell separation in the spatial single-cell proteomics experiments on the CosMxTM SMI would be beneficial for interpretation. In the methods section on page 22 lines 27-28 it is described:

'The positivity cut-off was set to 30 or 100, using mouse IgG1 and rabbit IgG isotype controls, respectively.'

We have now additionally included a graphical representation of the positivity threshold setting, exemplified on the total cells of a non-IBD control (Figure 23a), in Supplementary Figure 2c of the revised manuscript. Moreover, the reviewer refers to Figure 2g of the original manuscript, which shows only representative images of stained cells alongside the relative abundance of PBs and PCs. In response, we have now also included the full separation strategy for PBs and PCs in Supplementary Figure 2c, starting with total cells of a non-IBD control (Figure 23b).

The separation of PCs into the respective subpopulations based on CD19 and CD45 staining was already illustrated in Supplementary Figure 2f of the original manuscript.

Fig. 23 | Separation strategy of PBs and PCs using spatial single-cell proteomics data. (a) Setting of positivity cut-offs for mouse- and rabbit-derived antibodies using mouse IgG1 and rabbit IgG isotype control signals of total cells from a representative non-IBD control. **(b)** Stepwise separation strategy of PBs and PCs starting with total cells from a representative non-IBD control. FI, fluorescence intensity.

The reviewer correctly observed that the percentage of CD3⁺ cells within the Vim⁺ LP cells shown in Figure 1e of the original manuscript is very low, which appears contradictory to the substantially higher percentages of CD4⁺ and CD8⁺ T cells. Indeed, the CD3 antibody from the commercially available 64-plex Human Immuno-Oncology protein panel (NanoString/Bruker) did not perform optimally in our experiments. Visual inspection of the staining shows that CD3 binding is specific, but the signal intensity per cell is considerably lower compared to CD4 or CD8 (Figure 24a). The representative data of one non-IBD control display that 30-40% of total Vim⁺ LP cells are positive for CD4 or CD8, respectively, while only 0.3% of the cells exhibit a positive signal for CD3, which is comparable to the rabbit isotype control (Figure 24b). We are currently in exchange with the Spatial Biology department at Bruker to assess whether the CD3 antibody used for cell segmentation might interfere with the CD3 signal observed in the subsequent proteomics analysis. While Bruker has excluded this possibility based on their internal validation, we are continuing to investigate this together to rule out any potential technical artifacts.

Fig. 24 | Low CD3 signal in spatial single-cell proteomics of colonic tissue. (a) Representative image of CD3 staining (red) in a CD4⁺ (blue) and a CD8⁺ (cyan) T cell. **(b)** Percentages of CD3⁺, CD4⁺, and CD8⁺ cells within the Vim⁺ cells of a representative non-IBD control (right) compared to isotype control signals (left). FI, fluorescence intensity.

The manuscript by Raschdorf *et al.* provides a deep dive into the role of IgA+ plasma cells and blasts in Crohn's disease. Unconventionally, they target patients in remission (*i.e.*, without active inflammation). They apply bulk RNA sequencing, fluorescent microscopy, CosMX, mass-spec, *ex vivo* profiling, and *in vitro* mouse modelling to propose a cell-intrinsic defect in the terminal differentiation of CD19-CD45-CD138+ long-lived plasma cells. Increased CD38 expression and the employment of ketogenesis and glycolytic activity suggest OXPHOS deficiency in these cells. Together, the manuscript sheds light on altered IgA plasma cell profiles driving dysregulated barrier control in CD, rather than being simply in response to inflammation. The manuscript is well written and logical. The numbers of participants included in the study are not large (*i.e.*, 7 vs 12, 6 vs 5 and 6 vs 6), but the chosen methods are appropriate and executed effectively. I have only minor comments.

(1) To further support defective regulation of the microbiota in CD rem, the authors could consider B cell receptor repertoire analysis or analysis of IgA binding to faecal microbiota (e.g., by flow cytometry).

- The authors fully appreciate that the findings presented in this study raise additional questions regarding potential differences in IgA diversity and binding to intestinal bacterial species in CD^{rem} patients compared to non-IBD controls and how this may affect microbiota composition and homeostasis. However, a detailed analysis of the gut microbiota profile as well as the B-cell receptor repertoire is beyond the scope of this manuscript and will be addressed as part of future studies.

We here included fecal levels of IgA autoantibodies directed against glycoprotein 2 (GP2; variant 2 and 4), which acts as a bacterial receptor in the gut lumen. GP2 is produced by the exocrine pancreas (variant 2) and intestinal M cells (variant 4) and functions to limit mucosal invasion of adherent bacteria via FimH binding, thereby maintaining gut homeostasis²¹. In a previous study, we observed elevated serum levels of GP2-specific IgA in CD patients²². In contrast, while non-IBD controls exhibited substantial amounts of anti-GP2#2 IgA in stool samples, these levels were markedly reduced in CD^{rem} patients (Figure 25).

In addition to Figure 2e of the original manuscript, which demonstrates reduced levels of total fecal IgA, we here show that antigen-specific IgA responses against GP2#2 are likewise diminished in the colon of CD^{rem} patients.

Fig. 25 | Anti-GP2 variant 2 (GP2#2) and anti-GP2 variant 4 (GP2#4) IgA levels were quantified in feces samples of CD^{rem} patients and non-IBD controls using ELISA experiments. Two-way ANOVA with Šídák's multiple comparisons test. *** $P < 0.0001$. GP2, glycoprotein 2.

The presence of fecal anti-GP2#2 IgA in non-IBD controls may reflect a physiological, mucosal interaction with GP2, potentially contributing to immune tolerance towards high levels of luminal GP2 at the epithelial surface. Conversely, reduced levels in CD^{rem} patients may contribute to a loss of barrier function or microbial dysbiosis. As further studies are still under investigation by our group with the aim to decipher the role of reduced fecal anti-GP2 autoantibody levels on intestinal barrier homeostasis in CD patients, we have decided not to include these data in the current manuscript.

However, the reviewer's comment prompted us to address the potential impact of altered IgA diversity on the microbiome in a sentence within the discussion on page 16, lines 5-7 (highlighted in yellow):

'Potential additional changes in IgA diversity and their impact on host-microbiome interactions warrant further investigation in subsequent studies.'

(2) Methods page 17 line 18: Patients were classified as being in remission if the colon presented without macroscopic evidence of mucosal inflammation. Was this confirmed histologically?

- We thank the reviewer for drawing attention to this critical aspect. At this point, we would like to emphasize that our analyses focus on CD patients in clinical remission and without macroscopic evidence of colonic inflammation during endoscopy. We have clarified this accordingly in the methods section on page 18, line 15 of the revised manuscript file with further details on page 18, lines 18-20 (highlighted in yellow).

The classification of remission mentioned by the reviewer applies to ten CD patients for blood collection and sampling of colon biopsies during colonoscopy. While no pathology report was available for one patient (only blood), histological evidence of non-inflamed colonic mucosa was confirmed in eight patients. For one patient, 'granulomatous colitis' was described in the pathology report.

However, based on our analyzed parameters of the deeply phenotyped CD^{rem} patients (n=6), we could not find evidence of inflammation in this patient, in contrast to other patients with documented active disease. The plasma calprotectin (CRP) level was not increased (Figure 26a) and the number of isolated colonic LPMNCs was even relatively low compared to other patients in remission (Figure 26b). Further, the density of Vim⁺ cells per unit area of the colonic mucosa FFPE section was similar to that observed in other patients classified as being in remission (Figure 26c). Based on the combined data, we infer that this patient does not exhibit significant active inflammation. The reported findings may represent residual changes from prior inflammation, as mucosa in remission naturally differs from that of healthy individuals.

Of note, two patients with active CD were inadvertently omitted from the initial analysis of plasma CRP levels in the original manuscript and have now been included in Supplementary Figure 1d and Supplementary Table 4 of the revised manuscript. Additionally, we have identified a small error in the graph of the colonic LPMNC counts, where one active CD patient without colitis was inadvertently included, while another CD patient with endoscopically confirmed active colitis had been excluded. This has now been corrected in Supplementary Figure 1c and Supplementary Table 4 of the revised manuscript.

Fig. 26 | Classification of CD remission. (a) CRP levels were quantified in plasma samples from CD patients (remission or active) and non-IBD controls using liquid chromatography-coupled mass spectrometry (LC-MS). (b) Total number of MNCs isolated from seven colonic biopsies per CD patient (remission or active) or non-IBD control. (c) Absolute number of Vim⁺ cells per tissue area [mm²] quantified in colonic biopsy sections of CD^{rem} patients and non-IBD controls in the CosMx™ SMI. (a) to (c) The enlarged magenta data point in each graph corresponds to the patient with the described ‘granulomatous colitis’ in the pathology report. (a) and (b) Kruskal-Wallis test with Dunn’s multiple comparisons test, (c) Mann-Whitney U test.

In response to the reviewer’s comment, the authors decided to add the density of LP cells per area of colonic mucosa in the revised manuscript file as Supplementary Figure 1b to indicate histological absence of leukocyte infiltrates in the colonic mucosa of the deeply phenotyped CD^{rem} patients. Respective changes in the text have been highlighted in yellow and added to the results section on page 7, lines 22-24.

(3) Methods page 19 line 19: The ‘different combinations of cytokines’ used in the differentiation of PBs and PCs should be explicitly stated.

- Following the reviewer’s recommendation, the authors have added the sequential cytokines used for ex vivo PC generation on page 20, lines 28-29 and page 21, lines 1-7 of the revised manuscript (highlighted in yellow), including the respective manufacturers and catalogue numbers.

(4) Please include catalogue numbers for kits included throughout the methods section.

- The authors are thankful for the suggestion and included the catalogue numbers of all kits and specific reagents in the methods section of the revised manuscript (highlighted in yellow).

References:

1. Elmentaite, R. *et al.* Cells of the human intestinal tract mapped across space and time. *Nature* **597**, 250–255 (2021).
2. Wei, Z. *et al.* Serological assays to measure dimeric IgA antibodies in SARS-CoV-2 infections. *Immunol Cell Biol* **101**, 857–866 (2023).
3. Yagi, M. *et al.* p32/gC1qR is indispensable for fetal development and mitochondrial translation: importance of its RNA-binding ability. *Nucleic Acids Res* **40**, 9717–9737 (2012).
4. Hillman, G. A. & Henry, M. F. The yeast protein Mam33 functions in the assembly of the mitochondrial ribosome. *Journal of Biological Chemistry* **294**, 9813–9829 (2019).
5. Sünderhauf, A. *et al.* GC1qR Cleavage by Caspase-1 Drives Aerobic Glycolysis in Tumor Cells. *Front Oncol* **10**, (2020).
6. Raschdorf, A. *et al.* Heterozygous P32/C1QBP/HABP1 Polymorphism rs56014026 Reduces Mitochondrial Oxidative Phosphorylation and Is Expressed in Low-grade Colorectal Carcinomas. *Front Oncol* **10**, (2021).
7. Ursin, R. L. *et al.* Greater Breadth of Vaccine-Induced Immunity in Females than Males Is Mediated by Increased Antibody Diversity in Germinal Center B Cells. *mBio* **13**, (2022).
8. Fink, A. L., Engle, K., Ursin, R. L., Tang, W.-Y. & Klein, S. L. Biological sex affects vaccine efficacy and protection against influenza in mice. *Proceedings of the National Academy of Sciences* **115**, 12477–12482 (2018).
9. Lamason, R. *et al.* Sexual dimorphism in immune response genes as a function of puberty. *BMC Immunol* **7**, 2 (2006).
10. Skibbe, K. *et al.* Colorectal Cancer Progression Is Potently Reduced by a Glucose-Free, High-Protein Diet: Comparison to Anti-EGFR Therapy. *Cancers (Basel)* **13**, 5817 (2021).
11. Snapper, C. M., Finkelman, F. D. & Paul, W. E. Differential regulation of IgG1 and IgE synthesis by interleukin 4. *J Exp Med* **167**, 183–196 (1988).
12. Hashiguchi, M., Asatsuma-Okumura, T. & Iwai, Y. Interleukin 21 promotes IgG1+ plasma cell differentiation instead of class switching to IgE via Blimp1 expression. *Eur J Immunol* **54**, (2024).
13. Feagan, B. *et al.* Performance of Crohn’s disease Clinical Trial Endpoints based upon Different Cutoffs for Patient Reported Outcomes or Endoscopic Activity: Analysis of EXTEND Data. *Inflamm Bowel Dis* **24**, 932–942 (2018).
14. Sandborn, W. J. *et al.* Association Between Proposed Definitions of Clinical Remission/Response and Well-Being in Patients With Crohn’s Disease. *J Crohns Colitis* **16**, 444–451 (2022).
15. Khanna, R. *et al.* A retrospective analysis: the development of patient reported outcome measures for the assessment of Crohn’s disease activity. *Aliment Pharmacol Ther* **41**, 77–86 (2015).
16. Tejedor Vaquero, S. *et al.* Immunomolecular and reactivity landscapes of gut IgA subclasses in homeostasis and inflammatory bowel disease. *J Exp Med* **221**, (2024).
17. Huang, J. *et al.* Site-Specific Glycosylation of Secretory Immunoglobulin A from Human Colostrum. *J Proteome Res* **14**, 1335–1349 (2015).

18. Kanayama, M. & Luo, J. CD38-Induced Apoptosis and Mitochondrial Damage is Restored by Nicotinamide in Prostate Cancer. *Front Mol Biosci* **9**, (2022).
19. Camacho-Pereira, J. *et al.* CD38 Dictates Age-Related NAD Decline and Mitochondrial Dysfunction through an SIRT3-Dependent Mechanism. *Cell Metab* **23**, 1127–1139 (2016).
20. Chen, P.-M. *et al.* CD38 reduces mitochondrial fitness and cytotoxic T cell response against viral infection in lupus patients by suppressing mitophagy. *Sci Adv* **8**, (2022).
21. Kurashima, Y. *et al.* Pancreatic glycoprotein 2 is a first line of defense for mucosal protection in intestinal inflammation. *Nat Commun* **12**, 1067 (2021).
22. Derer, S. *et al.* Inflammatory Bowel Disease-associated GP2 Autoantibodies Inhibit Mucosal Immune Response to Adherent-invasive Bacteria. *Inflamm Bowel Dis* **26**, 1856–1868 (2020).

Reviewer #1

The authors have made significant changes to the manuscript and have addressed or provided reasonable explanations for most of the concerns I raised in the initial review. Indeed, the authors clearly went to great lengths to seriously engage with the reviewers' comments, which should be commended. The development of a dIgA ELISA in such a short timeframe is especially impressive.

The one potential difficulty still remaining is the extremely limited number of biological replicates in the mouse experiments, which the authors were able to only partially address. My understanding is that this is due to unfortunate and unpredictable circumstances and certainly not due to any lack of willingness by the authors to engage with the problem. At this point, the possible solutions would be to either remove the mouse data or to proceed with the small number of replicates to publication. On the whole, I very much favour the latter option, providing that the limitation inherent in the small sample sizes is clearly indicated in the text. The mouse data is not the primary focus of the manuscript but is still informative and provides a potential mechanistic explanation for the observed phenotype that is novel and interesting.

Ultimately, the final decision lies with the editors and adherence to the editorial policy of Nature Communications. However, from the point of view of scientific peer review I am happy to endorse the publication of the manuscript.

- We thank this reviewer for proposing two opportunities to handle the existing circumstances that prevent improving the limited number of biological replicates in the mouse experiments. In line with the reviewer's opinion, and considering that several of our analyses include a larger number of animals, we also support keeping the data from both mouse models in the manuscript.

For the Atp8-mutant versus B6-WT model, only the ex vivo stimulation of splenic B cells, the analysis of MLNs/PPs, and the quantification of colonic IgA⁺ cells were performed with n=3 mice per group. In contrast, the quantification of colonic B-cell and PC marker expression, the characterization of splenic tissue under chronic colitis, and the analysis of colonic IgA levels included a higher number of biological replicates. In the GFHPD versus chow diet mouse experiment, solely the measurement of fecal IgA was limited to n=3 mice per group, whereas the quantification of colonic IgA⁺ cells and the analysis of B-cell and PC marker expression in the colon were performed with larger sample sizes.

Despite the small sample size in some experiments, we believe these data clearly illustrate the importance of functional mitochondria and OXPHOS in humoral immunity. They complement the findings from the human samples and provide a potential mechanism for the impaired IgA⁺ PC response observed in CD patients. Additionally, the data support the concept of metabolically targeted therapeutic approaches.

In addition to the numbers of replicates reported in the legends of Figure 7 and Supplementary Figure 5, we have now clearly indicated the limited sample size (n=3) for the corresponding murine experiments in the results section on page 13, lines 7 and 22, and page 14, line 3 of the revised manuscript (highlighted in yellow). Moreover, we have now addressed the restricted number of biological replicates in the following sentence on page 17, lines 10-12 of the discussion section of the revised manuscript (highlighted in yellow):

'Accordingly, preliminary data from Atp8-mutant mice (n=3) suggested a reduction in mature B220⁺ CD138⁺ PCs in mesenteric lymph nodes and Peyer's patches, consistent with the impaired IgA-mediated humoral immunity observed in the colon.'

We hope that these amendments satisfactorily address the reviewer's comment.

Reviewer #2

The authors have resolved most of my concerns in their rather lengthy rebuttal. Some concerns remain, which can probably be handled textually.

Concern (1) - The use of samples including those with inflammation in the small bowel

The authors extensively describe their samples in their response, which largely satisfies my concerns regarding their patient selection. The only thing I would like to confirm before assigning these patients as 'in remission' is whether these patients previously showed colonic disease. I agree that someone with previous colonic involvement and currently only aftous lesions on the ileocecal anastomosis is in remission. However, to differentiate their study from earlier work, it is crucial these patients did in fact have colonic involvement at some point in their disease history. I am willing to assume that is the case, but it would be important to confirm this in the patient characteristics.

- The authors thank the reviewer for raising this comment and fully agree that, given the focus of our study on colonic tissue, it is crucial for the interpretation of the data that all included CD patients presented with evidence of colonic involvement at some point during the course of their disease. To address this comment, we have carefully reviewed the clinical histories of all analyzed patients again. Among the CD patients from whom biopsies and/or blood samples were obtained, all patients with active disease currently presented, or have previously presented, with colonic involvement. Of the ten CD patients in remission, nine have a documented history of colonic involvement, while for one patient no specific information regarding disease location is available.

In the cohort for stool sample collection (n=47 CD patients in remission; Figure 2e of the original manuscript), disease manifestation in the colon was reported for nearly all patients (n=44) except for three, in whom disease was documented only in the upper gastrointestinal tract or the ileum (Supplementary Table 1 of the original manuscript). We decided to remove these patients from the analysis and have revised the graph in Figure 2e of the updated manuscript accordingly (Figure 1). Further, we have corrected the number of CD remission patients included in the fecal IgA ELISA to n=44 in the results section on page 8, line 19, the legend of Figure 2e, Supplementary Figure 6, and Supplementary Table 1 of the revised manuscript (highlighted in yellow).

Fig. 1 | SigA levels were quantified in fecal samples from n=44 CD^{rem} patients with colonic involvement and n=85 non-IBD controls using IgA-specific ELISA experiments.

To clarify the disease location in the manuscript, we have revised the sentence on page 18, lines 14-15 of the methods section as follows (highlighted in yellow):

'Patients with a confirmed diagnosis of CD and colonic involvement were classified as being in remission or active disease based on clinical and endoscopic parameters.'

In addition, we have now included this information in the Supplementary Tables 1-4 summarizing patient characteristics (highlighted in yellow), with the one patient lacking information on disease location being referred to as 'Unknown'.

Concern (2) - The use of low sample numbers, which for sure has underpowered several analyses

I appreciate the fact that the study was focused on techniques not suitable for high throughput analysis (ie spatial transcriptomics) and would not require to increase sample size for these experiments. However, I would be very careful in stating anything regarding the RNASeq, which clearly is underpowered. Representation of the data is now improved, and I can agree to that. However, still any mention regarding the effect of medication here should be removed to the small group sizes.

The additional analysis of IgA based on medication does unfortunately not resolve matters. These data show no significant differences, but include many different treatments grouped together or still in too small numbers to provide meaningful information. Although not significant, for example anti-TNF vs other biologicals does appear to show a trend to higher IgA in the 'other' group. This may very well be explained by differences in initial disease activity (patients receiving other biologicals almost always have failed anti-TNF and thus suffer more refractory disease), but still confounds the data. Additionally, no patient in this sub-cohort appears to be using no biologic at all, making this an unusual patient cohort.

Since the authors do not focus on the effect of medication at all, I would remove most comments regarding this data, and at best state that although numbers were too small for statistical analysis, they did not find data suggesting an effect of the medication (if they want to mention medication at all cost).

- We acknowledge the reviewer's concern regarding the limited sample size in the RNA-seq analysis and agree that the cohort is underpowered to draw conclusions about the effect of medication on gene expression profiles.

Therefore, we have removed the graphs displaying IGHA1/2 expression stratified by medication (Supplementary Figure 1g) as well as any RNA-seq data interpretation related to medication from page 7, line 8, and page 8, lines 11-13 of the first revised manuscript. We have chosen to keep the PCA plot displaying bulk RNA-seq data of colonic tissue (Supplementary Figure 1a of the first revised manuscript)

Fig. 2 | PCA displaying results from bulk RNA-seq (standardized TPM values of the 1,000 most variable genes) of colonic biopsies from CD^{rem} patients (n=7) and non-IBD controls (n=12).

to illustrate that overall transcriptional profiles did not differ substantially between CD^{rem} patients and non-IBD controls. However, we have now removed the medication-based coloring from the PCA plot (Figure 2), as well as any related statements regarding different medications in the text.

We also recognize the reviewer's concern regarding the analysis of fecal IgA levels grouped by medication in Supplementary Figure 1j of the first revised manuscript. We agree that the number of samples per treatment group is too small to allow for a meaningful analysis, and that any conclusions derived from grouping different treatments together are inherently limited. Additionally, the reviewer raised a very valid point that anti-TNF α agents are often used as first-line therapy, whereas other biologics are typically prescribed in cases of refractory disease. This treatment hierarchy introduces potential confounding, which could influence fecal IgA levels independently of the medication itself. In response to the reviewer's comment, we removed this graph and the corresponding interpretation from the results section on page 8, line 20 of the first revised manuscript.

The following sentence in the discussion on page 17, lines 12-14 of the first revised manuscript addresses that future studies with larger patient numbers are required to analyze the effects of different medications on the observed phenotypes in more detail:

'Overall, the influence of therapeutic regimens on the dysregulated humoral immune response and metabolism in CD patients requires further investigation in follow-up studies involving larger patient cohorts.'

Concern (3) - The data regarding metabolomics was mainly associative and lacked functional assays

I appreciate the initial focus of the study did not allow for the inclusion of for example Seahorse experiments and the added metabolomic analysis did add support for the claims of the authors. I would appreciate some statements that this requires further study etc in the discussion.

- We thank the reviewer for pointing out the missing perspective on the need for additional metabolic analyses on colonic PCs in CD patients in future studies. To address this in the discussion, we have now added the following sentence on page 17, lines 12-14 of the revised manuscript (highlighted in yellow):

'Decreased expression of mitochondrial OXPHOS-related genes, accompanied by reduced L-lactate secretion, suggests a quiescent metabolic state in CD^{rem}-derived colonic PCs, highlighting the need to investigate their metabolic profile in more detail.'

We hope this aligns with the reviewer's expectations.